# NSD2 is a requisite subunit of the AR/FOXA1 neo-enhanceosome in promoting prostate tumorigenesis

Abhijit Parolia [1,2,3,4] ✉, Sanjana Eyunni[1,2,5,13], Brijesh Kumar Verma[6,13], Eleanor Young [1], Yihan Liu[1,7], Lianchao Liu[8], James George [1], Shweta Aras[6], Chandan Kanta Das[6], Rahul Mannan [1,2], Reyaz ur Rasool[6], Erick Mitchell-Velasquez[6], Somnath Mahapatra [1,2], Jie Luo[1,2], Sandra E. Carson [1], Lanbo Xiao[1,2], Prathibha R. Gajjala[1,2], Sharan Venkatesh[6], Mustapha Jaber [1], Xiaoju Wang[1,2], Tongchen He[1], Yuanyuan Qiao [1,2], Matthew Pang[1], Yuping Zhang[1,2], Jean Ching-Yi Tien[1,2], Micheala Louw[2], Mohammed Alhusayan[6], Xuhong Cao[1,2,9], Fengyun Su[1,2], Omid Tavana[10], Caiyun Hou[8], Zhen Wang[8], Ke Ding[8], Arul M. Chinnaiyan [1,2,3,4,9] ✉ & Irfan A. Asangani [6,11,12] ✉

Androgen receptor (AR) is a ligand-responsive transcription factor that drives terminal differentiation of the prostatic luminal epithelia. By contrast, in tumors originating from these cells, AR chromatin occupancy is extensively reprogrammed to activate malignant phenotypes, the molecular mechanisms of which remain unknown. Here, we show that tumor-specific AR enhancers are critically reliant on H3K36 dimethyltransferase activity of NSD2. NSD2 expression is abnormally induced in prostate cancer, where its inactivation impairs AR transactivation potential by disrupting over 65% of its cistrome. NSD2-dependent AR sites distinctively harbor the chimeric FOXA1:AR half-motif, which exclusively comprise tumor-specific AR enhancer circuitries defined from patient specimens. NSD2 inactivation also engenders increased dependency on the NSD1 paralog, and a dual NSD1/2 PROTAC degrader is preferentially cytotoxic in AR-dependent prostate cancer models. Altogether, we characterize NSD2 as an essential AR neo-enhanceosome subunit that enables its oncogenic activity, and position NSD1/2 as viable co-targets in advanced prostate cancer.

Prostate cancer (PCa) is the most commonly diagnosed malignancy in North American men, with over 95% of the primary disease expressing the androgen receptor (AR) protein[1]. AR is a transcription factor that dimerizes and shuttles into the nucleus upon binding to its ligand (that is, androgen), where it activates the expression of genes that drive terminal (that is, nonproliferative) differentiation of luminal epithelial cells. In concert with chromatin and epigenetic regulatory proteins, AR primarily binds at distal *cis*-regulatory sites (also known as enhancers) containing a canonical androgen response element (ARE) that comprises a 15-bp palindromic DNA sequence with two invertedly oriented hexameric 5'-AGAACA-3' half-sites[2], separately recognized by each half of the AR homodimer[3].

In PCa cells, AR activity is extensively reprogrammed to enable and maintain malignant phenotypes[4–6]. Consequently, the androgen/AR axis is the primary target of all therapies following surgical resection or radiation of the organ-confined disease[7]. This acute dependency on AR

**Fig. 1 | Epigenetics-focused CRISPR screen shows NSD2 as an AR coactivator.** **a**, Schematic of the epigenetic-targeted CRISPR screen using LNCaP-mCherry-KLK3 AR reporter lines. **b**, *Left*: mCherry immunofluorescence images of LNCaP reporters treated with labeled epigenetic drugs. *Right*: Barplot showing quantification of the mCherry signal from treated reporter cells normalized to the DMSO treatment (n = 3 biological replicates). Mean ± standard error of the mean (s.e.m.) are shown. Scale bar: 200 μm. **c**, sgRNA enrichment rank plot based on guide RNA ratio in mCherry-LOW to mCherry-HIGH cells. **d**, Immunoblots of listed proteins upon treatment with control (siNC) or NSD2-targeting (siNSD2) siRNAs. Total H3 is used as loading control. LNCaP lysates were collected at day 15. VCaP lysates were collected at day 10 or 15 after treatment. **e**, Representative protein map of NSD2-Long (NSD2-L) and NSD2-Short (NSD2-S) isoforms. HMG: High mobility group; PHD: Plant homeodomain. **f**, Immunoblots of noted proteins in CRISPR-mediated stable knockout (KO) of both NSD2 isoforms or NSD2-L alone. Total H3 is used as loading control. **g**, Gene set enrichment analysis (GSEA) plots for AR and E2F upregulated genes using the fold-change rank-ordered

genes from the NSD2 knockout (KO) vs wild-type (WT) LNCaP cells. DEGS, differentially expressed genes (n = 2 biological replicates; GSEA enrichment test). **h**, Immunoblots of listed proteins in NSD2-KO LNCaP cells stimulated with 10 nM DHT. **i**, GSEA plots of AR hallmark genes in NSD2 wild-type (WT) vs knockout (KO) LNCaP cells using the fold-change rank-ordered genes from DHT (10 nM for 24 h) vs DMSO treatment. DEGS, differentially expressed genes (n = 2 biological replicates; GSEA enrichment test). **j**, Representative immunohistochemistry (IHC) images of NSD2 in prostatectomy patient specimens. Scale bar: 100 μm. **k**, NSD2 signal intensity from IHC staining in panel j (n = 4 patient tumors; two-sided *t*-test). Box plot center, median; box, quartiles 1-3, whiskers, quartiles 1-3 ± 1.5× interquartile range, dot, outliers. **l**, Representative multiplex immunofluorescence (IF) images of KRT8, AR, and NSD2 in benign prostate, primary PCa or mCRPC patient specimens. Scale bar: 5 μm. **m**, Quantification of NSD2 IF signal intensity per KRT8+ luminal epithelial cell from images in panel l (two-sided t-test; Normal=39, primary PCa = 145, mCRPC=381 nuclei). Box plot center, median; box, quartile 1-3; whiskers, 10th and 90th percentile; dot, outliers.

activity is further reinforced in relapsed metastatic castration-resistant PCa (mCRPC) through activating mutations or copy amplification of AR or its cofactors[8–12]. Seminal studies profiling the AR cistrome in primary PCa uncovered de novo genesis of enhancers in the malignant state (that is, neo-enhancers), resulting in a two- to threefold expansion of the AR enhancer circuitry[5,6,13–15]. This process engenders an acute dependency on chromatin-binding AR cofactors, such as SWI/SNF, BRD4, MED1 and p300/CBP, all of which have been independently assessed for therapeutic druggability in mCRPC[16–22]. Yet, the molecular mechanisms underlying chromatin redistribution of AR upon transformation or distinctive subunits of the AR transcriptional complex that assembles at neo-enhancer elements (that is, the neo-enhanceosome) are poorly studied and, thus, unexplored for therapeutic targetability.

In this study, using an epigenetics-targeted functional CRISPR screen, we identified nuclear receptor binding SET domain protein 2 (NSD2, also known as MMSET, WHSC1) as a subunit of the AR enhanceosome complex in PCa cells. NSD2 is a histone 3 lysine 36 (H3K36) mono- and dimethyltransferase that activates gene expression by protecting the chromatin from accumulating repressive epigenetic marks, such as H3K27me3 (refs. 23–25). NSD2 is a bona fide oncogene in hematologic cancers and harbors recurrent activating alterations in over 15–20% of multiple myeloma[26–28] and 10% of childhood acute lymphoblastic leukemia[29–31].

In PCa, we found NSD2 to be exclusively expressed in the transformed cells—with no detectable expression in the normal epithelia—where it directly interacts with AR to enable its binding at chimeric AR half-motifs in concert with FOXA1 or other driver oncogenes. Inactivation of NSD2 entirely disrupted AR binding at over 65% of its tumor cistrome, importantly without affecting AR protein levels, and attenuated hallmark cancer phenotypes. NSD2 deficiency also engendered an increased dependency on NSD1, positioning the two paralogs as a digenic dependency. Concordantly, a dual NSD1/2 PROTAC degrader, called LLC0150, showed selective potency in AR-dependent as well as

NSD2-altered human cancers. These findings mechanistically explain how AR gets reprogrammed, away from prodifferentiation physiological functions, to instead fuel PCa growth and survival, and offer NSD1 and NSD2 as therapeutic vulnerabilities in the advanced disease.

## Results

### Functional CRISPR screen reveals NSD2 as an AR coactivator

Conventional plasmid-based reporter systems fail to capture intricate epigenetic or chromatin-level regulation of gene expression as they lack the native histone composition or higher-order chromosomal structure. Thus, we engineered an endogenous AR reporter system by using the CRISPR/Cas9 and homologous recombination methodologies. We edited the *KLK3* gene (also known as prostate-specific antigen, *PSA*) locus in AR-driven LNCaP cells to knock-in the mCherry coding sequence directly downstream of the endogenous promoter and fused in-frame via an endopeptidase sequence to the *KLK3* gene (Fig. 1a and Extended Data Fig. 1a–c). In the monoclonal reporter cell line, akin to PSA, mCherry expression is directly regulated by the AR transcriptional complex (Extended Data Fig. 1d–f) and, most importantly, captures chromatin or epigenetic-level changes in AR transactivation potential. Like PSA, mCherry expression was attenuated upon pharmacologic inhibition of coactivators like BRD4 (ref. 16), SWI/SNF[18] or P300/CBP[19] while increasing upon inhibition of the repressive PRC2/EZH2 complex[32,33] (Fig. 1b and Extended Data Fig. 1g). Using these endogenous AR reporter cell lines, we carried out a functional CRISPR screen, wherein we treated the cells with a custom single guide RNA (sgRNA) library targeting druggable transcriptional cofactors[34] (Extended Data Fig. 1h) for 8 days, stimulated with DHT for 16 h and FACS-sorted into mCherry^HIGH and mCherry^LOW populations. Genomic sgRNAs were sequenced and the ratio of normalized counts in mCherry^LOW to mCherry^HIGH cell populations was used to rank individual sgRNAs. Here, ranked alongside BRD4 (ref. 16) and TRIM24 (refs. 35,36), we identified NSD2 as an AR coactivator (Fig. 1c and Extended Data Fig. 1i). In contrast, subunits of the PRC2 complex, namely EZH2 and JARID2, that repress

**Fig. 2 | NSD2 expands the AR neo-enhancer circuitry to include chimeric AR half-sites.** **a**, Venn diagram showing overlaps of AR ChIP-seq peaks in NSD2 wild-type (WT) and knockout (KO) LNCaP cell lines. **b**, Genomic location of NSD2-dependent and independent AR sites defined from the overlap analysis in panel a. **c**, ChIP-seq read-density heatmaps of AR, FOXA1, and H3K27ac at top 1,000 AR enhancer sites in LNCaP NSD2 WT and KO cell lines. **d**, Top five known HOMER motifs enriched within NSD2-dependent and independent AR sites in LNCaP cells (HOMER, hypergeometric test). **e**, ChIP-seq read-density tracks of AR and H3K27ac in NSD2 WT and KO LNCaP cell lines. HOMER motifs detected within AR peaks are shown below with gray boxes highlighting NSD2-dependent and independent AR elements. **f**, Fold-change heatmap of HOMER motifs enrichment within AR binding sites specific to HOXB13, FOXA1 or FOXA1 + HOXB13 overexpression in LHSAR cells (data from Pomerantz et al.[5]).

**g**, Fold-change and significance of HOMER motifs enriched within primary PCa-specific AR sites over normal AR enhancers (data from Pomerantz et al.[5]; HOMER, hypergeometric test). **h**, AR ChIP-seq read-density box plot at sites containing the ARE or the FOXA1:AR chimeric motif in primary normal and tumor patient samples (normal prostate, n = 7; primary PCa, n = 13; mCRPC, n = 15). In box plots, the center line shows the median, box edges mark quartiles 1-3, and whiskers span quartiles 1-3 ± 1.5× interquartile range (one-way ANOVA). **i**, Rank-ordered plot of AR super-enhancers (HOMER ROSE algorithm) in NSD2 WT and KO LNCaP cells with select known AR target genes shown. **j**, Box plot of AR super-enhancer scores (HOMER ROSE algorithm) of top 100 *cis*-elements in NSD2 WT or KO LNCaP cells (two-sided t-test). Box plot center, median; box, quartiles 1-3; whiskers, quartiles 1-3 ± 1.5× interquartile range; dot, outliers.

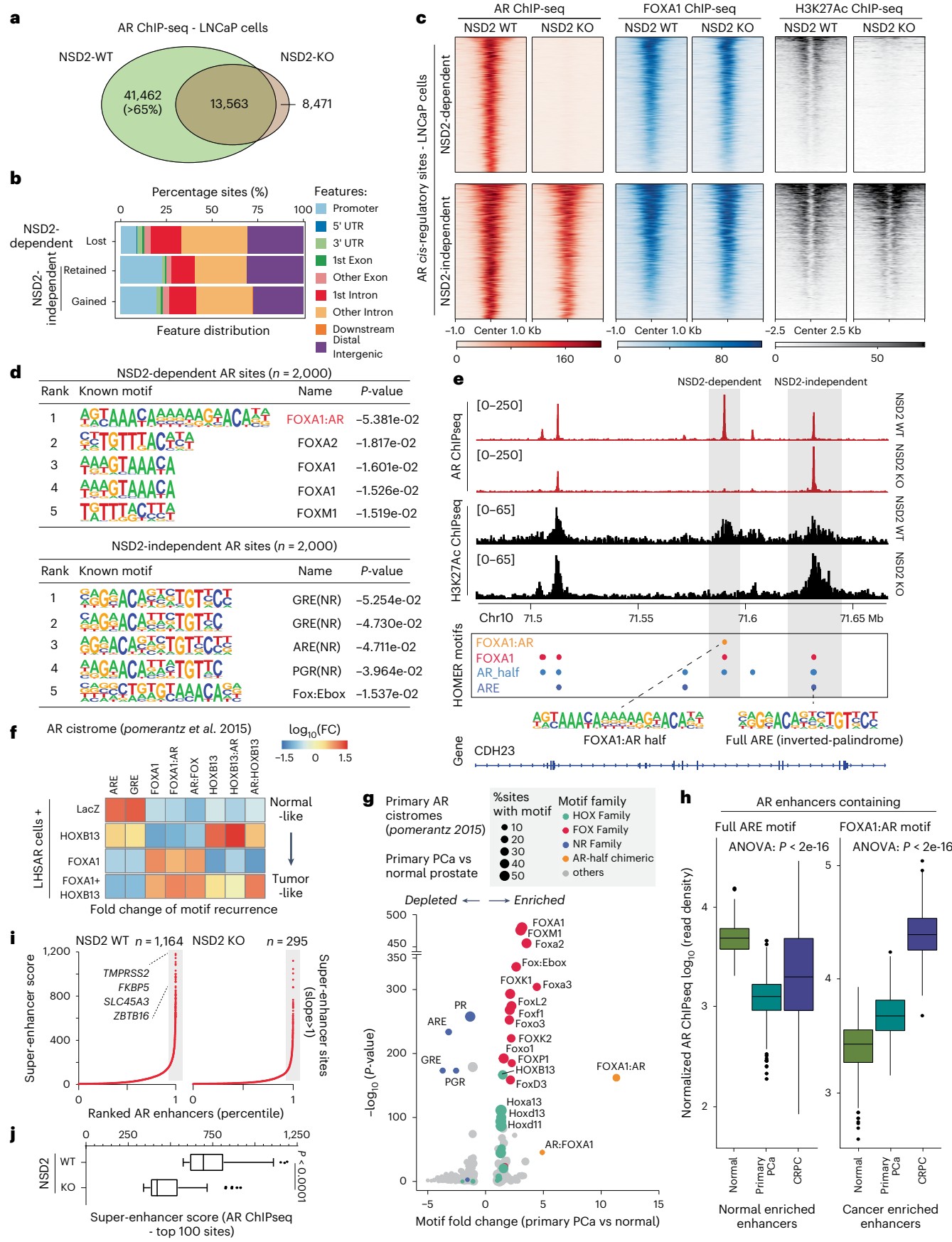

AR activity[32] were enriched in the mCherry[HIGH] cells. Validating the screening results, siRNA-mediated knockdown of NSD2 attenuated the expression of PSA/KLK3 in PCa cell lines (Fig. 1d).

The NSD2 gene templates two splice isoforms producing a long, catalytically active form (hereafter referred to as NSD2-L) as well as a truncated shorter isoform (called NSD2-S) containing only the reader and protein-protein interacting PWWP and HMG domains, respectively. We found both NSD2 isoforms to be robustly expressed in PCa cells (Fig. 1e and Extended Data Fig. 2a). Deletion of NSD2-L alone strongly attenuated the expression of AR target genes in LNCaP cells, which was comparable to complete loss of the NSD2 protein (Fig. 1f). The transcriptomic analysis further showed global AR activity to be significantly dampened in NSD2-deficient LNCaP cells with a parallel loss in hyperproliferative gene expression programs (Fig. 1g). AR and NSD2 transcriptional activities were also positively correlated in primary prostate tumors from the TCGA cohort (Extended Data Fig. 2b; $R = 0.68$, $P = 2.2 \times 10^{-16}$). Notably, there was no change in the abundance of AR transcript or protein itself in NSD2-deleted cells (Fig. 1f and Extended Data Fig. 2c), yet stimulation with DHT failed to significantly up-regulate the expression of AR target genes (Fig. 1h,i and Extended Data Fig. 2c,d).

To date, several studies have implicated NSD2 in PCa[37–41]; however, it is worth noting that these studies were focused on the AR-negative disease. Using tissue microarrays, these studies showed NSD2 protein to be elevated in cancer specimens, showing a stage-wise increase from primary to mCRPC or neuroendocrine PCa[39,40]. Building on these findings, in primary prostatectomy specimens we found NSD2 levels to be undetectable in the normal or adjacent benign foci with marked gain in expression in malignant cells (Fig. 1j,k). Consistent with this, in single-cell RNA-seq data from patient tumors, we found the *NSD2* transcript to be exclusively expressed in the AR[+] luminal epithelial cells (Extended Data Fig. 2e). Pseudo-bulk analyses confirmed NSD2 expression to be markedly elevated in the matched tumor vs the normal luminal compartment ($n = 18$), and NSD2 expression positively correlated with Gleason score of the primary disease (Extended Data Fig. 2f,g). Multiplex immunofluorescence in additional prostatectomy and patient tumor specimens further confirmed the KRT8[+]/AR[+] normal epithelial cells to have no detectable expression of NSD2, which was robustly expressed in the transformed epithelial cells (Fig. 1l,m and Extended Data Fig. 3a–e). Altogether, these data suggest that NSD2 is abnormally expressed in the transformed prostate luminal epithelial cells, wherein its methyltransferase function is critical for maintaining transcriptional activity of the AR complex.

## NSD2 activates neo-enhancers with chimeric AR half-motifs

Given that NSD2 loss had no impact on the abundance of the AR protein, we next profiled AR binding on chromatin. AR chromatin immunoprecipitation with sequencing (ChIP-seq) in NSD2-deficient LNCaP cells showed a dramatic and complete off-loading of the AR protein from over 40,000 genomic sites that comprise over 65% of the tumor cistrome (Fig. 2a). The majority of the lost sites (that is, NSD2-dependent) were within intronic or intergenic regions associated with *cis*-regulatory DNA elements (Fig. 2b). At these sites there was no change in the binding of FOXA1 upon NSD2 inactivation. Yet, disruption of AR binding was sufficient to trigger loss of the H3K27ac mark that demarcates active enhancers (Fig. 2c). In contrast, AR remains bound at over 20,000 genomic sites independent of NSD2, which also retained the H3K27ac active mark in the NSD2-null PCa cells (Fig. 2a–c). Next, ChIP-seq-based profiling of the chemical chromatin state showed NSD2-dependent AR sites to have higher abundance of H3K36me2 as well as active enhancer-associated H3K4me1/2 and H3K27ac modifications compared to the NSD2-independent elements (Extended Data Fig. 3d–f). Contrastingly, NSD2-independent AR sites had higher levels of the PRC2/EZH2 catalyzed repressive H3K27me3 mark. More importantly, NSD2 inactivation led to a significant decrease in H3K36me2 levels at the NSD2-dependent sites, reducing it to the levels at NSD2-independent sites in the wild-type cells, with a parallel increase in H3K27me3 (Extended Data Fig. 3g,h).

Motif analyses (HOMER[42]) of the NSD2-dependent AR sites identified a chimeric motif comprising a FOXA1 element juxtaposed to the AR half site (called FOXA1:AR half-motif) as the most significantly enriched DNA sequence (Fig. 2d), with 40% of these enhancers harboring this motif (Supplementary Tables 1 and 2). In contrast, NSD2-independent AR sites, a large fraction of which showed increased AR binding upon NSD2 inactivation, housed the palindromic ARE (Fig. 2d). These distinct modes of AR DNA interaction were evident within a Chr10 gene locus, wherein the loss of NSD2 completely disrupted AR binding at the FOXA1:AR half-motif, without affecting AR's interaction with a canonical ARE element in *cis*-proximity (Fig. 2e). Furthermore, custom motif analyses showed enrichment of other transcription factor motifs, including HOXB13 and ETS, within 25 bp of AR half elements detected within the NSD2-dependent AR sites (Extended Data Fig. 4a). We also custom-assembled chimeric AR half-motifs with FOXA1 and HOXB13 elements (in both 5′ and 3′ confirmations, see Methods) and interrogated their recurrence in published AR ChIP-seq data derived from non-cancerous LHSAR cells[5]. Here, we found the overexpression of FOXA1 and HOXB13 alone, or in combination, to markedly shift the AR cistrome away from full AREs (normal-like) towards chimeric AR half elements in the tumor-like state (Fig. 2f and Extended Data Fig. 4b). Strikingly, motif analysis of the AR cistromes generated from patient specimens[5,6] revealed the FOXA1:AR half-motif to be exclusively detected in the tumor-specific AR enhancer circuitries, with such chimeric motifs being essentially absent at normal AR sites (Fig. 2g and Extended Data Fig. 4c–f). In these analyses, we also found palindromic AREs to be depleted within cancer-specific enhancers (Fig. 2g). Concordantly, AR ChIP-seq signal at ARE sites was strongest in normal

**Fig. 3 | NSD1 and NSD2 independently enable oncogenic AR activity. a**, *Left*: Growth curves of cells treated with control (siNC) or NSD2-targeting siRNAs (n = 6 biological replicates; two-sided t-test). *Right*: Growth curves of NSD2 knockout (KO) or wild-type (WT) cells ($n = 3$ biological replicates; two-sided *t*-test). Mean ± s.e.m. are shown. **b**, *Left*: Boyden chamber images of NSD2-KO and WT cells. Scale bar: 500 μm. *Right*: Quantification of fluorescence signal ($n = 3$ biological replication; one-way ANOVA + Tukey's test). Mean ± s.e.m. are shown. **c**, *Left*: Representative images of NSD2-KO and WT 22RV1 cell colonies (n = 3 biological replicates). Scale bar: 1 cm. *Right*: Staining intensity of cell colonies (two-sided t-test). Mean ± s.e.m. are shown. **d**, Reverse Kaplan-Meier plot of tumor grafting of 22RV1 WT, NSD2-KO, or NSD2-KO + NSD2-L cells. **e**, Tumor volumes of 22RV1 NSD2-KO + NSD2-L-FKBP12[F36V] xenografts ± dTAGv-1 treatment. Mean ± s.e.m. are shown ($n = 10$ biological replicates; two-sided t-test). **f**, Immunoblots of listed proteins in whole-cell or chromatin fractions of LNCaP NSD2-FKBP12[F36V] cells ± dTAG-13. **g**, Schematic of coimmunoprecipitation (coIP) protein fragments. Dashed red box marks interacting domains. *Inset*: AR-NSD2 co-IP interaction summary. Red circles, interaction. Gray circles, no detectable binding. **h**, *Left*: co-IP immunoblots of AR DNA-binding domain (DBD) with HA-NSD2-HMG mutants. TM, triple mutant. *Right*: co-IP immunoblots of wheatgerm-purified Halo-AR-DBD with His-NSD2-HMG fragments. Input fractions are shown as control. **i**, GSEA plots for AR and MYC target genes in NSD1 KO vs WT LNCaP cells. DEGS, differentially expressed genes ($n = 2$ biological replicates; GSEA enrichment test). **j**, Immunoblots of labeled proteins upon treatment with siNC or NSD1 and/or NSD2 targeting siRNAs (siNSD1 or siNSD2). H3 is a loading control. **k**, *Top*: GSEA enrichment scores of EZH2/PRC2-repressed genesets in siNSD1 versus siNC-treated cells. *Bottom*: GSEA enrichment scores of PCa-specific EZH2 signature in siNSD1 and/or siNSD2 vs siNC-treated cells. **l**, Immunoblots of noted proteins in siNSD1 and/or siNSD2 treated cells ± EPZ-6438. **m**, Immunoblot of listed proteins in siNC or siNSD1 and/or siNSD2 treated cells. **n**, *Left*: Growth curves of cells treated with siNC, siNSD1 or siNSD1 + NSD2. *Right*: Growth curves of control (sgNC) or NSD1-deficient (sgNSD1) cells ± siNSD2 treatment ($n = 5$ biological replicates; two-sided *t*-test). Mean ± s.e.m. are shown.

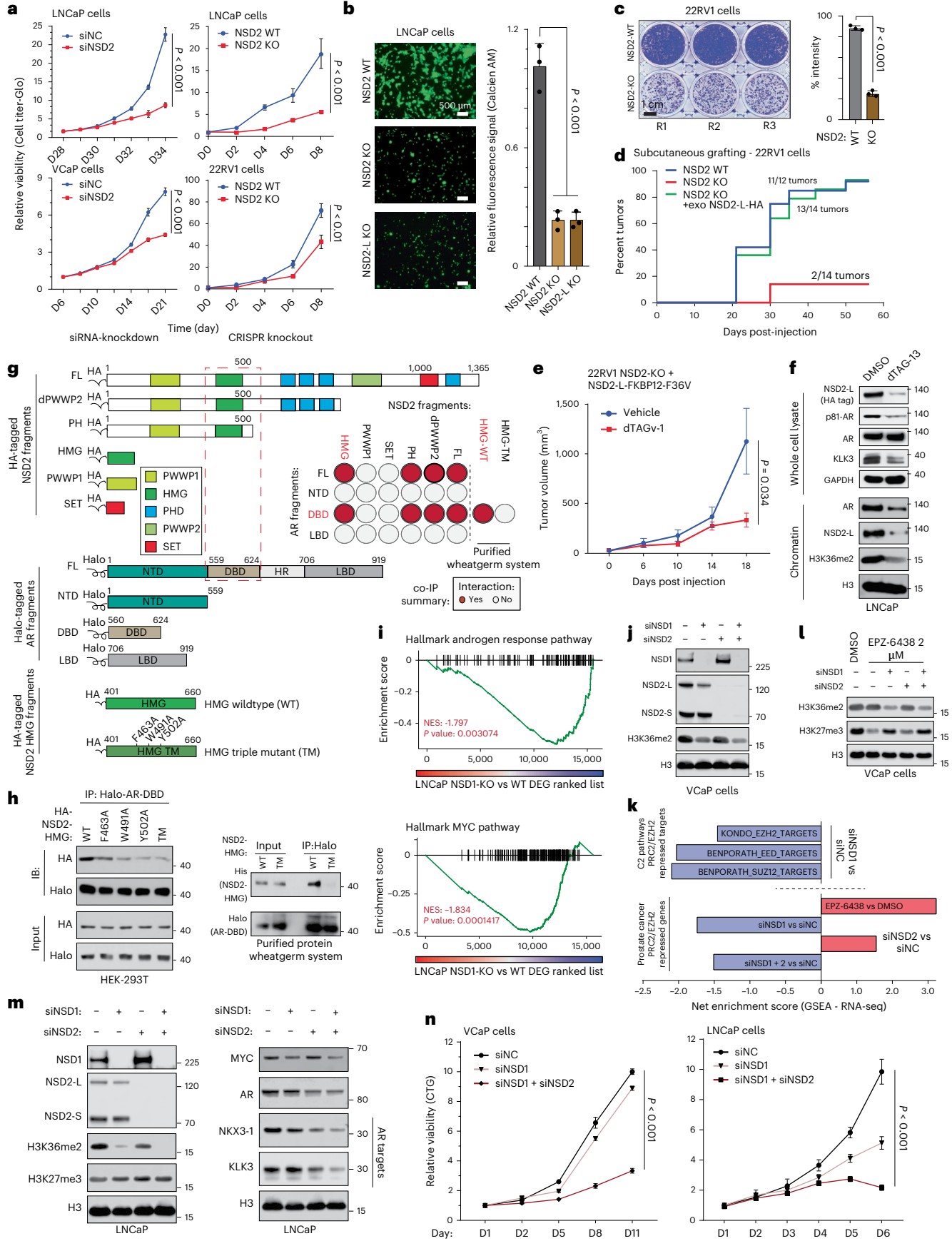

prostate tissues, whereas enhancers containing the chimeric FOXA1:AR half-motif had higher AR binding in PCa specimens (Fig. 2h). H3K27ac ChIP-seq signal from matched tumors showed similar redistribution, with FOXA1:AR half-sites being strongly activated in mCRPC tumors (Extended Data Fig. 4g).

In tumor cells, the aberrant expression of oncogenes is frequently amplified through dense clusters of closely spaced enhancers, often referred to as super-enhancers[43]. NSD2 inactivation resulted in the loss of over 75% of the AR-bound super-enhancers in PCa (Fig. 2i), including those that are hijacked by activating translocations[44] (Extended Data Fig. 4h). The residual super-enhancers also showed a significant decrease in the enhancer strength in the NSD2-null relative to the wild-type LNCaP cells (Fig. 2j). Altogether, these data suggest that, upon ectopic expression, NSD2 assists oncogenic transcription factors (namely FOXA1 and HOXB13) in expanding the AR enhancer circuitry to include chimeric AR half-sites that constitute over two-thirds of PCa AR cistromes.

## NSD1 and NSD2 independently enable oncogenic AR activity

Given NSD2 inactivation resulted in disruption of the cancer-specific AR cistrome, we set out to phenotypically characterize NSD2-deficient PCa cells. Here, siRNA/shRNA knockdown or CRISPR knockout of NSD2 significantly impaired hyperproliferative ability of AR-positive PCa cell lines (Fig. 3a and Extended Data Fig. 5a,b). NSD2-deficient cells also lost their ability to invade through Matrigel (Fig. 3b) and form colonies starting from single cells in clonogenic assays (Fig. 3c and Extended Data Fig. 5c). NSD2-null 22RV1 cells also lost their ability to graft when injected subcutaneously in NOD/SCID mice (Fig. 3d). Strikingly, exogenous reintroduction of NSD2-L restored the xenografting potential (green line, Fig. 3d), with resulting tumors growing at a rate comparable to those established with the parental wild-type cells (Extended Data Fig. 5d). NSD2-L re-expression also restored the invasive ability (Extended Data Fig. 5e), along with restoring the expression of KLK3 (Extended Data Fig. 5f). In the same experiment, re-expression of NSD2 variant lacking the SET domain (dSET) failed to rescue KLK3 (Extended Data Fig. 5g), whereas expression of the hyperactive NSD2-E1099K SET-domain mutant[29] completely restored KLK3 levels (Extended Data Fig. 5h). Next, we engineered the NSD2-null 22RV1 cells to stably express dTAG-version of the NSD2-L protein fused to the FKBP12[F36V] tag[45], which is rapidly degraded upon treatment with an FKBP12 degrader (Extended Data Fig. 5i). The 22RV1 NSD2-KO+ NSD2-L-FKBP12[F36V] cells successfully grafted and grew to form tumors in vivo; however, dosing of host animals with an FKBP12 degrader significantly diminished the growth of tumor xenografts (Fig. 3e and Extended Data Fig. 5j). Even at the molecular level, degradation of the exogenous NSD2-L-FKBP12[F36V] protein resulted in lower levels of KLK3 and chromatin-bound (that is, p-S81) AR without a decrease in total AR expression (top panel, Fig. 3f). Concordantly, chromatin fractionation in these cells showed a marked loss of AR binding with a parallel decrease in H3K36me2 upon NSD2-L-FKBP12[F36V] degradation (bottom panel, Fig. 3f). In LNCaP NSD2-dTAG models, NSD2 degradation led to downregulation of multiple AR target genes in a time-dependent manner (Extended Data Fig. 5k). Also, other genes encoded in *cis*-proximity of NSD2-dependent AR sites (see Methods) were similarly downregulated in the NSD2-deficient LNCaP cells (Extended Data Fig. 6a and Supplementary Table 4), and were enriched for oncogenic KRAS, angiogenesis, and G2M checkpoint pathways (Extended Data Fig. 6b). In contrast, genes associated with NSD2-independent AR sites were enriched for developmental pathways and AR/NKX3-1 signaling (Extended Data Fig. 6c). These findings position NSD2 as a molecular 'switch' that activates oncogenic AR cistrome and enables hallmark cancer properties.

Next, in size exclusion chromatography we found NSD2 to co-elute with higher-order AR transcriptional complexes (Extended Data Fig. 7a). NSD2 also co-precipitated with AR in several PCa cell lines (Extended Data Fig. 7b). As previously reported[46], using fragment-based coimmunoprecipitation, we confirmed the high mobility group box (HMG-box) domain of NSD2 to interact with the DNA-binding domain (DBD) of AR (Fig. 3g and Extended Data Fig. 7c,d). Furthermore, alanine substitution of three highly conserved HMG-box residues (that is, F463/W491/Y502A) individually or together (triple mutant, Fig. 3g) disrupted its interaction with the AR-DBD in the ectopic HEK293T as well as cell-free purified wheatgerm extract systems (Fig. 3h). This finding suggests that NSD2 directly, and independent of DNA, interacts with the AR-DBD through its HMG-box domain, which is notably absent in other NSD family histone methyltransferases.

Despite a striking loss of neoplastic features, NSD2-deficient PCa cells remained viable. Thus, we speculated if NSD2 paralogs could sustain AR activity through alternative mechanisms. To test this, we knocked-out NSD1 or NSD3 individually in LNCaP and assessed its transcriptional impact. Unlike NSD3, NSD1 loss significantly attenuated the AR and MYC gene programs (Fig. 3i and Extended Data Fig. 7e,f), in addition to reducing the AR protein levels (Extended Data Fig. 7g). NSD1 inactivation also diminished hyperproliferative gene pathways (like E2F and G2M; Extended Data Fig. 7h) and had the strongest reduction in H3K36me2 levels upon a single-gene loss (Fig. 3j and Extended Data Fig. 7g), positioning NSD1 as the predominant H3K36 dimethyltransferase in PCa cells. NSD-catalyzed H3K36me2 mark was recently shown to sterically hinder loading of the H3K27 residue into catalytic pocket of the EZH2 enzyme[47], and NSD1 was reported to primarily antagonize the repressive PRC2/EZH2 complex[48]. Consistently, we found NSD1 loss to trigger a marked increase in EZH2/PRC2 activity, with several repressed target gene signatures being significantly downregulated upon NSD1 knockdown in VCaP cells (top panel, Fig. 3k). This was also confirmed using a PCa-specific PRC2 gene signature (bottom panel, Fig. 3k). Concordantly, treatment of siNSD1 cells with EPZ-6438 had substantially higher residual levels of the H3K27me3 mark relative to the control as well as NSD2-inactivated VCaP cells (Fig. 3l). In these experiments, loss of NSD2 alone had little to no effect on EZH2/PRC2 activity (Fig. 3j–l). These results position NSD1 as the primary writer of the H3K36me2 histone mark that counterbalances the EZH2/PRC2 repressive complex in PCa cells to maintain the hyper-transcriptional AR and MYC gene programs.

**Fig. 4 | LLC0150 is an NSD1/2 PROTAC with preferential cytotoxicity in AR-driven PCa. a**, Structure of LLC0150 and schema of NSD1 and NSD2 functional domains. LLC0150-binding PWWP1 domain is highlighted using a dashed red box. HMG: High mobility group; PHD: Plant homeodomain. **b**, Immunoblots of listed proteins in LNCaP cells treated with UNC6934 (warhead), LLC0150-dead (epimer control) or LLC0150 for 12 h at 1 μM. Total histone H3 is used as a loading control. **c**, Immunoblots of listed proteins in VCaP cells treated with LLC0150 (2uM) for increasing time durations. Total histone H3 is used as a loading control. **d**, GSEA plots of MYC target genes using the fold-change rank-ordered genes from LLC0150 vs DMSO treated LNCaP cells. DEGS, differentially expressed genes (*n* = 2 biological replicates; GSEA enrichment test). **e**, Venn diagram showing the overlap of AR ChIP-seq peaks in LNCaP cells treated with LLC0150 (2 μM for 48 h) or DMSO as control.

**f**, ChIP-seq read-density heatmaps of AR, FOXA1, and H3K27ac at enhancers that are co-bound by AR and FOXA1 in LNCaP cells plus/minus treatment with LLC0150 (2 μM for 48 h). **g**, Percent growth inhibition (Cell-titer Glo) of LNCaP cells upon co-treatment with varying concentrations of LLC0150 and enzalutamide. **h**, Dose-response curves of LLC0150 or enzalutamide in parental or enzalutamide-resistant VCaP cells. Data are presented as mean ± SEM (*n* = 2 biological replicates). Serving as a control, enzalutamide dose-response curve credentials the enzalutamide-resistant VCaP cell line. **i**, IC50 rank-order plot of over 110 human-derived normal or cancer cell lines after 5 days of treatment with LLC0150. AR+ PCa models are highlighted in red, and NSD2-mutant hematologic cell lines are shown in purple as well as marked with an asterisk (*). Each cell line's originating tissue lineages and known NSD2 alteration status are shown below.

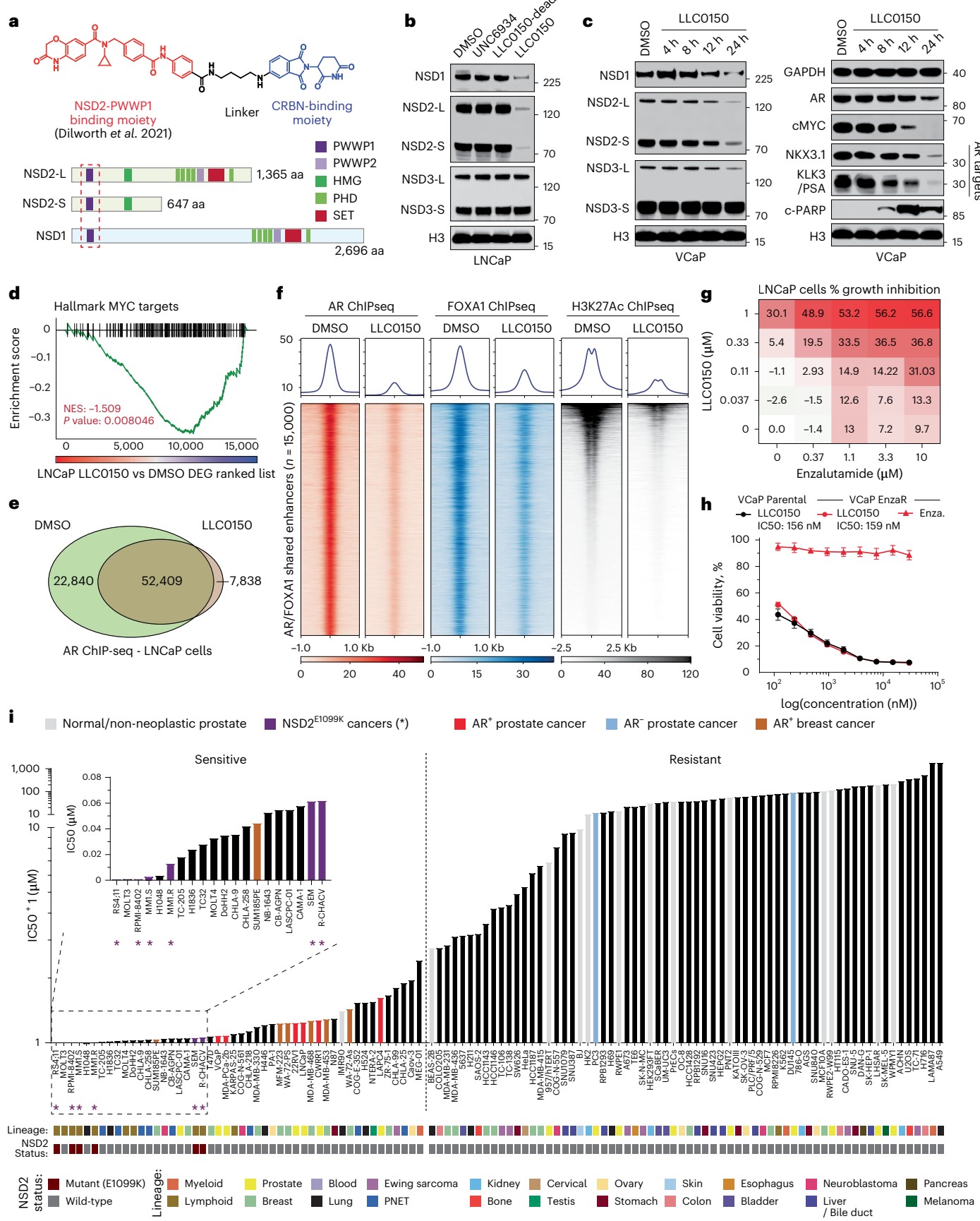

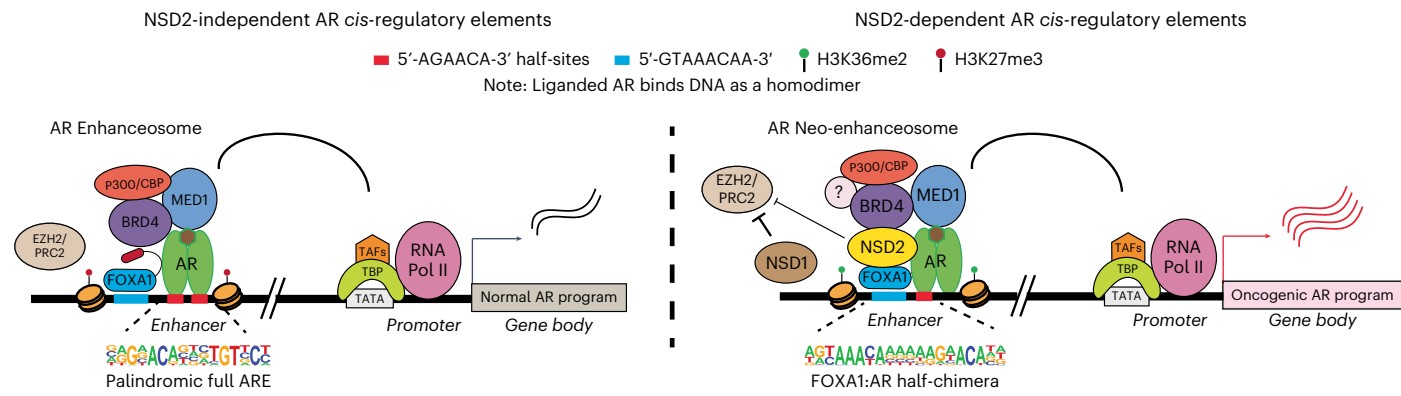

**Fig. 5 | Schema depicting NSD2's role in loading the AR enhanceosome at tumor-enriched chimeric AR neo-enhancer elements.** Chromatin loading of AR in prostate epithelial cells follows two distinct modes of DNA interactions: *Left:* NSD2-independent binding at *cis*-elements harboring the canonical, 15 bp palindromic AREs that are predominantly found in the physiological/normal enhancer circuitry, and *Right:* NSD2-dependent loading at *cis*-regulatory elements harboring chimeric AR half-motifs juxtaposed to the FOXA1 sequence that distinctively constitute the PCa-specific enhancer/super-enhancer (that is, AR neo-enhancer) circuitries. NSD1, partly supported by NSD2, counteracts repressive activity of the PRC2/EZH2 complex, thus further amplifying AR/MYC gene expression programs in mCRPC cells.

Interestingly, the loss of NSD2 led to a marked increase in NSD1 levels in PCa cells (Fig. 3j,m), likely suggesting that NSD1 could sustain residual oncogenic AR activity in these cells. Parallel inactivation of NSD1 and NSD2 in PCa cells resulted in the strongest decrease in H3K36me2 levels and AR target gene expression (Fig. 3j,m), triggering an accumulation of apoptotic marker cleaved-PARP (Extended Data Fig. 7i). Consistently, combined NSD1 and NSD2 inhibition resulted in significant cytotoxicity in AR-positive PCa cells (Fig. 3n), whereas inactivation of either genes alone had little to no tumor-killing effect in prostatic cell lines (Extended Data Fig. 7j). Altogether, these data suggest that NSD1 and NSD2, through distinct mechanisms, promote a hypertranscriptional chromatin state or enable oncogenic AR activity, respectively, in PCa cells.

### NSD1/2 dual PROTAC preferentially kills AR⁺ PCa
Following a medicinal chemistry campaign, we developed a proteolysis targeting chimera (PROTAC) compound, called LLC0150, which co-targets NSD1 and NSD2 (Fig. 4a and Supplementary Notes). LLC0150 links an NSD2 PWWP-domain binding warhead[49] to a cereblon E3-ligase-recruiting moiety pomalidomide. Treatment with LLC0150 triggers degradation of NSD1 and NSD2, while sparing NSD3 (Fig. 4b), in a proteasome and cereblon-dependent manner (Extended Data Fig. 8a). LLC0150 had no effect on other PWWP-domain-containing proteins, but showed partial neo-substrate activity (Extended Data Fig. 8b). This first-generation PROTAC also had poor solubility and pharmacokinetic properties for in vivo use. In line with our genetic data, in PCa cells, NSD1/2 co-degradation with LLC0150 triggered a decrease in the expression of AR and MYC, as well as their downstream gene targets (Fig. 4c,d). Acute loss of NSD1 and NSD2 in LLC0150-treated LNCaP cells resulted in impaired AR and FOXA1 chromatin binding (Fig. 4e,f and Extended Data Fig. 8c), with a parallel loss of H3K27ac activation mark at shared AR/FOXA1 enhancer sites (Fig. 4f). Treatment with LLC0150 also diminished the chromatin-bound AR fraction and the H3K36me2 histone mark (Extended Data Fig. 8d). In absence of NSD1/2, DHT-induced expression of AR target genes was significantly weakened (Extended Data Fig. 8e). LLC0150 treatment also markedly disrupted the assembly and activity of AR super-enhancers in LNCaP cells (Extended Data Fig. 8f)

Global transcriptomic analyses of LNCaP and VCaP cells treated with LLC0150 further showed a significant attenuation of proliferative pathways with a parallel induction of apoptotic signaling (Extended Data Fig. 9a). This was confirmed via massive accumulation of cleaved-PARP in the LLC0150-treated AR-positive PCa cell lines (Fig. 4c and Extended Data Fig. 9b). AR-positive PCa cell lines were considerably more sensitive to treatment with LLC0150 relative to the AR-negative disease models, immortalized normal, as well as primary prostate epithelial cells (Extended Data Fig. 9c). Inactive epimer control of LLC0150 (labeled as LLC0150-dead) did not affect the NSD1/2 levels or the viability of PCa cells (Fig. 4b and Extended Data Fig. 9d). Notably, LLC0150 showed marked synergy with enzalutamide—an AR-antagonistic drug—in killing LNCaP and VCaP cells (Fig. 4g and Extended Data Fig. 9e, f). More impressively, LLC0150 also retained cytotoxicity in cell line models that had acquired resistance to enzalutamide (Fig. 4h and Extended Data Fig. 9g). Similarly, several models of AR-positive mCRPC organoids that robustly express NSD2 (Extended Data Fig. 10a), showed significant attenuation of growth upon treatment with LLC0150 in a dose-dependent manner (Extended Data Fig. 10b–d).

Next, we characterized the cytotoxic effect of LLC0150 in a panel of over 110 human-derived normal and cancer cell lines originating from 22 different lineages (Supplementary Table 3). As expected, hematologic cancers harboring activating NSD2 mutations emerged as the most sensitive to treatment with LLC0150 (IC50 ranging from 0.274 - 69.68 nM), which was immediately followed by AR-positive PCa cell lines (shown in red, Fig. 4i). Notably, AR-positive disease models showed preferential cytotoxicity to NSD1/2 combined loss relative to AR-negative disease models as well as a host of normal cell lines. As proof of concept, we next performed direct intratumoral injection of LLC0150 in mice bearing VCaP xenograft tumors (Extended Data Fig. 10e). LLC0150 triggered marked degradation of NSD1/2 in tumor xenografts with a parallel loss in proliferative and gain of apoptotic markers (Extended Data Fig. 10f,g). Altogether, this data suggests that combined loss of NSD1 and NSD2 leads to a dramatic, almost complete, loss of the H3K36me2 histone mark and disruption of the AR/FOXA1 neo-enhancer circuitry, resulting in apoptotic PCa cell death. This positions NSD1/2 paralogs as a targetable digenic dependency in AR-driven, therapy-resistant PCa.

### Discussion
Most targeted therapies following surgical resection or radiation of primary PCa inhibit the androgen/AR signaling axis[4]. However, how

the prodifferentiation AR pathway in normal physiology gets reprogrammed to serve as the central oncogene in PCa remains largely unknown. Global AR chromatin-binding profiles are markedly different between normal and transformed prostate epithelia[5,6,13–15,50], and FOXA1 and HOXB13 have been implicated in driving AR's reprogramming upon transformation[5,14]. However, both FOXA1 and HOXB13 are also expressed in the normal epithelial cells, raising the possibility for additional cofactors to underlie the recruitment of AR to PCa-specific enhancer elements. Here, in a functional CRISPR screen, we identify NSD2 as a coactivator of the AR/FOXA1 enhanceosome. NSD2 is exclusively expressed in PCa cells, wherein it enables functional binding of AR at chimeric AR half-motifs, which majorly comprise the AR neo-enhancer circuitries. Consequently, NSD2 inactivation abolishes hallmark cancer phenotypes, whereas its re-expression in deficient cells restores neoplastic features. This positions NSD2 as a neo-coactivator of AR that assists transcription factors, like FOXA1, HOXB13, and ETS, in redistributing AR on the chromatin, thereby unlocking its oncogenic gene programs.

Intriguingly, in motif analyses of the PCa-specific AR cistromes, we also found significant depletion of the canonical ARE elements. Despite magnitude-folds increase in AR abundance in mCRPC, its loading at *cis*-regulatory elements comprising only palindromic AREs was significantly diminished. Also, the full ARE-containing sites were particularly inactivated in mCRPC tumors as evidenced by the loss of H3K27ac. This raises an intriguing possibility for the AR transcriptional activity stemming from a subset of canonical elements to rather impede tumor formation and/or progression, which is consistent with the physiological role of AR as a prodifferentiation factor. In fact, hyper-stimulation of AR activity has anti-proliferative effects in PCa cells[51], and bipolar androgen therapy involving cyclical inhibition and hyperactivation of AR is being currently tested in advanced patients[52,53]. These are exciting areas for further research.

We further found the loss of NSD2 in PCa cells to up-regulate NSD1, and co-inactivation of both NSD1/2 paralogs to be acutely cytotoxic. We uncovered that NSD1 and NSD2, through disparate mechanisms converge on wiring and maintaining the oncogenic AR gene program. Although NSD2 directly binds to AR and stabilizes the AR enhanceosome at de novo neo-enhancer elements, NSD1 functions as the primary writer enzyme for the H3K36me2 mark that antagonizes the PRC2/EZH2 repressive complex[32,33]. We envision the NSD2 function to evolve from enabling oncogenic AR activity in primary AR-dependent PCa to additionally supporting NSD1 in counter-balancing the canonical repressive PRC2 activity in the metastatic castration-resistant disease. Thus, the loss of NSD2 creates an increased dependency on NSD1 in AR-addicted PCa cells, positioning the NSD1/2 paralogs as targetable co-vulnerabilities in advanced disease. Here, we also characterized a dual PROTAC of NSD1 and NSD2 that confirmed co-degradation of these proteins to result in apoptotic cell death in AR-positive PCa. Notably both NSD1 and NSD2 are recurrently altered in hematological malignancies where they function as driver oncogenes (Supplementary Notes). Accordingly, we found LLC0150 to have the highest potency in NSD2-altered cancers. This finding highlights the potential application of this compound in studying and treating these tumors.

In summary, we identify and characterize NSD2 as an essential coactivator of the AR neo-enhanceosome that is exclusively expressed in PCa cells. NSD2 directly binds to AR and enables its loading at *cis*-regulatory elements harboring chimeric AR half-motifs, comprising over 65% of the malignant AR cistrome. We coalesce these mechanistic insights to propose that AR has two distinct modes of interacting with chromatin: 1) NSD2-independent binding at *cis*-elements harboring canonical full AREs that are predominantly found in the physiological enhancer circuitry, and 2) NSD2-dependent binding at *cis*-regulatory elements harboring chimeric AR half-motifs (like FOXA1:AR half) that distinctively constitute the cancer-specific enhancer circuitries of AR (Fig. 5). Furthermore, we uncover NSD1 and NSD2 as a digenic dependency in AR-positive PCa, and develop an NSD1/2 dual PROTAC degrader that shows preferential cytotoxicity in AR-positive PCa. Our findings warrant a focused development of new NSD-targeting therapeutics and evaluation of their efficacy and safety in preclinical and clinical studies.

## Online content

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

[1]Michigan Center for Translational Pathology, University of Michigan, Ann Arbor, MI, USA. [2]Department of Pathology, University of Michigan, Ann Arbor, MI, USA. [3]Rogel Cancer Center, University of Michigan, Ann Arbor, MI, USA. [4]Department of Urology, University of Michigan, Ann Arbor, MI, USA. [5]Molecular and Cellular Pathology Program, University of Michigan, Ann Arbor, MI, USA. [6]Department of Cancer Biology, Perelman School of Medicine, University of Pennsylvania, Philadelphia, PA, USA. [7]Cancer Biology Program, University of Michigan, Ann Arbor, MI, USA. [8]State Key Laboratory of Chemical Biology, Shanghai Institute of Organic Chemistry, Chinese Academy of Sciences, Shanghai, China. [9]Howard Hughes Medical Institute, University of Michigan, Ann Arbor, MI, USA. [10]Bioscience, Research and Early Development, Oncology R&D, AstraZeneca, Waltham, MA, USA. [11]Abramson Family Cancer Research Institute, Perelman School of Medicine, University of Pennsylvania, Philadelphia, PA, USA. [12]Epigenetics Institute, Perelman School of Medicine, University of Pennsylvania, Philadelphia, PA, USA. [13]These authors contributed equally: Sanjana Eyunni, Brijesh Kumar Verma. ✉e-mail: aparolia@umich.edu; arul@umich.edu; asangani@upenn.edu

## Methods

### Ethical statement

All experiments detailed in this paper were performed in compliance with the Institutional Review Board and the Institutional Animal Care and Use Committee at the University of Pennsylvania and the University of Michigan.

**Animal procurement.** Animal studies were approved by the Institutional Animal Care and Use Committee at the University of Pennsylvania and/or the University of Michigan. Animal use and care were in strict compliance with institutional guidelines, and all experiments conformed to the relevant regulatory standards by the universities. NOD SCID or NCI SCID/NCr athymic nude mice were obtained from the Jackson Laboratory (strain code: 005557) and Charles River (strain code: 561). All in vivo experiments were initiated with male mice aged 5–8 weeks. All mice were housed in a pathogen-free animal barrier facility and all in vivo experiments were initiated with male mice aged 5-8 weeks. All mice were maintained under the conditions of pathogen-free, 12 h light/12 h dark cycle, temperatures of 18–23 °C, and 40–60% humidity.

**Statement on use of human specimens.** Prostate tumor patient tissues were acquired from the University of Michigan pathology archives. These tissues were utilized for Immunohistochemistry and multiplex Immunofluorescence experiments to assess for Cytokeratin-8 and NSD2 expression in tumor or adjacent normal prostate cells. Formalin-fixed paraffin-embedded specimens from the archives were used upon approval by the University of Michigan Institutional Review Board and does not require patient consent.

### Cell lines

Most cell lines were purchased from the American Type Culture Collection (ATCC) and were cultured following ATCC protocols. For all experiments, LNCaP and 22RV1 cells were grown in RPMI 1640 medium (Gibco) and VCaP cells in DMEM with Glutamax (Gibco) medium supplemented with 10% fetal bovine serum (FBS; Invitrogen). HEK293FT cells were grown in DMEM (Gibco) medium with 10% FBS. All cells were grown in a humidified 5% $CO_2$ incubator at 37 °C. Mycoplasma and cell line genotyping were performed once a fortnight and every month respectively at the University of Michigan Sequencing Core using Profiler Plus (Applied Biosystems). Results from these were compared with corresponding short tandem repeat profiles in the ATCC database to authenticate their identity.

### Antibodies

For immunoblotting, the following antibodies were used: NSD1 (NeuroMab: 75-280, 1:1000); NSD2 (Abcam:ab75359, 1:1,000); NSD3 (Cell Signaling Technologies: 92056 S, 1:1,000); KLK3/PSA (Dako:A0562, 1:1,000); FKBP5(Cell Signaling Technologies: 12210, 1:1,000); NKX3-1 (Cell Signaling Technologies:83700 S, 1:1,000); FOXA1 N-terminal (Cell Signaling Technologies: 58613 S; Sigma-Aldrich: SAB2100835, 1:1,000); FOXA1 C-terminal (ThermoFisher Scientific: PA5-27157, 1:1,000); AR (Millipore: 06-680, 1:1,000); AR (Abcam: ab133273, ab108341, 1:1,000); H3 (Cell Signaling Technologies: 3638 S, 1:2,500); GAPDH (Cell Signaling Technologies: 3683, 1:2,500); H3K27me3(Millipore: 07-449, 1:2500); H3K36me2 (Cell Signaling Technologies: 2901 S, Abcam: ab9049, 1:2500); H3K27ac (Active Motif, catalog no39336, catalog no39133, 1:2500); Phospho-AR (Ser-81) (Millipore, catalog no 07-1375-EMD, 1:1,000); HALO (ThermoFisher Scientific, catalog no G9281, 1:1,000); HA (Cell Signaling Technologies, catalog no 3724 S, 1:1000); His (Cell Signaling Technologies, catalog no2365 S, 1:1,000). ChIP-seq assays were performed using the following antibodies: FOXA1 (ThermoFisher Scientific: PA5-27157); AR (Millipore: 06-680); H3K4me1 (Abcam: ab8895); H3K4me2 (CST: C64G9); H3K36me2 (Abcam: ab9049); H3K27me3 (EMD: 07-449), and H3K27ac (Active Motif, catalog no39336).

### Cell-free protein-protein interaction studies

In vitro protein expression was carried out by cloning the desired expression cassettes downstream of a Halo- or His-tag to produce fusion proteins. Briefly, AR-DBD was subcloned in pFN21K containing Halo-tag, and NSD2-HMGa was cloned in pcDNA4c containing His-tag. After cloning, the fusion proteins were expressed using the cell-free transcription and translation system (catalog no L4140, Promega) following the manufacturer's protocol. For each reaction, protein expression was confirmed by Western blot.

A total of 10 μl cell-free reaction containing halo- and His-tag fusion proteins was incubated in PBST (0.1% tween) at 4 °C overnight. Ten microliter HaloLink beads (catalog noG931, Promega) were blocked in BSA at 4 °C for overnight. After washes with PBS, the beads were mixed with AR-NSD2-HMGa and TM mixture and incubated at room temperature for 1 h. Halolink beads were then washed with PBST for four times and eluted in SDS loading buffer. Proteins were separated on SDS gel and blotted with anti-His Ab (CST: catalog no2365 S).

### Colony formation assays

For the colony formation assay, approximately 10,000 cells/well in six-well plates (n = 3) were seeded and treated with the required drugs/compounds or vehicle for 12–14 days. Media was replenished every 3-4 days. Colonies were fixed and stained using 0.5% (w/v) crystal violet (Sigma, C0775) in 20% (v/v) methanol for 30 min, washed with distilled deionized water, and air-dried. After scanning the plate, the stained wells were destained with 500 μl 10% acetic acid, and the absorbance was determined at 590 nm using a spectrophotometer (Synergy HT, BioTek Instruments).

### Cellular protein fractionation assays

Chromatin-bound proteins were extracted following a protocol previously described[17]. In brief, 10 million cells were collected, washed with DPBS, and resuspended in 250 μl Buffer A (10 mM HEPES pH 7.9, 10 mM KCl, 1.5 mM $MgCl_2$, 0.34 M sucrose, 10% glycerol and 1 mM DTT) supplemented with 0.1% TritonX-100. After incubation on ice for 10 min, the nuclear pellet was collected by centrifugation at 1,300g for 5 min at 4 °C, washed in Buffer A, and resuspended in Buffer B (3 mM EDTA, 0.2 mM EGTA and 1 mM DTT) with the same centrifugation settings, and incubated on ice for 30 min. The chromatin pellet was collected by centrifugation at 1,700g for 5 min at 4 °C, washed and resuspended in Buffer B with 150 mM NaCl, and incubated on ice for 20 min. After centrifugation at 1,700g for 5 min to remove proteins soluble in 150 mM salt concentrations, the pellet was then incubated in Buffer B with 300 mM NaCl on ice for 20 min and centrifuged again at 1,700g to obtain the final chromatin pellet. The chromatin pellet was dissolved in a sample buffer, sonicated for 15 s, and boiled at 95 °C for 10 min. Immunoblot analysis was conducted on samples as described above. All buffers were supplemented with Pierce protease inhibitor and Halt protease & phosphatase inhibitors.

### RNA isolation and quantitative real-time PCR

Standard protocol from the miRNeasy Mini kit (Qiagen) was used to extract total RNA with the inclusion of on-column genomic DNA digestion step using the RNase-free DNase Kit (Qiagen). RNA concentration was estimated using the NanoDrop 2000 spectrophotometer (ThermoFisher Scientific), and 1 g total RNA was used for complementary DNA (cDNA) synthesis using the SuperScript III Reverse Transcriptase enzyme (ThermoFisher Scientific) following manufacturer's instructions. 20 ng cDNA was used for each polymerase chain reaction (PCR) using the FAST SYBR Green Universal Master Mix (ThermoFisher Scientific), and every sample was quantified in triplicates. Gene expression was normalized and calculated relative to *GAPDH* and *HPRT1* (loading control) using the delta-delta Ct method and normalized to the control group for graphing. Quantitative PCR (qPCR) primers were designed using the Primer3Plus tool (http://www.bioinformatics.nl/cgi-bin/primer3plus/primer3plus.cgi) and synthesized by Integrated DNA

Technologies. Primers used in this study are provided in Supplementary Table 5.

## siRNA/ASO-mediated gene knockdown

Mammalian cells were seeded in a 6-well plate format at the density ranging from 100,000–250,000 cells per well. 12 h post seeding, cells were transfected with 25 nM of gene-targeting ON-TARGETplus SMARTpool siRNAs (or ASOs) or non-targeting pool siRNAs (or ASOs) as negative control (Dharmacon) using the RNAiMAX reagent (Life Technologies; catalog no: 13778075) on two consecutive days, following manufacturer's instructions. 72 h after transfection, total RNA and protein were extracted to confirm efficient (>80%) knockdown of the target genes. For the siRNA-treated VCaP DMSO/EPZ-6438 RNA-seq experiment (Fig. 3k), cells were pre-treated with control siRNA (siNC) or siRNA targeting NSD1, NSD2, or NSD1/2 (siNSD1, siNSD2) for 30 days, followed by 72 h of EPZ-6438 treatment. Catalog numbers and guide RNA sequences of siRNA SMARTpools (Dharmacon) are provided in Supplementary Table 5.

## CRISPR-Cas9-mediated gene knockout

For gene knockouts, cells were seeded in a 6-well plate at a density of 200,000–300,000 cells per well and transduced with viral particles with lentiCRISPR-V2 plasmids coding either non-targeting (sgNC) or sgRNAs targeting NSD1 and NSD2. This was followed by 3 days of puromycin selection, after which proliferation assays were carried out as described below. The lentiCRISPR-V2 vector was a gift from Dr. Feng Zhang's lab (Addgene plasmid #52961). sgRNA sequences are provided in Supplementary Table 5.

## Proliferation assays

For siRNA growth assays, cells were directly plated in a 96-well plate at the density of 2,500–8,000 cells per well and transfected with gene-specific or non-targeting siRNAs, as described above, on day 0 and day 1. Every treatment was carried out in six independent replicate wells. CellTiter-Glo reagent (Promega) was used to assess cell viability at multiple time points after transfection, following the manufacturer's protocol. Data were normalized to readings from siNC treatment on day 1 and plotted as relative cell viability to generate growth curves. Alternatively, for CRISPR sgRNA growth assays, cells were treated as described above for target gene inactivation and seeded into a 96-well plate at 2500 cells per well, with five-six replicates per group.

## Matrigel invasion assay

LNCaP CRISPR clones were grown in 10% CSS-supplemented medium for 48 h for androgen starvation. Matrigel-coated invasion chambers were used and additionally coated with polyethylene terephthalate membrane to allow for fluorescent quantification of the invaded cells (Biocoat: 24-well format, no. 354166). On the upper layer of the chamber, fifty thousand starved cells were resuspended in serum-free medium and were added to each invasion chamber while 20% FBS-supplemented medium was added to the bottom wells to serve as a chemoattractant. 12 h later, medium from the bottom well was aspirated and replaced with 1x HBSS (Gibco) containing 2 µg/ml Calcein-green AM dye (ThermoFisher Scientific; C3100MP) and incubated for 30 min at 37 °C. Invasion chambers were then placed in a fluorescent plate reader (Tecan-Infinite M1000 PRO), and fluorescent signals from the invaded cells at the bottom were averaged across 16 distinct regions per chamber to determine the extent of invasion. For rescue experiments, stable lines overexpressing the NSD2 isoforms were generated. Briefly, to LNCaP NSD2-KO lines, GFP or NSD2-Long isoform containing viruses were added. These lines were then used to perform the invasion assay as described above.

## RNA-seq and analysis

RiboErase RNA-seq libraries were prepared using 200–1,000 ng total RNA. Ribosomal RNA was removed by enzymatic digestion of the specific probe-bound duplex rRNA (KAPA RNA Hyper+RiboErase HMR, Roche) and then fragmented to around 200-300 bp with heat in the fragmentation buffer. Following this, double-stranded cDNA was generated, and end-repair and ligation was performed using New England Biolabs (NEB) adapters. Final library preparation was performed by amplification with the 2x KAPA HiFi HotStart mix and NEB dual barcode following the manufacturer's protocol. Library quality was measured on an Agilent 2100 Bioanalyzer for product size and concentration. Paired-end libraries were sequenced with the Illumina HiSeq 2500, (2 × 100 nucleotide read length) with sequence coverage to 15–20 M paired reads.

RNA data was first processed using kallisto (version 0.46.1)[54]. Then analysis was performed in R, first read counts were normalized and filtered (counts >10) using EdgeR[55] (edgeR_3.39.6), and differential expression was performed using Limma-Voom (limma_3.53.10)[56]. GSEA was performed using fgsea (fgsea_1.24.0)[57] and comparisons were made to several signatures, including an experimentally derived AR signature, the human hallmark MsigDB signatures (/www.gsea-msigdb.org), and the hallmark androgen response signature (HALLMARK_ANDROGEN_RESPONSE.v7.5.1.gmt). In addition, R packages tidyverse, gtable, gplots, ggplot2 and EnhancedVolcano (EnhancedVolcano_1.15.0) were also used for generating summary figures (R version 4.2.1 (refs. 58–60)).

## ChIP-seq and data analysis

ChIP experiments were carried out using the Ideal ChIP-seq Kit for Transcription Factors or Histones (Diagenode) as per the manufacturer's protocol. Chromatin from $2 × 10^6$ cells (for transcription factors) and $1×10^6$ cells (for histones) was used for each ChIP reaction with 4 or 2 µg of the target protein antibody, respectively. In brief, cells were trypsinized and washed twice with 1× PBS, followed by crosslinking for 8 min in 1% formaldehyde solution. Crosslinking was terminated by the addition of 1/10 volume 1.25 M glycine for 5 min at room temperature followed by cell lysis and sonication (Bioruptor, Diagenode), resulting in an average chromatin fragment size of 200 bp. Fragmented chromatin was then used for immunoprecipitation using various antibodies, with overnight incubation at 4 °C. ChIP DNA was de-crosslinked and purified using the iPure Kit V2 (Diagenode) using the standard protocol. Purified DNA was then prepared for sequencing as per the manufacturer's instructions (Illumina). ChIP samples (1–10 ng) were converted to blunt-ended fragments using T4 DNA polymerase, *Escherichia coli* DNA polymerase I large fragment (Klenow polymerase), and T4 polynucleotide kinase (New England BioLabs (NEB)). A single adenine base was added to fragment ends by Klenow fragment (3′ to 5′ exo minus; NEB), followed by ligation of Illumina adaptors (Quick ligase, NEB). The adaptor-ligated DNA fragments were enriched by PCR using the Illumina Barcode primers and Phusion DNA polymerase (NEB). PCR products were size selected using 3% NuSieve agarose gels (Lonza) followed by gel extraction using QIAEX II reagents (Qiagen). Libraries were quantified and quality checked using the Bioanalyzer 2100 (Agilent) and sequenced on the Illumina HiSeq 2500 Sequencer (125-nt read length).

ChIP-seq analysis was carried out by first assessing reads and performing trimming using Trimmomatic version 0.39 (settings TruSeq3-PE-2.fa:2:30:10, minlen 50)[61]. Paired-end reads were aligned to hg38 (GRCh38) human genome reference using bwa ("bwa mem" command with options −5SP -T0, version 0.7.17-r1198-dirty)[62]. Alignments were then filtered using both samtools[63] (v1.1, quality score cutoff of 20) and picard[64] MarkDuplicates (v(2.26.0-1-gbaf4d27-SNAPSHOT), removed duplicates). Peak calling was performed using MACS2 (v2.2.7.1)[65] using narrowpeak setting for narrow peaks and a second set for broad peaks (for example, H3K27ac,−broad -B−cutoff-analysis−broad-cutoff 0.05−max-gap 500). Finally, bedtools (v2.27.1)[66] was used to remove blacklisted regions of the genome from the peak list (Encode's exclusion list ENCFF356LFX.bed). UCSC's tool wigtoBigwig (v2.8) was used for conversion to bigwig formats[67].

## Overlap analysis of ChIP-seq data

Peak lists from MACS were compared between samples using R package ChIPpeakAnno[68–70]. Peaks within 500 bp of each other were reduced to single peaks. Overlaps were calculated using settings maxgap = −1L, minoverlap=0 L, ignore.strand=TRUE, connectedPeaks=c('keepAll', 'min', 'merge'). Comparisons of enrichment sites to the known gene database (TxDb.Hsapiens.UCSC.hg38.knownGene) were performed using R package ChIPseeker. A distance of ±1 kb was used to assess relative distance from gene regions.

## HOMER motif calling

*De novo* and known motif enrichment analysis was performed using HOMER (version v.4.10)[42,71]. Custom motif matrices were generated manually, then assigned score thresholds using HOMER's utility seq2profile, allowing for two mismatches. This setting was chosen after iteratively comparing performance with the pre-existing FOXA1:AR motif. Further customization was achieved by checking for presence of motif elements with different spacings, ranging from 0-8 'N's added between elements, and flipping the order of elements in each of these: FOXA1-ARE, ARE-FOXA1, FOXA-N-ARE, ARE-N-FOXA1, FOXA1-NN-ARE, ARE-NN-FOXA1, etc.

Custom motifs were then further validated using XSTREME (v5.5.5)[72] from the MEME Suite[71] to check for additional configurations and variations in padding between motif elements.

## Enrichment heatmaps

The software Deeptools (v3.5.1) was used to generate enrichment plots and read-density heatmaps. A reference point parameter of ±2.5 kb for histone signals and ±1.5 kb for AR/FOX signals was used. Other settings included using 'skipzeros', 'averagetype mean,' and 'plotype se'. The Encode blacklist ENCFF356LFX was used[73].

## Motif and signal plots

Sushi (Sushi_1.32.0) package in R was used to layer signal tracks. The plotBedgraph(), plotGenes(), plotBed() functions were used with output from ChIP-seq alignments and output from HOMER motif enrichment analysis[74].

## Superenhancer analysis

Super-enhancer regions were identified with findPeaks function from HOMER (version v.4.10)[42] using options "-style super -o auto". In addition, the option "-superSlope −1000" was added to include all potential peaks, which were used to generate the super-enhancer plot (super-enhancer score versus ranked peaks). The slope value of greater than or equal to 1 was used to identify super-enhancer clusters. The input files to findPeaks were tag directories generated from alignment files in SAM format with makeTagDirectory function from HOMER. Super-enhancer scores were plotted using the normalized tag count values between the datasets.

## Single-cell data analysis

Three public scRNA-seq datasets from primary PCa were downloaded from GEO or a website provided by the author (GSE193337, GSE185344, www.prostatecellatlas.org)[75]. Using cell annotation from the Tuong et al. dataset as reference, luminal cells were annotated for the other two datasets with the label transfer method of Seurat[29]. Pseudo-bulk expression profiles[30] were generated by summing counts from all cells annotated as luminal cells for each patient (tumor and normal samples separately). Normalization was achieved by computing normalization factors with the trimmed mean of M-values method[31] and applying the cpm function from edgeR (v3.36.0)[32]. Box plots of *NSD2* and *PCA3* expression were generated with ggpubr[33] and paired Wilcoxon test was used to test the significance of the difference between benign and tumor (only patients with paired benign and tumor samples were included).

## IHC and immunofluorescence

IHC was performed on 4-μm-thick formalin-fixed, paraffin-embedded tissue sections using anti-NSD2 mouse monoclonal primary antibody (catalog no. ab75359, Abcam), anti-AR rabbit monoclonal primary antibody (catalog no. 760-4605, Roche-Ventana), and anti-CK-8 rabbit monoclonal primary antibody (catalog no. ab53280, Abcam). Singleplex IHC was carried out on the Ventana ULTRA automated slide staining system (Roche-Ventana Medical Systems) using the OmniView Universal diaminobenzidine detection kit (catalog no. 760-500, Roche-Ventana) and hematoxylin II (catalog no. 790-2208, Roche-Ventana) for counterstain. Staining was evaluated under 100× and 200× magnification using a brightfield microscope.

## Assessment of drug synergism

To determine the synergy between two drug treatments, cells were treated with increasing concentrations of either drug for 120 h, followed by the determination of viable cells using the CellTiter-Glo Luminescent Cell Viability Assay (Promega). The experiment was carried out in four biological replicates. The data were expressed as percentage inhibition relative to baseline, and the presence of synergy was determined by the Bliss method using the synergy finder R package.

## Statistics and reproducibility

All immunoblot experiments were repeated at least two to three times. For immunofluorescent staining experiments, number of biological replicates used in each case are noted in the figure legend. While representative images are shown in some panels, for example Figure 3b and Extended Data Fig. 5e, quantitation from all independent replicates is included. No statistical methods were used to predetermine sample sizes for any experiments. For all analyses, data distribution was assumed to be normal, but this was not formally tested. All immunofluorescence data quantification was performed in a double-blinded manner by in-house pathologists. For in vivo animal experiments.

## Reporting summary

Further information on research design is available in the Nature Portfolio Reporting Summary linked to this article.

## Data availability

All data are available in the manuscript or the supplementary information. Raw next-generation sequencing data, including ChIP-seq and RNA-seq, generated in this study are deposited in the Gene Expression Omnibus (GEO) repository (accession number: GSE242737) at National Center for Biotechnology Information. ChIP-seq data from normal, primary PCa and mCRPC were pulled from GEO repositories GSE130408 and GSE70079. Three public scRNA-seq datasets from primary PCa were downloaded from GEO (GSE193337 and GSE185344) or a web portal provided by the authors. Source data are provided with this paper.

## Code availability

All custom codes used for data analyses are freely available from the following public repositories: Github: https://github.com/mctp/NSD2_req_subunit (ref. 76) and Zenodo: https://doi.org/10.5281/zenodo.12979564 (ref. 77).

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

## Acknowledgements

We thank R. Wang from the sequencing team at the Michigan Center for Translational Pathology (MCTP) for generating NGS libraries and coordinating sequencing. We thank J. Waninger from MCTP for her technical help with size exclusion chromatography. We thank R. Rebernick for help with patient gene correlation analyses. We also acknowledge critical support from the University of Michigan Biomedical Research Core Facilities, especially the Vector, Flow Cytometry, and Proteomics and Peptide Synthesis Cores. We thank C. Vakoc from Cold Spring Harbor Laboratories for generously sharing the domain-focused epigenetics sgRNA library. We thank A. Heller, O. Rivera, A. Pawar and R. Natesan from the University of Pennsylvania for providing technical and bioinformatics assistance. This research was supported by the following mechanisms: Prostate Cancer Foundation (PCF), Prostate Specialized Programs of Research Excellence (SPORE) Grant P50-CA186786, National Cancer Institute Outstanding Investigator Award R35-CA231996, National Cancer Institute P30-CA046592 (Rogel Cancer Center investigators, including A.P. and A.M.C.), Early Detection Research Network Grant U2C CA271854 (A.M.C.), and National Cancer Institute R00-CA187664. A.P. is supported by the NIH/NCI K00 fellowship (K00-CA245825), Michigan SPORE Career Enhancement Program, PCF Young Investigator Award, and Rogel Fellowship. L.X. is supported by a Department of Defense Prostate Cancer Research Program Idea Development Award (W81XWH-21-1-0500) and a PCF Young Investigator Award. A.M.C. is a Howard Hughes Medical Institute Investigator, A. Alfred Taubman Scholar, and American Cancer Society Professor. I.A.A. is supported by grants from the National Institute of Health and Department of Defense (R01-CA249210-0 and W81XWH-17-0404).

## Author contributions

A.P., A.M.C. and I.A.A. conceived the study and the experiments; A.P. and S.E. designed and carried out the multiomics and functional experiments with assistance from J.L., S.E.C., Y.L., L.X., X.W., T.H., Y.Q., P.G., M.J., R.R., S.M., M.P., E.M., J.C.T., M.L., F.S. and X.C.; B.K.V. designed and carried out the fragment-based coimmunoprecipitation and functional experiments with assistance from S.A. and R.R.; B.K.V. and C.K.D. conducted in vivo experiments. A.P. and E.Y. carried out all the bioinformatics analyses with assistance from J.G., S.V. and M.A.; Y.Z. analyzed the single-cell RNA-seq datasets; O.T. provided guidance; L.L., C.H., Z.W. and K.D. were involved in the synthesis and structural validation of the LLC0150 compound; A.P. wrote the manuscript and developed the figures with feedback from A.M.C. and I.A.A.; S.E. wrote the Methods section, with help from B.K.V., E.Y., Z.W., R.M. and Y.Z. Next-generation sequencing related to data shown in Figs. 1c, 3k,m and 4d as well as related supplementary figures was performed at the University of Michigan, whereas next-generation sequencing related to data shown in Figs. 1g,i, 2a–e,i,j and 4e,f as well as related supplementary figures was carried out at the University of Pennsylvania. All authors discussed the results, provided feedback and reviewed the final manuscript.

## Competing interests

All the authors declare no competing financial interests.

## Additional information

**Extended data** is available for this paper at https://doi.org/10.1038/s41588-024-01893-6.

**Correspondence and requests for materials** should be addressed to Abhijit Parolia, Arul M. Chinnaiyan or Irfan A. Asangani.

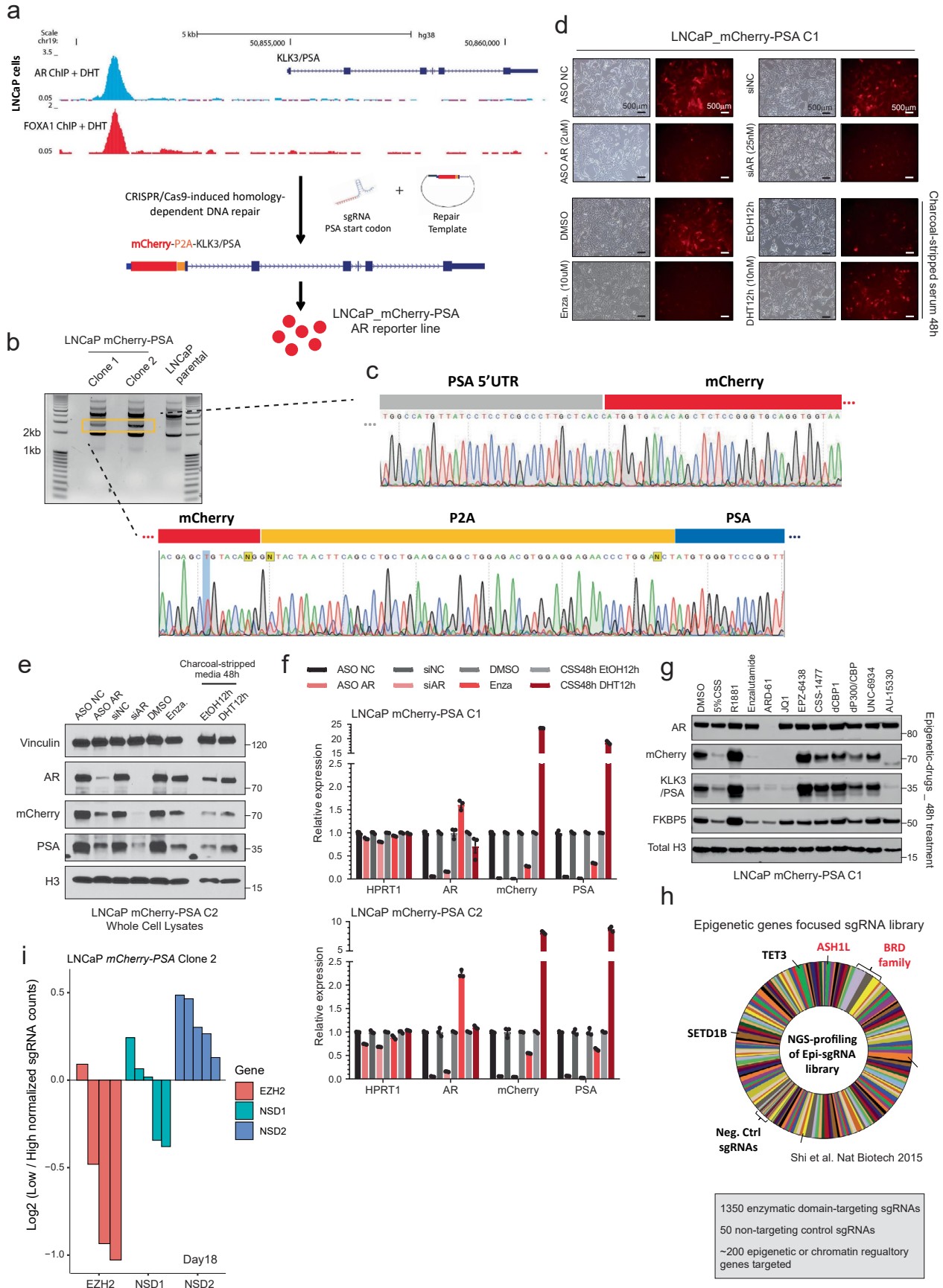

**Extended Data Fig. 1 | See next page for caption.**

**Extended Data Fig. 1 | Generation and characterization of the endogenous mCherry-PSA AR reporter cell lines. a**) Schematic representation of the workflow of LNCaP-mCherry-PSA AR reporter cell line generation. **b**) DNA gel electrophoresis image showing the exogenously inserted mCherry amplicon in the LNCaP-mCherry-PSA lines. Clones 1 and 2 were used for the functional CRISPR screen. **c**) Sanger sequencing chromatograms of the PCR amplicon from reporter cells in panel (**b**) showing the KLK3/PSA gene promoter and exon 1 start codon junctions. **d**) Representative brightfield and mCherry immunofluorescence images of the LNCaP-mCherry-PSA clone 1 treated with (top) AR-targeting siRNA or antisense oligonucleotides (ASOs) (siAR and ASO AR respectively) or enzalutamide (bottom left). Reporter cells were also serum starved for 48 h and stimulated with DHT (10 nM for 12 h) to showcase gain in signal (bottom right).

All treatments were repeated at least twice. Scale bar: 500 μm. **e**) Immunoblots of noted proteins in LNCaP reporter cells as in panel (d). **f**) Expression (qPCR) of noted genes in reporter monoclones treated as in panel (d) to manipulate AR signaling (n = 3 biological replicates). Mean +/- SEM is shown. **g**) Immunoblots of noted proteins, including the exogenously introduced mCherry protein, in LNCaP reporter cells treated with AR-targeting epigenetic drugs. Total H3 is used as a loading control. **h**) Next-generation sequencing-based abundance of sgRNAs in the epigenetic-focused library used in the CRISPR screen highlighting some of the known epigenetic regulators of AR. **i**) Individual NSD1, NSD2, or EZH2-targeting sgRNA ratios in mCherry-LOW to mCherry-HIGH cells in the CRISPR screen.

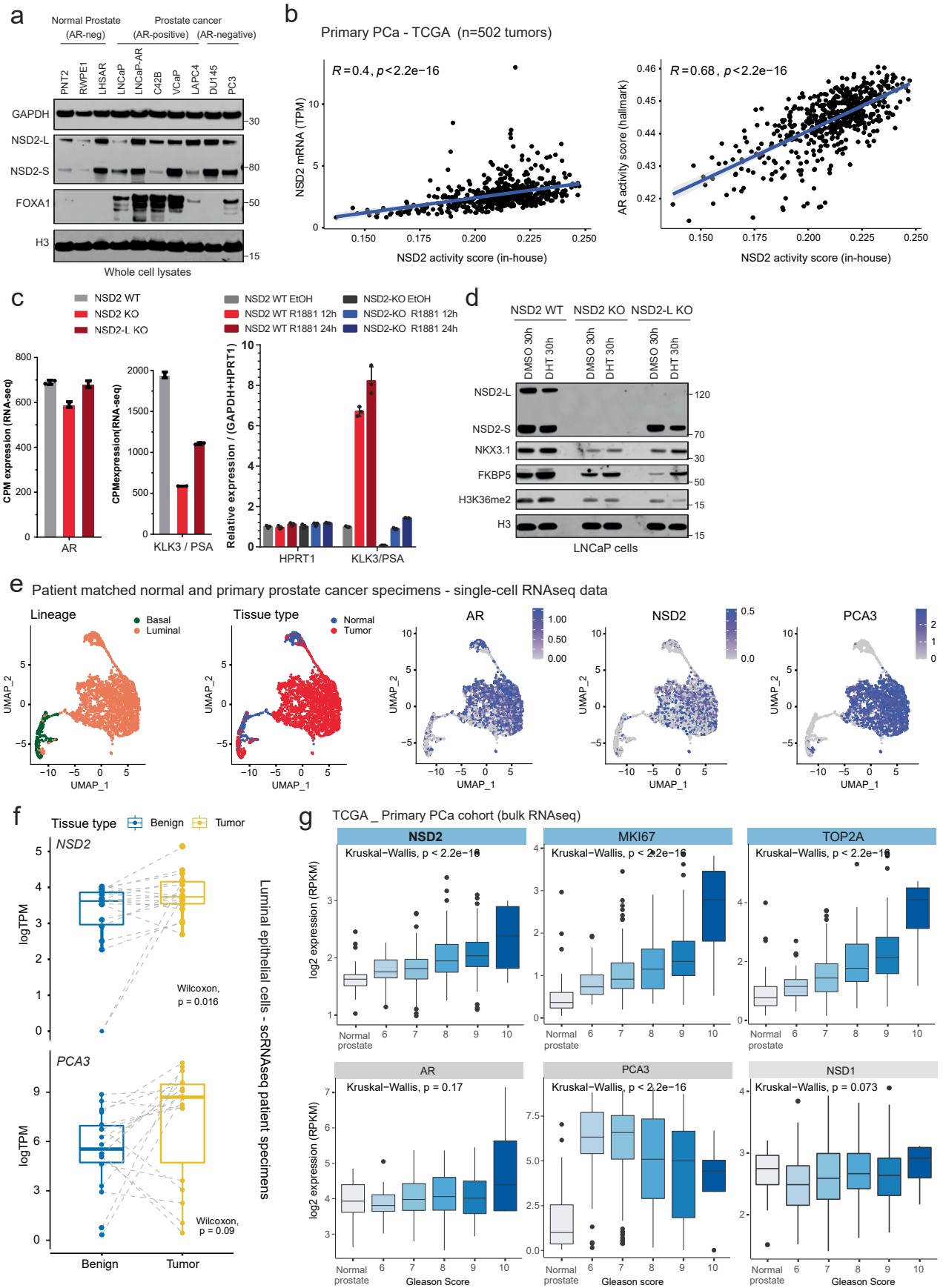

**Extended Data Fig. 2 | See next page for caption.**

**Extended Data Fig. 2 | NSD2 transcript and protein expression in primary patient specimens. a)** Immunoblot of labeled proteins in a collection of AR-positive and AR-negative prostate cell lines. GAPDH and H3 are used as a loading control. **b)** *Left:* Correlation plots showing the *NSD2* transcript expression and gene signature-based "NSD2 activity score" in primary prostate cancers from the TCGA cohort (n = 502 tumors). ***Right:*** Correlation plots showing NSD2 activity score and the widely-used hallmark AR activity score in primary PCa tumors. (Pearson's linear correlation coefficient, permutation test). Line, mean; shaded region, SEM. **c)** Relative expression (qPCR or RNA-seq) of *AR* and *KLK3* transcripts in CRISPR-edited NSD2-KO or NSD2-L-KO LNCaP cells (***left;*** n = 2 biological replicates) or NSD2 CRISPR-edited cells stimulated with R1881 for 12 or 24 h (***right;*** n = 3 biological replicates). *HPRT1* is used as a loading control. Mean +/-

SEM is shown. **d)** Immunoblot of labeled proteins in LNCaP NSD2 WT and KO cells stimulated with DHT for 30 h. **e)** UMAP plots from patient-matched normal and primary prostate cancer single-cell RNA-seq data. **f)** *NSD2* and *PCA3* transcript expression in patient-matched normal and primary prostate cancer luminal epithelial cells (pseudo-bulk analyses from single-cell data; n = 15 biological replicates, two-sided Wilcoxon test). Box plot center, median; box, quartiles 1-3; whiskers, quartiles 1-3 ± 1.5 × interquartile range; dot are outliers. **g)** Box plot showing RNA expression of labeled genes in primary prostate cancer specimens (TCGA cohort) stratified by the Gleason score (normal = 52; Gleason 6 = 46; Gleason 7 = 249, Gleason 8 = 65; Gleason 9 = 138, Gleason 10 = 4 tumor specimens. One-way ANOVA and Kruskal-Wallis test). Box plot center, median; box, quartiles 1-3; whiskers, quartiles 1-3 ± 1.5 × interquartile range; dot, outliers.

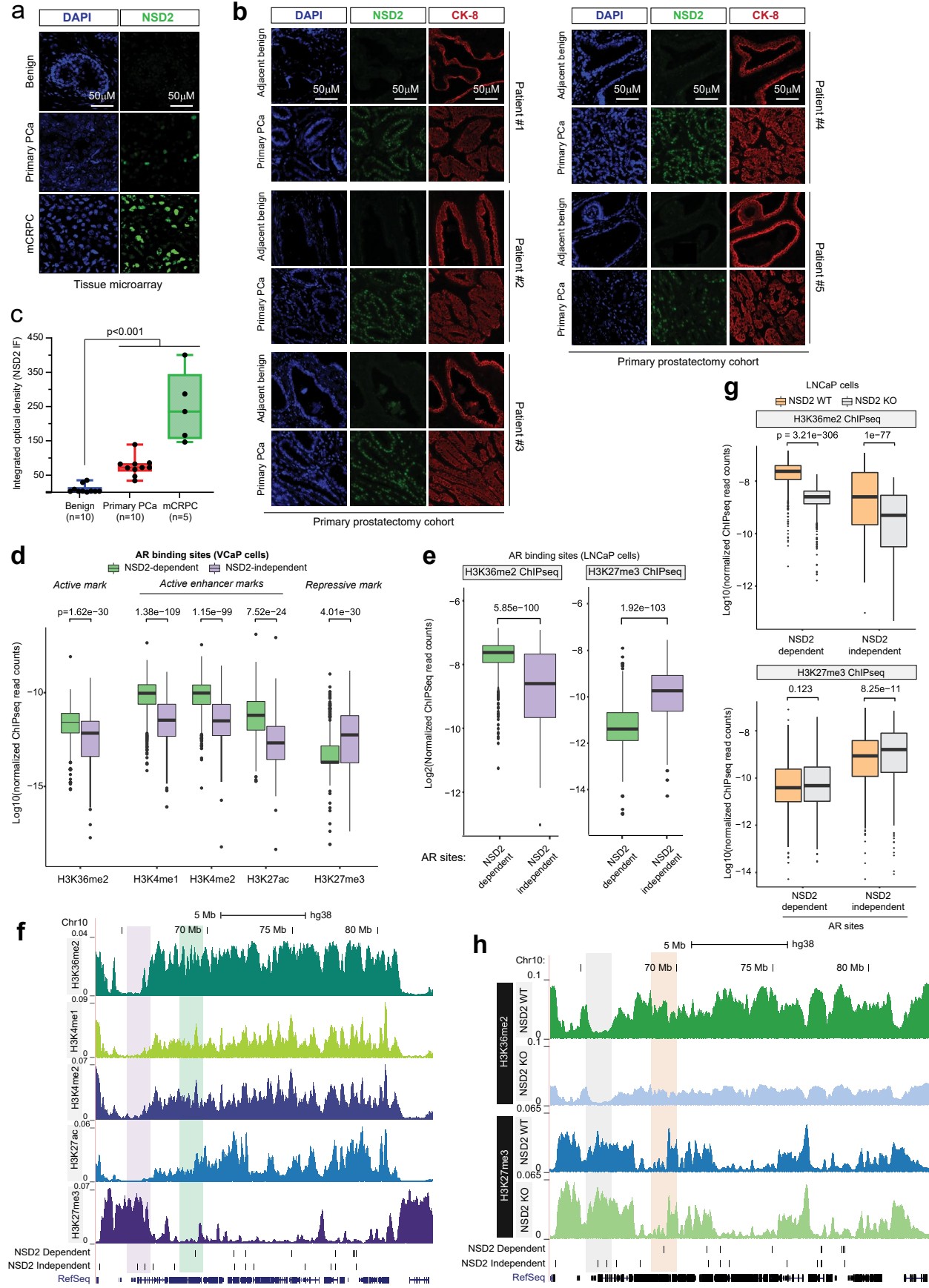

**Extended Data Fig. 3 | See next page for caption.**

**Extended Data Fig. 3 | NSD2 and H3K36me2 expression in patient tumors and prostate cancer cell lines. a**) Representative immunofluorescence (IF) images of NSD2 in benign prostate, primary prostate cancer (PCa), and metastatic CRPC tissue microarray. Scale bar: 50 μm. **b**) Representative multiplex IF images of NSD2 and CK-8 in adjacent benign and primary prostate cancer lesions in patient prostatectomies (n = 5 biological replicates, Scale bar: 50 μm). **c**) Integrated optical density quantification of NSD2 IF staining in benign (n = 10), primary PCa (n = 10), and mCRPC (n = 5) tissues. Box plot center, median; box, quartiles 1-3; whiskers, min and max values. **d**) Box plots of normalized ChIP-seq reads of distinct activating and repressive histone modifications at NSD2-dependent and NSD2-independent AR sites in VCaP cells (n = top 2000 sites, two-sided t-test). Box plot center, median; box, quartiles 1-3; whiskers, quartiles 1-3 ± 1.5 × interquartile range; dot, outliers. **e**) Box plots of normalized ChIP-seq reads of NSD2-catalyzed H3K36me2 and EZH2/PRC2-catalyzed H3K27me3 histone marks at NSD2-dependent and NSD2-independent AR sites in LNCaP cells (n = top 2000 sites, two-sided t-test). Box plot center, median; box, quartiles 1-3; whiskers, quartiles 1-3 ± 1.5 × interquartile range; dot, outliers. **f**) ChIP-seq read-density tracks of histone modification within a Chr10 locus in VCaP cells. NSD2-dependent and independent AR sites are marked in the tracks below with representative enhancers highlighted. **g**) ChIP-seq read-density box plots showing H3K36m2 (***top***) and H3K27me3 (***bottom***) signals at AR sites in NSD2-KO or WT LNCaP cells (n = top 2000 sites, two-sided t-test). Box plot center, median; box, quartiles 1-3; whiskers, quartiles 1-3 ± 1.5 × interquartile range; dot, outliers. **h**) ChIP-seq read-density tracks of H3K36me2 and H3K27me3 within a Chr10 locus in NSD2 WT and KO LNCaP cell lines. NSD2-dependent and independent AR sites are marked and highlighted.

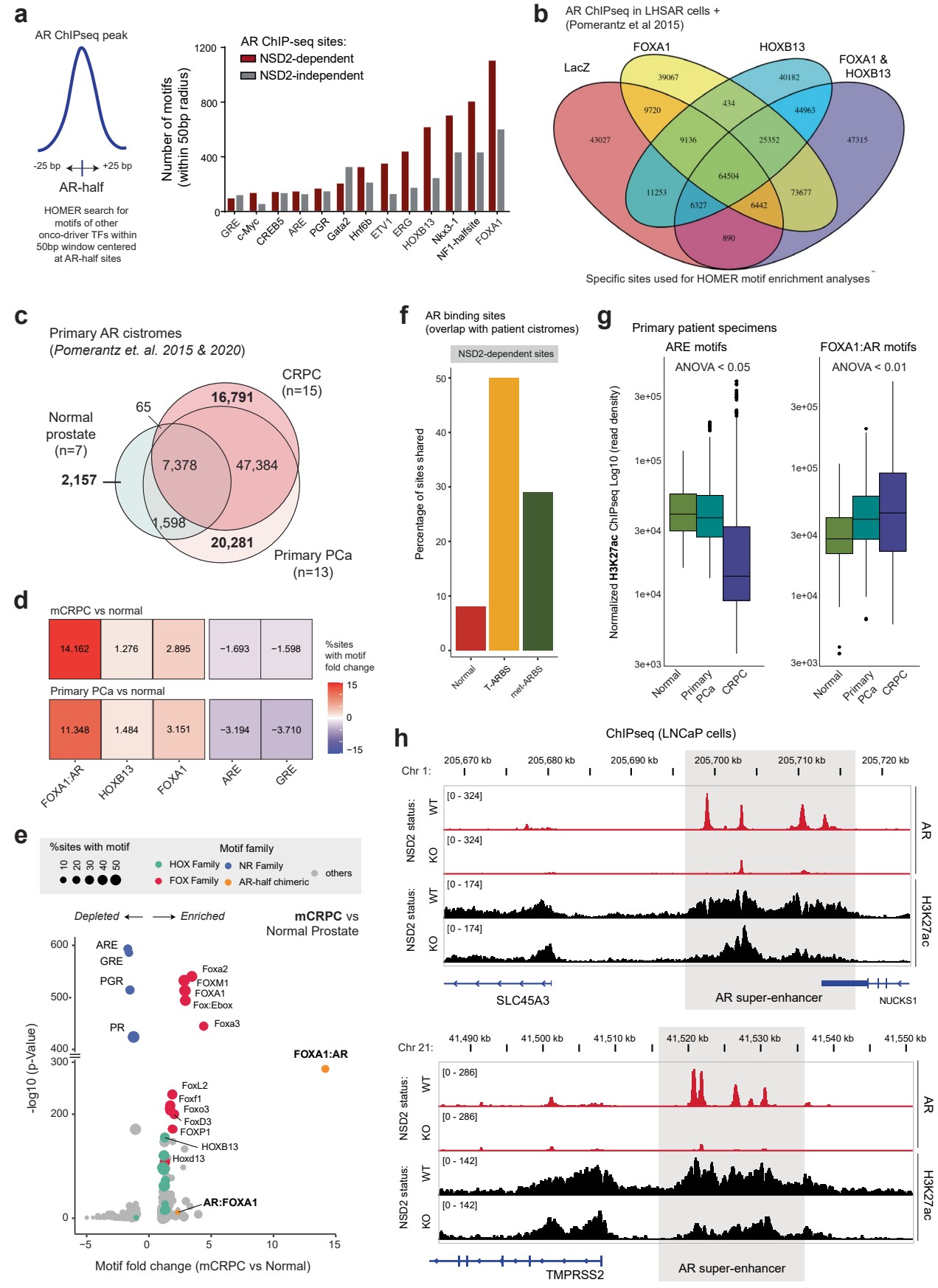

**Extended Data Fig. 4 | See next page for caption.**

**Extended Data Fig. 4 | Motif characterization of the NSD2-enabled AR neo-cistrome in prostate cancer cells. a**) *Left*: Schematic representation of the half-motif enrichment analysis. *Right*: Motif enrichment plot of AR half-motifs with neighboring motifs of other transcription factors at NSD2-dependent and independent AR sites in LNCaP cells. **b**) Venn diagram showing overlaps between AR ChIP-seq sites in LHSAR cells with LacZ (control), FOXA1, HOXB13, FOXA1 + HOXB13 overexpression. **c**) Venn diagram showing overlap of AR cistromes (ChIP-seq) in normal prostate, primary prostate cancer, and castration-resistant prostate cancer specimens. (Pomerantz et. al.[5,6]). **d**) Motif fold-change heatmap in normal, primary cancer, and castration-resistant prostate cancer specimens. **e**) Fold-change and significance of HOMER motifs enriched within mCRPC cancer-specific AR sites over normal tissue-specific AR elements (data from Pomerantz[5,6]). **f**) Barplot showing percentage of shared sites between the NSD2-dependent AR sites and AR cistromes from the normal prostate, primary PCa (T-ARBS), or metastatic CRPC (met-ARBS) patient tumors. **g**) Box plot showing H3K27ac ChIP-seq read density at sites containing the ARE or the FOXA1:AR motif in normal and tumor patient samples (normal prostate, n = 7; primary prostate cancer, n = 13; castration-resistant prostate cancer - CRPC, n = 15; one-way ANOVA and Tukey's test). Box plot center, median; box, quartiles 1-3; whiskers, quartiles 1-3 ± 1.5 × interquartile range; dot, outliers. **h**) ChIP-seq read-density tracks of AR and H3K27ac within the *SLC45A3* and *TMPRSS2* loci in NSD2 WT and NSD2-KO LNCaP cells. Super-enhancer clusters are highlighted in a gray box.

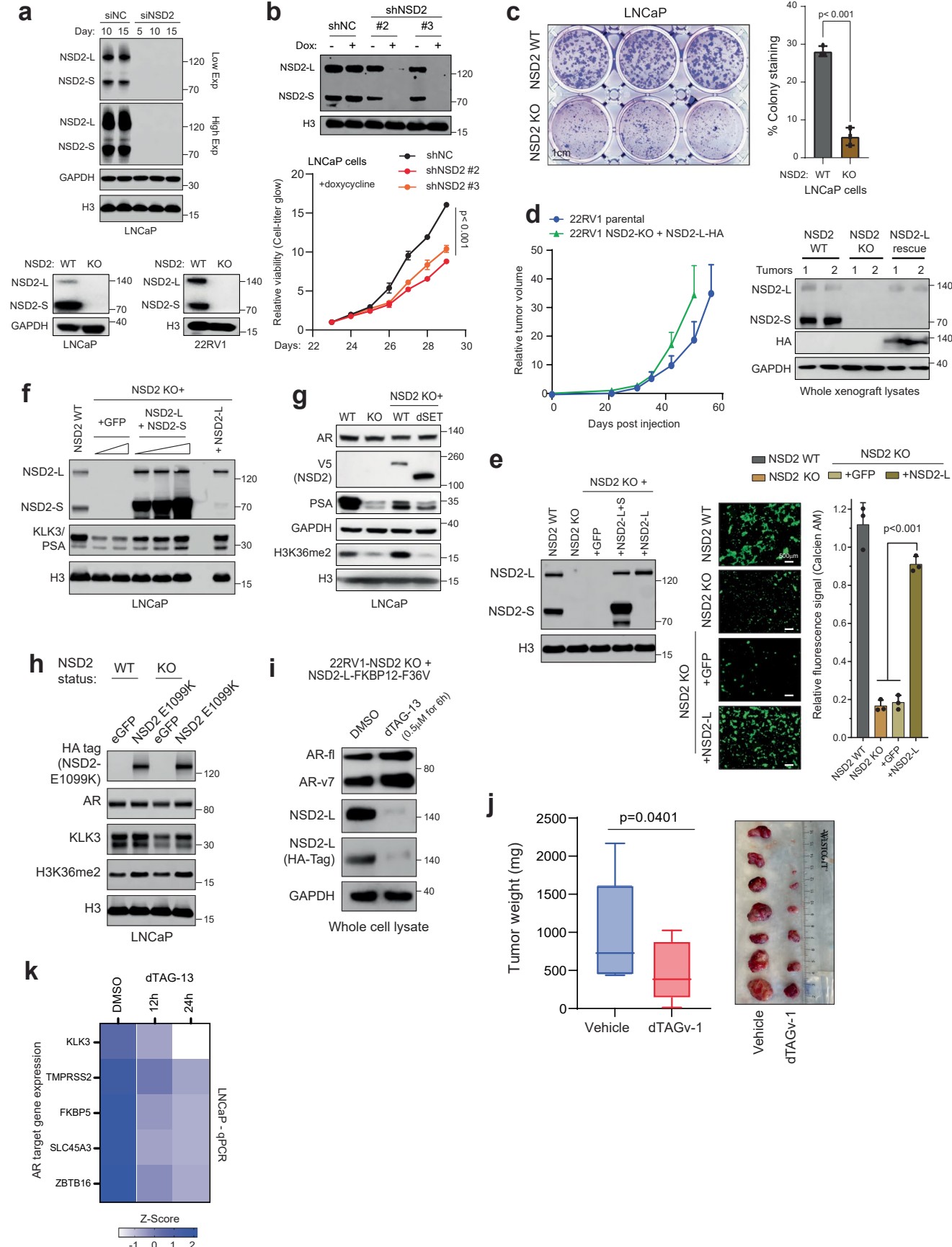

**Extended Data Fig. 5 | See next page for caption.**

**Extended Data Fig. 5 | Molecular characterization of the NSD2-rescued prostate cancer cells. a**) *Top*: Immunoblots of noted proteins upon long-term treatment with control (siNC) or NSD2-targeting siRNA (siNSD2). *Bottom*: Immunoblot of NSD2 in LNCaP and 22RV1 cells treated with a control sgRNA or sgRNA targeting NSD2. GAPDH and H3 are used as loading controls. **b**) *Top:* Immunoblots of NSD2 and H3 from stable shNSD2-expressing LNCaP cells +/- doxycycline (1ug/ml for 72 h). ***Bottom***: Growth curves (CTG) of control shRNA or shNSD2-expressing LNCaP cells plus doxycycline (n = 4 biological replicates, two-sided t-test). Mean +/- SEM are shown. **c**) *Left:* Representative images of colonies of control or NSD2-null LNCaP cells. *Right*: Quantification of stained colonies from left panel (n = 3 biological replicates, two-sided t-test). Mean +/- SEM are shown. Scale bar:1 cm. **d**) *Left*: Tumor volumes of 22RV1 parental or NSD2-KO + HA-tagged NSD2-L xenografts in mice. *Right*: Immunoblot of noted proteins from the 22RV1 xenograft tumors. (parental, n = 8; NSD2-KO, n = 7). Mean +/- SEM are shown. **e**) *Left*: Immunoblots showing expression of listed proteins in the eGFP or NSD2 overexpressing LNCaP cells. *Right:* Representative images from the Boyden chamber assay in the LNCaP NSD2 WT and KO or NSD2-L rescued lines. Fluorescence signal from invaded cells is shown (n = 3 biological replicates; one-way ANOVA and Tukey's test, Scale bar:500 µm). **f**) Immunoblot of listed proteins in wild-type or NSD2-KO LNCaP cells with stable exogenous overexpression of NSD2-L and/or NSD2-S isoforms. eGFP is used as control. **g**) Immunoblots of noted proteins in the NSD2 wild-type or NSD2-KO LNCaP cells rescued with exogenous WT or SET domain-deleted mutants. **h**) Immunoblots of noted proteins in LNCaP cells with hyper-catalytic NSD2 SET domain E1099K mutant. **i**) Immunoblots of noted proteins in the 22RV1-NSD2-KO + NSD2-L-FKBP12-F36V engineered cell lines +/- dTAG-13 treatment. **j**) *Left*: Tumor weights of 22RV1 + NSD2-FKBP12-F36V xenografts at endpoint (day 18) +/- dTAGv-1 (n = 10 biological replicates; two-sided t-test). *Right*: Tumor images at the endpoint from the animal growth studies. Box plot center, median; box, quartiles 1-3; whiskers, min and max values. **k**) Expression of AR target genes in the 22RV1 NSD2-KO + NSD2-L-FKBP12-F36V cell line +/- dTAG-13 treatment for 12 h or 24 h.

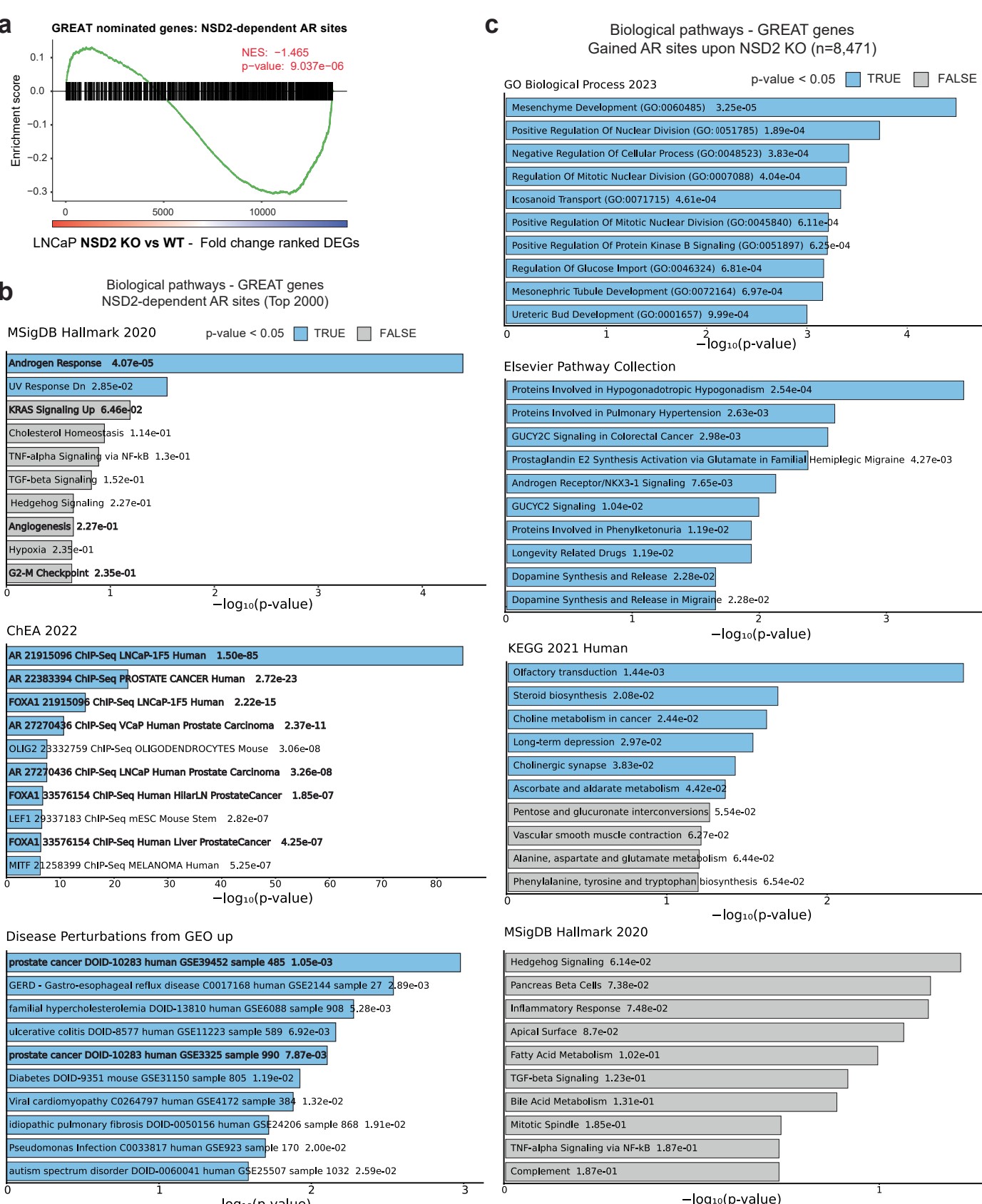

**Extended Data Fig. 6 | GREAT neighboring genes and pathway enrichment analyses. a)** Gene set enrichment analyses (GSEA) of GREAT nominated genes associated with the NSD2-dependent chimeric AR sites in NSD2-KO vs WT LNCaP cells (n = 2 biological replicates; GSEA enrichment test). **b)** GREAT and Enrichr analyses of putative chimeric AR gene targets in molecular signature and biology pathway databases (Fisher's exact test). **c)** GREAT and Enrichr analyses of putative gene targets of gained AR sites in the NSD2-KO LNCaP cells in pathway collections and databases (Fisher's exact test).

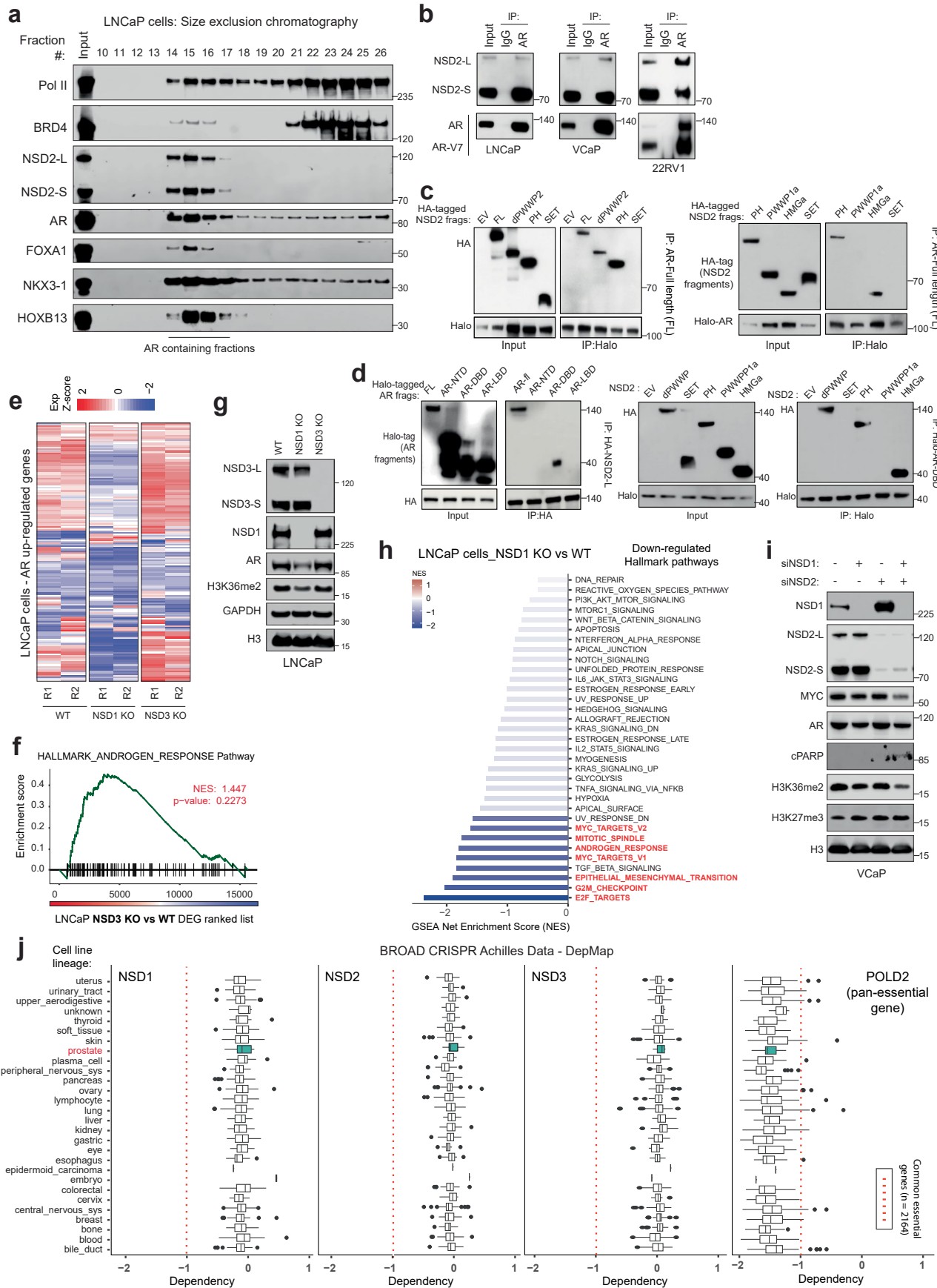

**Extended Data Fig. 7 | See next page for caption.**

**Extended Data Fig. 7 | Fragment-based NSD2–AR coimmunoprecipitation and characterization of the NSD paralog knockout prostate cancer cells.**
**a**) Immunoblots of noted proteins in size-exclusion chromatography fractions of nuclear lysate extracted from wild-type LNCaP cells. Fractions containing the AR protein are marked. **b**) Immunoblots of indicated proteins upon coimmunoprecipitation of AR in prostate cancer cells. **c**) Immunoblots of indicated proteins upon immunoprecipitation of exogenously expressed Halo-tagged full-length AR protein in HEK293FT cells that express HA-tagged NSD2 fragments. Both input (left) and immunoprecipitation (right) blots are shown. **d**) *Left*: Immunoblots of HA-tag-based immunoprecipitation of full-length NSD2 in HEK293FT cells that express the Halo-tagged AR protein fragments. *Right*: Immunoblots of Halo-tag-based immunoprecipitation of the DNA-binding domain (DBD) of AR in HEK293FT cells that overexpress different HA-tagged NSD2 fragments. For both experiments, input and immunoprecipitation

blots are shown. **e**) Heatmap of AR upregulated genes (z-score) in NSD1 or NSD3 knockout (KO) LNCaP cells. **f**) GSEA plots for AR-regulated genes using the fold-change rank-ordered genes from LNCaP NSD3 knockout (NSD3 KO) vs control cell lines. DEGS, differentially expressed genes (n = 2 biological replicates; GSEA enrichment test). **g**) Immunoblot of indicated proteins in NSD1 or NSD3-deficient LNCaP cells. **h**) GSEA net enrichment score (NES) plot of downregulated hallmark pathways in LNCaP NSD1 knocked out (KO) vs wild-type control cells. **i**) Immunoblot of indicated proteins upon treatment with NSD1 and NSD2-targeting siRNAs (labeled as siNSD1 and siNSD2) independently or in combination in VCaP cells. **j**) Dependency map (DepMap) plots showing the dependency scores for NSD1, NSD2, NSD3, and POLD2 (positive control; pan-essential gene) across cell lines from distinct originating tissues. The red dotted line indicates pan-essentiality z-score cutoff. Box plot center, median; box, quartiles 1-3; whiskers, quartiles 1-3 ± 1.5 × interquartile range; dot, outliers.

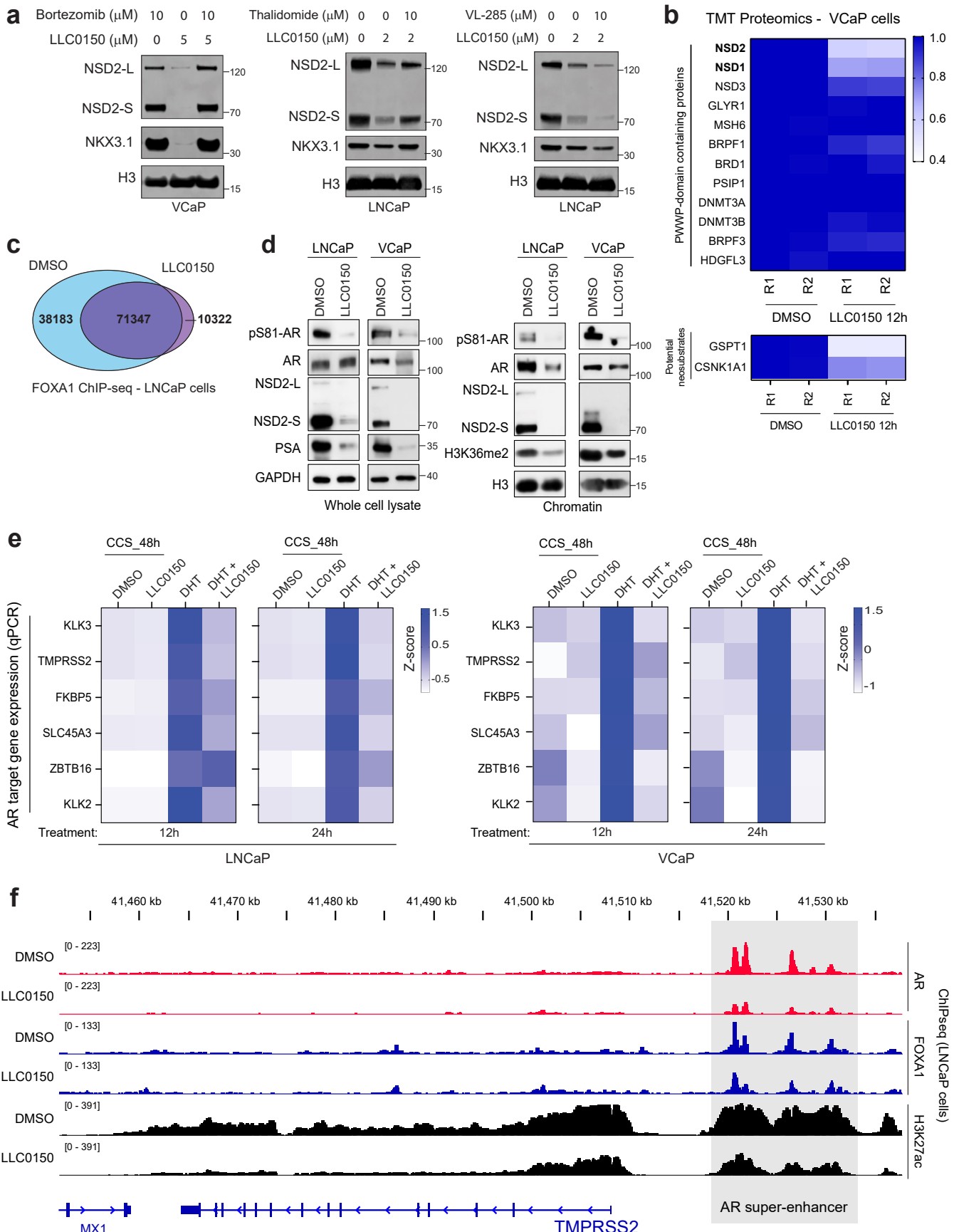

**Extended Data Fig. 8 | See next page for caption.**

**Extended Data Fig. 8 | Mechanistic characterization of the NSD1/2 PROTAC degrader LLC0150. a**) Immunoblots of indicated proteins in VCaP and LNCaP cells pre-treated with bortezomib, thalidomide, or VL-285 followed by treatment with LLC0150 at noted concentrations. **b**) Heatmap of relative abundance of several PWWP-domain-containing and known neo-substrate proteins detected via Tandem Mass Tag (TMT) based quantitative MS upon 12 h treatment with LLC0150 in VCaP cells. **c**) Genome-wide changes in FOXA1 ChIP-seq peaks in LNCaP cells treated with LLC0150 (2uM for 48 h). **d**) Immunoblots of noted proteins in whole-cell or chromatin lysates from VCaP and LNCaP cells treated with LLC0150 (2uM) for 24 h. **e**) Heatmap of z-score normalized expression (qRT-PCR) of AR target genes in LNCaP and VCaP cells treated with LLC0150 followed by DHT stimulation (10 nM for 24 h). Treatment with DHT alone is used as a control. CCS, charcoal-stripped serum. **f**) Read-density ChIP-seq tracks of AR, FOXA1, and H3K27ac within the *TMPRSS2* super-enhancer in LNCaP cells treated with LLC0150 (2uM for 24 h). Super-enhancer cluster is highlighted in a gray box.

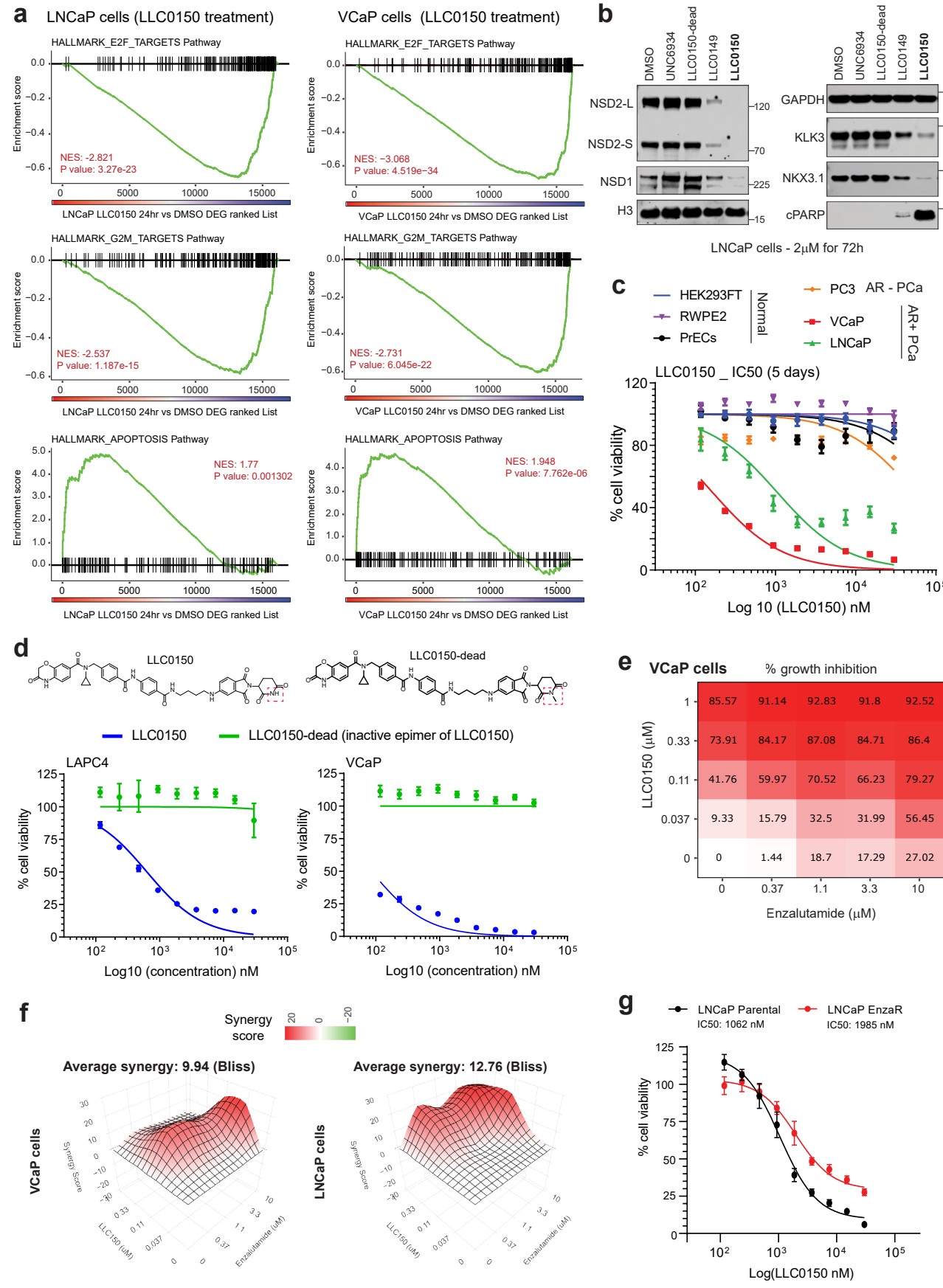

Extended Data Fig. 9 | See next page for caption.

**Extended Data Fig. 9 | Transcriptomic effect and drug synergism of LLC0150 in prostate cancer cells. a**) GSEA plots for E2F, G2M, and apoptosis pathway genes using the fold-change rank-ordered genes from the LLC0150 vs DMSO treated LNCaP (left) or VCaP (right) cell lines. DEGS, differentially expressed genes (n = 2 biological replicates; GSEA enrichment test). **b**) Immunoblot of noted proteins in LNCaP cells treated with LLC0150 (2uM for 72 h), dead-analog (LLC0150-dead), or the warhead alone (UNC6934). LLC0149 is an independent NSD1/2 PROTAC. **c**) Dose-response curves of LLC0150 in normal prostate, AR-positive, or AR-negative prostate cancer cell lines at the indicated concentrations for five days. (PrECs, n = 3 biological replicates; others, n = 6 biological

replicates). Mean +/- SEM are shown. **d**) Dose-response curves of LLC0150 and its inactive epimer control (LLC0150-dead) in LAPC4 and VCaP cell lines (n = 6 biological replicates). Mean +/- SEM are shown. **e**) Percent growth inhibition (Cell-titer Glo) of VCaP cells upon co-treatment with varying concentrations of LLC0150 and enzalutamide for 5 days. **f**) 3D synergy plots of LLC0150 and enzalutamide co-treated LNCaP and VCaP cells. Red peaks in the 3D plots denote synergy with the average synergy scores noted above. **g**) Dose-response curves of LLC0150 in LNCaP parental and enzalutamide-resistant cell lines at varying concentrations for five days. Half-maximal inhibitory concentrations (IC50) are noted (n = 5 biological replicates). Mean +/- SEM are shown.

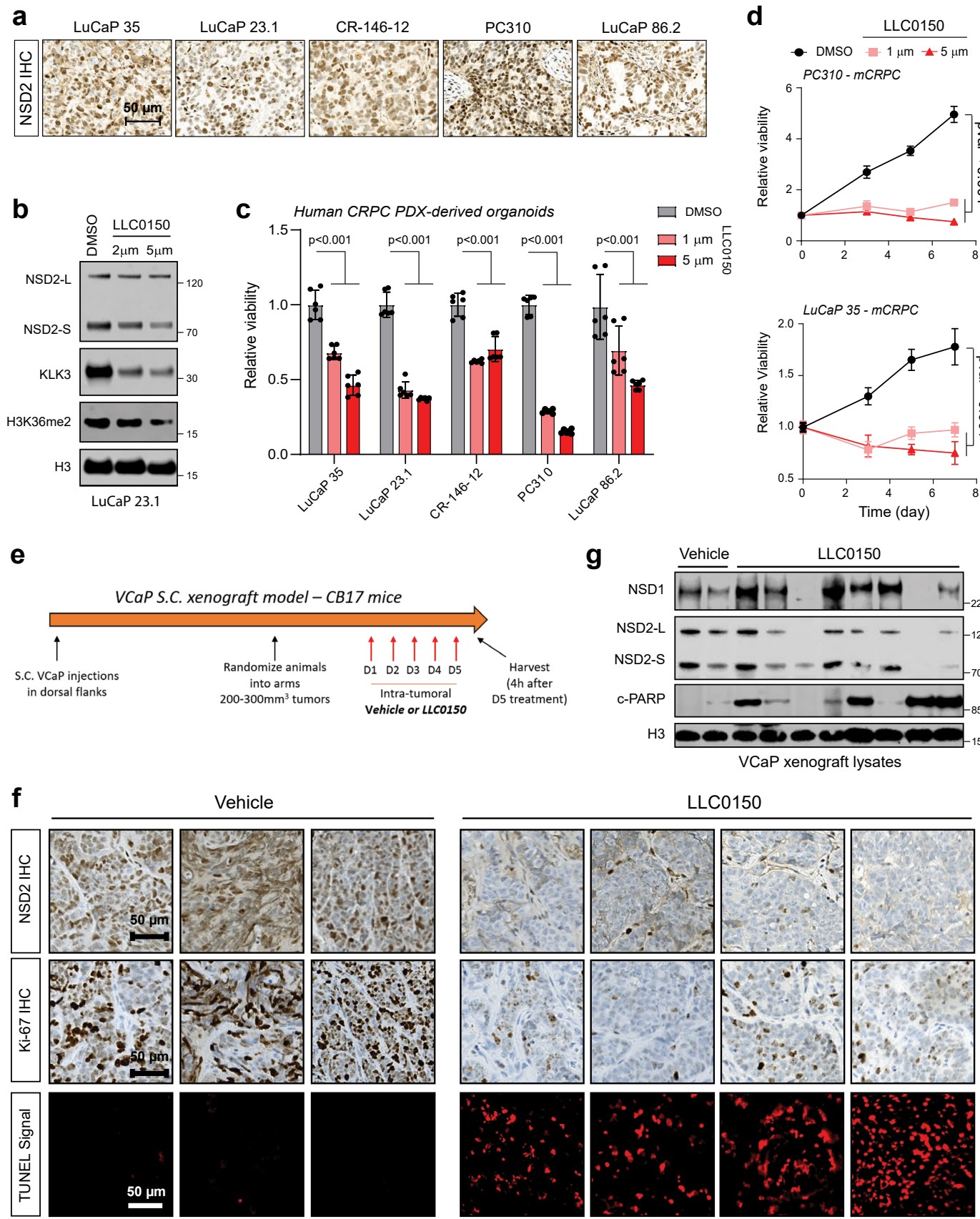

**Extended Data Fig. 10 | See next page for caption.**

**Extended Data Fig. 10 | Efficacy assessment of LLC0150 in prostate cancer organoids and xenografts. a**) Representative images of NSD2 IHC in a panel of patient-derived xenografts (PDXs). Scale bar:50 μm. **b**) Immunoblot of NSD2, AR targets, and histone marks in the LuCaP 23.1 PDX-derived organoid line treated with LLC0150, the NSD1/2 degrader. Total H3 is the loading control. **c**) Barplots showing relative viability of the PDX-derived organoid lines treated with two doses of LLC0150. In all lines, degradation of NSD1/2 reduces cell viability (n = 6 biological replicates; one-way ANOVA and Tukey's test). Mean +/- SEM are shown. **d**) Growth curves (Cell-titer Glo) of two representative AR/NSD2-positive PDX-derived organoid lines treated with DMSO or LLC0150 (n = 6 biological replicates;

one-way ANOVA and Tukey's test). Mean +/- SEM are shown. **e**) Schematic overview of the LLC0150 intratumoral injection study in a VCaP xenograft model. **f**) ***Top***: Representative NSD2 and KI67 IHC images in the vehicle and LLC0150-treated tumors. Loss of NSD2 correlates with a reduction in proliferating KI67-positive cells. ***Bottom***: TUNEL assay in the vehicle and LLC0150-treated tumors reveal a high number of apoptotic (TUNEL-positive) cells in the drug-treated tumors.Vehicle, n = 3 tumors; LLC0150, n = 4 tumors. Scale bar: 50 μm. **g**) Immunoblot of noted proteins in matched tumor lysates from f. Total H3 is used as a loading control.

# Reporting Summary

## Statistics

For all statistical analyses, confirm that the following items are present in the figure legend, table legend, main text, or Methods section.

| n/a | Confirmed | |
|---|---|---|
| ☐ | ☒ | The exact sample size (*n*) for each experimental group/condition, given as a discrete number and unit of measurement |
| ☐ | ☒ | A statement on whether measurements were taken from distinct samples or whether the same sample was measured repeatedly |
| ☐ | ☒ | The statistical test(s) used AND whether they are one- or two-sided *Only common tests should be described solely by name; describe more complex techniques in the Methods section.* |
| ☐ | ☒ | A description of all covariates tested |
| ☐ | ☒ | A description of any assumptions or corrections, such as tests of normality and adjustment for multiple comparisons |
| ☐ | ☒ | A full description of the statistical parameters including central tendency (e.g. means) or other basic estimates (e.g. regression coefficient) AND variation (e.g. standard deviation) or associated estimates of uncertainty (e.g. confidence intervals) |
| ☐ | ☒ | For null hypothesis testing, the test statistic (e.g. *F*, *t*, *r*) with confidence intervals, effect sizes, degrees of freedom and *P* value noted *Give P values as exact values whenever suitable.* |
| ☒ | ☐ | For Bayesian analysis, information on the choice of priors and Markov chain Monte Carlo settings |
| ☒ | ☐ | For hierarchical and complex designs, identification of the appropriate level for tests and full reporting of outcomes |
| ☒ | ☐ | Estimates of effect sizes (e.g. Cohen's *d*, Pearson's *r*), indicating how they were calculated |

*Our web collection on statistics for biologists contains articles on many of the points above.*

## Software and code

Policy information about availability of computer code

| Data collection | No software was used for data collection. |
|---|---|
| Data analysis | All custom codes used for data analyses are freely available from the following public repositories: <br> https://github.com/mcieslik-mctp/papy <br> https://github.com/mcieslik-mctp/hpseq <br> https://github.com/mcieslik-mctp/bootstrap-rnascape <br> https://github.com/mcieslik-mctp/codac <br> https://github.com/mcieslik-mctp/crisp <br> https://github.com/mcieslik-mctp/ <br> https://github.com/mctp/ <br><br> Computational tools used: <br> GraphPad Prism 9 and in-built statistical tools <br> bcl2fastq conversion software (v2.20) <br> Pairtools (version 0.3.0) <br> EdgeR (bulk: edgeR_3.39.6, single cell: v3.36.0) <br> Limma-Voom (limma_3.53.10) <br> fgsea (fgsea_1.24.0) <br> EnhancedVolcano (EnhancedVolcano_1.15.0) <br> R (R version 4.2.1 70–72) <br> ChipPeakAnno (version 3.0.0) |

ChipSeeker (version 1.29.1)
Sushi (Sushi_1.32.0)
Trimmomatic (version 0.39)
bwa ( version 0.7.17-r1198-dirty)
samtools (v1.1)
picard MarkDuplicates (v2.26.0-1-gbaf4d27-SNAPSHOT)
MACS2  (v2.2.7.1)
bedtools (v2.27.1)
UCSC's tool wigtoBigwig  (v2.8)
bowtie2 v2.5.1
HOMER (version v.4.10)
Deeptools (v3.5.1)
Kallisto (version 0.46.1)

Webtools used:
GREAT (v4.0.4; from http://great.stanford.edu/public/html/)
Enrichr (from http://amp.pharm.mssm.edu/Enrichr/
XSTREME from the MEME Suite (v 5.5.5; from https://meme-suite.org/meme/doc/xstreme.html?man_type=web)

For manuscripts utilizing custom algorithms or software that are central to the research but not yet described in published literature, software must be made available to editors and reviewers. We strongly encourage code deposition in a community repository (e.g. GitHub). See the Nature Portfolio guidelines for submitting code & software for further information.

# Data

Policy information about availability of data

All manuscripts must include a data availability statement. This statement should provide the following information, where applicable:
- Accession codes, unique identifiers, or web links for publicly available datasets
- A description of any restrictions on data availability
- For clinical datasets or third party data, please ensure that the statement adheres to our policy

All data are available in the manuscript or the supplementary information. All materials are available from the authors upon request. All raw next-generation sequencing, including ChIP-seq and RNA-seq, data generated in this study are deposited in the Gene Expression Omnibus (GEO) repository (accession number: GSE242737) at NCBI and is publicly available. Additionally, AR ChIPseq from castration-resistant prostate cancer (CRPC), normal, and primary prostate cancer (PCa) were pulled from GEO repositories from Baca et al and Pomerantz et al  (GSE130408 and GSE70079). Public scRNA-seq datasets from primary prostate cancer were downloaded from GEO or a website provided by the authors (GSE193337, GSE185344, www.prostatecellatlas.org). Functional gene sets were pulled from hallmark and C2 MsigDB pathways (/www.gsea-msigdb.org).

# Research involving human participants, their data, or biological material

Policy information about studies with human participants or human data. See also policy information about sex, gender (identity/presentation), and sexual orientation and race, ethnicity and racism.

| Reporting on sex and gender | All patient samples used in this study are from the University of Michigan pathological archives. All the specimens are from male patients given that the study is focused on prostate cancer. |
|---|---|
| Reporting on race, ethnicity, or other socially relevant groupings | All samples were from White, Non-Hispanic patients at the University of Michigan. |
| Population characteristics | N/A |
| Recruitment | Patient tissues from biopsies of prostate tumors were acquired from the University of Michigan pathology archives. |
| Ethics oversight | Use of clinical formalin-fixed paraffin embedded specimens from the archives was approved by the University of Michigan Institutional Review Board and does not require patient consent. |

Note that full information on the approval of the study protocol must also be provided in the manuscript.

# Field-specific reporting

Please select the one below that is the best fit for your research. If you are not sure, read the appropriate sections before making your selection.

☒ Life sciences          ☐ Behavioural & social sciences          ☐ Ecological, evolutionary & environmental sciences

For a reference copy of the document with all sections, see nature.com/documents/nr-reporting-summary-flat.pdf

# Life sciences study design

All studies must disclose on these points even when the disclosure is negative.

| | |
|---|---|
| Sample size | All sample size details for the analyses carried out in this study are reported in the Methods section and/or figure legends. No statistical methods were used to predetermine sample sizes. Sample sizes were based on prior research experience of similar assays rather than a power analyses. |
| Data exclusions | No data was excluded from the published publicly-available patient sequencing studies. For biologically experiments, no data exclusions were made. For the in vivo studies, we have used 5-7 mice bearing 10-14 tumors in each group to allow for statistical assessments. |
| Replication | All in vitro experiments were independently repeated at least three times, with all replication attempts producing similar results. Reproducibility between RNAseq and ChIPseq samples was assessed on normalized alignment files using principal component analysis, unsupervised hierarchal clustering, or correlation analyses with good reproducibility observed across replications. |
| Randomization | For animal studies, mice were randomly assigned to treatment groups. For all other in vitro experiments, we used a common cell suspension to plate for both control and treatment groups. |
| Blinding | All histo-pathological evaluations of tissues and IHC/staining-based scoring for drug toxicity studies were carried out in a blinded manner by two independent pathologists. For all other experiments, the analyses did not require blinding as data quantification was carried out using instruments and automated workflows with no manual steps. |

# Reporting for specific materials, systems and methods

We require information from authors about some types of materials, experimental systems and methods used in many studies. Here, indicate whether each material, system or method listed is relevant to your study. If you are not sure if a list item applies to your research, read the appropriate section before selecting a response.

### Materials & experimental systems

| n/a | Involved in the study |
|---|---|
| ☐ | ☒ Antibodies |
| ☐ | ☒ Eukaryotic cell lines |
| ☒ | ☐ Palaeontology and archaeology |
| ☐ | ☒ Animals and other organisms |
| ☒ | ☐ Clinical data |
| ☒ | ☐ Dual use research of concern |
| ☒ | ☐ Plants |

### Methods

| n/a | Involved in the study |
|---|---|
| ☐ | ☒ ChIP-seq |
| ☒ | ☐ Flow cytometry |
| ☒ | ☐ MRI-based neuroimaging |

## Antibodies

| | |
|---|---|
| Antibodies used | Target antigen; Vendor; Catalog number; Lot number; Application<br>NSD1 (NeuroMab: 75-280, Clone#N312/10;  Western Blot 1:500);<br>NSD2 (Abcam:ab75359, Clone#29D1, Western Blot 1:1000, Immunofluorescence 1:200);<br>NSD3(Cell Signaling Technologies: 92056S, Clone#D4N9N, Western Blot 1:1000);<br>KLK3/PSA (Dako:A0562, Lot: 00093790, Western Blot 1:2000),<br>FKBP5(Cell Signaling Technologies: 12210,Clone#: D5G2, Western Blot 1:1000),<br>NKX3-1(Cell Signaling Technologies:83700S, clone#: D2Y1A, Western Blot 1:1000),<br>FOXA1 C-terminal (Thermo Fisher Scientific: PA5-27157, Lot# VFS004672A, Western Blot 1:1000 and ChIP-seq 2ug/4M cells);<br>AR (Millipore: 06-680, Western blot 1:1000 and ChIP-seq 2ug/4M cells));<br>AR (Abcam: ab133273, Clone#: EPR1535(2), Western blot 1:1000);<br>H3 (Cell Signaling Technologies: 3638S, Clone#: 96C10, Western blot 1:2000);<br>GAPDH (Cell Signaling Technologies: 3683, Clone#: 14C10, Western blot) 1:2000;<br>H3K27me3(Millipore: 07-449, Western blot 1:2000 and ChIP-seq 1ug/2M cells);<br>H3K36me2 (Cell Signaling Technologies: 2901S, Western blot 1:2000);<br>H3K27Ac (Active Motif, Cat#39336; ChIP-seq 1ug/2M cells);<br>H3K4me1 (Abcam: ab8895; ChIP-seq 1ug/2M cells);<br>H3K4me2 (CST: C64G9; Clone#: C64G9, ChIP-seq 1ug/2M cells);<br>H3K36me2 (Abcam: ab9049; ChIP-seq 1ug/2M cells);<br>Phospho-AR (Ser-81) (Millipore, Cat# 07-1375-EMD, Western Blot 1:1000),<br>HALO (Promega ,Cat# G9281, Western Blot 1:1000 and co-immunopreciptation),<br>HA (Cell Signaling Technologies, Cat# 3724S, Clone#: C29F4, Western Blot 1:1000 and co-immunopreciptation),<br>His (Cell Signaling Technologies, Cat#2365,Western Blot 1:1000 and co-immunopreciptation)<br>CK8 (Abcam, epitope: Clone#: EP1628Y, Cat# ab53280, Immunofluorescence 1:200) |
| Validation | All antibodies used in this study are from reputed commercial vendors and have been validated by the vendors (see website). QC data is directly available from all the vendor listed above and these antibodies have been commonly used in other publications.<br>NSD1, https://www.antibodiesinc.com/products/anti-nsd1-antibody-n312-10-75-280 |

Manufacturer states that the antibody "is produced in-house from hybridoma clone N312/10. It detects human and mouse NSD1, and is purified by Protein A chromatography." Also, "[antibody] does not cross-react with NSD2 or NSD3. Each new lot of antibody is quality control tested on cells overexpressing target protein and confirmed to give the expected staining pattern."

NSD2, https://www.abcam.com/products/primary-antibodies/whsc1nsd2-antibody-29d1-ab75359.html. Manufacturer states, its "suitable for IHC, IP, WB and reacts with human samples."

NSD3, https://www.cellsignal.com/products/primary-antibodies/whsc1l1-d4n9n-rabbit-mab/92056. Manufacturer: "Monoclonal antibody is produced by immunizing animals with a synthetic peptide corresponding to residues surrounding Pro117 of human WHSC1L1 protein. It recognizes endogenous levels of total WHSC1L1 protein, both long and short isoforms."

KLK3/PSA, https://www.citeab.com/antibodies/3382929-a0562-prostate-specific-antigen-psa. We have validated this antibody in our lab by treating LNCaP and VCaP cells with AR antagonistic drugs that led to a marked decrease in KLK3/PSA levels. We have used it in numerous publications from our group.

FKBP5, https://www.cellsignal.com/products/primary-antibodies/fkbp5-d5g2-rabbit-mab/12210. Manufacturer: "Monoclonal antibody is produced by immunizing animals with a synthetic peptide corresponding to residues surrounding Arg222 of human FKBP5 protein. This antibody does not cross-react with FKBP4 protein."

NKX3-1, https://www.cellsignal.com/products/primary-antibodies/nkx3-1-d2y1a-xp-rabbit-mab/83700. Manufacturer states this antibody is reactive to human NKX3.1 and its specificity was confirmed by running a blot with prostate cancer cells (positive control) and DND-41 cells that are negative for NKX3.1 expression.

FOXA1, https://www.thermofisher.com/antibody/product/FOXA1-Antibody-Polyclonal/PA5-27157. Manufacturer: "This Antibody was verified by Knockdown to ensure that the antibody binds to the antigen stated."

AR Millipore, https://www.emdmillipore.com/US/en/product/Anti-Androgen-Receptor-Antibody,MM_NF-06-680. Manufacturer: "This antibody recognizes the Modulation Region within the N-terminus of Androgen Receptor." It was validated by western blotting in LNCaP cells.

AR abcam, https://www.abcam.com/products/primary-antibodies/androgen-receptor-antibody-epr15352-ab133273.html. Manufacturer states that this antibody reacts with Mouse, Rat, Human samples.

H3, https://www.cellsignal.com/products/primary-antibodies/histone-h3-96c10-mouse-mab/3638. Manufacturer states that this antibody "detects endogenous levels of total Histone H3 protein, including isoforms H3.1, H3.2, and H3.3. The antibody does not cross-react with other histone proteins, including the Histone H3 variant CENP-A." It has been validated using western blotting.

GAPDH, https://www.cellsignal.com/products/antibody-conjugates/gapdh-14c10-rabbit-mab-hrp-conjugate/3683. Manufacturer states it reacts with human, mouse, rat and monkey GAPDH and has been validated using western blotting.

H3K27me3, https://www.emdmillipore.com/US/en/product/Anti-trimethyl-Histone-H3-Lys27-Antibody,MM_NF-07-449?ReferrerURL=https%3A%2F%2Fwww.google.com%2F. Manufacturer: "[Antibody is] routinely evaluated by western blot in acid extracted proteins from HeLa cells."

H3K36me2, https://www.cellsignal.com/products/primary-antibodies/di-methyl-histone-h3-lys36-c75h12-rabbit-mab/2901. Manufacturer states that this antibody "detects endogenous levels of histone H3.1, histone H3.2, and histone H3.3, only when di-methylated on Lys36. The antibody does not cross-react with non-methylated, mono-methylated, or tri-methylated Lys36. In addition, the antibody does not cross-react with di-methylated histone H3 Lys4, Lys9, Lys27, Lys79 or di-methylated histone H4 Lys20." It has been validated by western blotting.

H3K27Ac , https://www.activemotif.com/catalog/details/39135/histone-h3-acetyl-lys27-antibody-pab-1. Manufacturer states that the antibody has been tested by Western blot as well as the dot blot analysis"

Phospho-AR (Ser-81) ,https://www.emdmillipore.com/US/en/product/Anti-phospho-Androgen-Receptor-Ser81-Antibody,MM_NF-07-1375. Manufacturer states, " Detects Androgen Receptor (AR) only when phosphorylated on Ser81" and is reactive only to human AR. It has been validated by running western blots on LNCaP lysates.

HALO, https://www.promega.com/products/protein-detection/primary-and-secondary-antibodies/anti-halotag-pab/?catNum=G9281. Manufacturer: " The antibody is purified using Protein G affinity resin and supplied at 1mg/ml in PBS. The antibody detects HaloTag® fusion proteins in Western blot hybridization and immunocytochemistry applications with high sensitivity and specificity."

HA, https://www.cellsignal.com/products/primary-antibodies/ha-tag-c29f4-rabbit-mab/3724. Manufacturer: The antibody was validated using "western blot analysis of extracts from HeLa cells untransfected or transfected with either HA-FoxO4 or HA-Akt3 plasmids. The antibody may cross-react with a protein of unknown origin ~100kDa."

His, https://www.cellsignal.com/products/images/2365_ific_jp.jpg6. Manufacturer: the antibody was validated using "western blot analysis of extracts from cells expressing C-terminal His-tagged protein or control extract. It detects recombinant proteins containing the 6xHis epitope tag. The antibody recognizes the 6xHis-tag fused to either the amino or carboxy terminus of targeted proteins in transfected cells. The antibody may cross-react with a protein of unknown origin ~60-70kDa."

H3K4me1 (Abcam: ab8895), https://www.abcam.com/products/primary-antibodies/histone-h3-mono-methyl-k4-antibody-chip-grade-ab8895.html Manufacturer: The anitbody is suitable for IHC-P, ICC/IF, ChIP, WB and reacts with Human, Mouse, Rat, Cow samples.

H3K4me2 (CST: C64G9), https://www.cellsignal.com/products/primary-antibodies/di-methyl-histone-h3-lys4-c64g9-rabbit-

mab/9725. Manufacturer: The antibody has been validated using "western blot analysis of whole cell lysates from HeLa, NIH/3T3, C6 and COS cells. It detects endogenous levels of histone H3 when di-methylated on Lys4. This antibody shows weak cross-reactivity with histone H3 that is mono-methylated on Lys4 but does not cross-react with non-methylated or tri-methylated histone H3 Lys4. In addition, the antibody does not cross-react with methylated histone H3 Lys9, Lys27, Lys36 or histone H4 Lys20."

CK8 (Abcam: ab53280): https://www.abcam.com/en-us/products/primary-antibodies/cytokeratin-8-antibody-ep1628y-cytoskeleton-marker-ab53280. Manufacturer: "Antibody is suitable for IP, WB, IHC-Fr, ICC/IF, IHC-P and reacts with Human, Mouse, Rat samples."

# Eukaryotic cell lines

Policy information about cell lines and Sex and Gender in Research

Cell line source(s)

Most cell lines were originally obtained from ATCC, DSMZ, ECACC, or internal stock. All the cells were genotyped to confirm their identity at the University of Michigan Sequencing Core and tested routinely for Mycoplasma contamination. Additionally, all the cell lines were genotyped every two months to confirm their identity. Cells were grown media conditions prescribed by ATCC, DSMZ or ECACC.

Here is the list of all the cell lines used in this study:
RS4;11
MOLT3
RPMI-8402
MM1.S
H1048
MM1.R
TC-205
H1836
TC32
MOLT4
DoHH2
CHLA-9
CHLA-258
SUM185PE
NB-1643
CB-AGPN
LASCPC-01
CAMA-1
SEM
R-CHACV
T47D
VCaP
MDA-PCa-2b
KARPAS-25
COG-N-561
CHLA-218
MDA-MB-330
H446
PA-1
MFM-223
WA-72-PS
22RV1
LNCaP
MDA-MB-468
CWRR1
MDA-MB-453
N87
IMR90
WA-72-As
COG-E-352
H524
NTERA-2
LAPC4
ZR-75-1
CHLA-99
CHLA-25
Caov-3
MEG-01
BEAS-2B
COLO205
MDA-MB-231
MDA-MB-436
5637
H211
SAOS-2

HCC1143
HCC1146
TC-106
TC-138
SW626
HeLa
HCC1187
MDA-MB-415
957/hTERT
COG-N-557
SNU1079
SNU387
BJ
HK2
PC3
RPB1293
H69
RWPE1
A673
TE6
SK-N-MC
HEK293FT
SCaBER
UM-UC3
PrECs
OC-8
HCC1428
RPB1292
SNU16
SNU423
HEPG2
PNT2
KATO III
SK-OV-3
PLC/PRF/5
COG-N-529
MCF7
RPMI8226
K562
DU145
786-O
AGS
SNU840
MCF10A
RWPE2-W99
HT115
CADO-ES-1
SNU-5
DAN-G
SK-HEP-1
LHSAR
SK-MEL-5
WPMY1
ACHN
U2OS
TC-71
H716
LAMA87
A549

Authentication

All cell lines were biweekly tested to be free of mycoplasma contamination and genotyped every month at the University of Michigan Sequencing Core using Profiler Plus (Applied Biosystems) and compared with corresponding short tandem repeat (STR) profiles in the ATCC database to authenticate their identity in culture between passages and experiments.

Mycoplasma contamination

All cells were biweekly tested for mycoplasma contamination using the MycoAlert PLUS Mycoplasma Detection Kit (Lonza) and were found to be continually negative. More details are included in the Methods section

Commonly misidentified lines
(See ICLAC register)

None

# Animals and other research organisms

Policy information about [studies involving animals](); [ARRIVE guidelines]() recommended for reporting animal research, and [Sex and Gender in Research]()

| | |
|---|---|
| Laboratory animals | NOD/SCID mice were obtained from commercial sources. All mice were housed in a pathogen-free animal barrier facility and all in vivo experiments were initiated with male mice aged 5-8 weeks. All mice were maintained under the conditions of pathogen-free, 12 hours light/12 hours dark cycle, temperatures of 18-23°C, and 40-60% humidity. |
| Wild animals | No wild animals were used in the study. |
| Reporting on sex | Male animals were used since prostate cancer is specific to males. |
| Field-collected samples | No field collected samples were used in the study. |
| Ethics oversight | The Institutional Animal Care & Use Committee (IACUC) ensures that the highest animal welfare standards are maintained along with the conduct of accurate, valid scientific research through the supervision, coordination, training, guidance, and review of every project proposed to include the use of vertebrate animals at the University of Michigan and the University of Pennsylvania. |

Note that full information on the approval of the study protocol must also be provided in the manuscript.

# Plants

| | |
|---|---|
| Seed stocks | Not applicable |
| Novel plant genotypes | Not applicable |
| Authentication | Not applicable. |

# ChIP-seq

## Data deposition

☒ Confirm that both raw and final processed data have been deposited in a public database such as [GEO]().

☒ Confirm that you have deposited or provided access to graph files (e.g. BED files) for the called peaks.

| | |
|---|---|
| Data access links *May remain private before publication.* | All raw next-generation sequencing, including ChIP-seq and RNA-seq, data generated in this study are deposited in the Gene Expression Omnibus (GEO) repository (accession number: GSE242737) at NCBI. The following secure token has been created to allow review of record GSE242737 while it remains in private status: uzmjkswedlmtbmb. |
| Files in database submission | These NGS fastq files will be deposited as part of this study to GEO:<br><br>RNASeq:<br>LNCaP_sgNCx3_rep1_R1.fq.gz<br>LNCaP_sgNCx3_rep2_R1.fq.gz<br>LNCaP_sgNSD1_rep1_R1.fq.gz<br>LNCaP_sgNSD1_rep2_R1.fq.gz<br>LNCaP_sgNSD3_rep1_R1.fq.gz<br>LNCaP_sgNSD3_rep2_R1.fq.gz<br>VCaP_siNC_DMSO_Rep1_R1.fq.gz<br>VCaP_siNC_DMSO_Rep2_R1.fq.gz<br>VCaP_siNC_EPZ6438_72h_Rep1_R1.fq.gz<br>VCaP_siNC_EPZ6438_72h_Rep2_R1.fq.gz<br>VCaP_siNSD1_DMSO_Rep1_R1.fq.gz<br>VCaP_siNSD1_DMSO_Rep2_R1.fq.gz<br>VCaP_siNSD1and2_DMSO_Rep1_R1.fq.gz<br>VCaP_siNSD1and2_DMSO_Rep2_R1.fq.gz<br>VCaP_siNSD2_DMSO_Rep1_R1.fq.gz<br>VCaP_siNSD2_DMSO_Rep2_R1.fq.gz<br>LNCaP_DMSO_rep1_R1.fq.gz<br>LNCaP_DMSO_rep2_R1.fq.gz<br>LNCaP_LLC0150_24hr_rep1_R1.fq.gz<br>LNCaP_LLC0150_24hr_rep2_R1.fq.gz<br>LNCaP_sgNCx3_rep1_R2.fq.gz<br>LNCaP_sgNCx3_rep2_R2.fq.gz<br>LNCaP_sgNSD1_rep1_R2.fq.gz<br>LNCaP_sgNSD1_rep2_R2.fq.gz<br>LNCaP_sgNSD3_rep1_R2.fq.gz<br>LNCaP_sgNSD3_rep2_R2.fq.gz |

```
VCaP_siNC_DMSO_Rep1_R2.fq.gz
VCaP_siNC_DMSO_Rep2_R2.fq.gz
VCaP_siNC_EPZ6438_72h_Rep1_R2.fq.gz
VCaP_siNC_EPZ6438_72h_Rep2_R2.fq.gz
VCaP_siNSD1_DMSO_Rep1_R2.fq.gz
VCaP_siNSD1_DMSO_Rep2_R2.fq.gz
VCaP_siNSD1and2_DMSO_Rep1_R2.fq.gz
VCaP_siNSD1and2_DMSO_Rep2_R2.fq.gz
VCaP_siNSD2_DMSO_Rep1_R2.fq.gz
VCaP_siNSD2_DMSO_Rep2_R2.fq.gz
LNCaP_DMSO_rep1_R2.fq.gz
LNCaP_DMSO_rep2_R2.fq.gz
LNCaP_LLC0150_24hr_rep1_R2.fq.gz
LNCaP_LLC0150_24hr_rep2_R2.fq.gz

ChIPSeq:
LNCaP_AR_NSD2KO_R1.fq.gz
LNCaP_AR_NSD2WT_R1.fq.gz
LNCaP_FOXA1_NSD2KO_R1.fastq.gz
LNCaP_FOXA1_NSD2WT_R1.fastq.gz
LNCaP_H3k27Ac_NSD2KO_R1.fq.gz
LNCaP_H3k27Ac_NSD2WT_R1.fq.gz
LNCaP_AR_DMSO_R1.fastq.gz
LNCaP_AR_LLC0150_R1.fastq.gz
LNCaP_FOXA1_DMSO_R1.fastq.gz
LNCaP_FOXA1_LLC0150_R1.fastq.gz
LNCaP_H3K27Ac_DMSO_R1.fastq.gz
LNCaP_H3K27Ac_LLC0150_R1.fastq.gz
LNCaP_AR_NSD2KO_R2.fq.gz
LNCaP_AR_NSD2WT_R2.fq.gz
LNCaP_FOXA1_NSD2KO_R2.fastq.gz
LNCaP_FOXA1_NSD2WT_R2.fastq.gz
LNCaP_H3k27Ac_NSD2KO_R2.fq.gz
LNCaP_H3k27Ac_NSD2WT_R2.fq.gz
LNCaP_AR_DMSO_R2.fastq.gz
LNCaP_AR_LLC0150_R2.fastq.gz
LNCaP_FOXA1_DMSO_R2.fastq.gz
LNCaP_FOXA1_LLC0150_R2.fastq.gz
LNCaP_H3K27Ac_DMSO_R2.fastq.gz
LNCaP_H3K27Ac_LLC0150_R2.fastq.gz
```

| Genome browser session (e.g. UCSC) | N/A |
|---|---|

## Methodology

| Replicates | Multiple biological as well as technical replicates are included. |
|---|---|
| Sequencing depth | ChIPseq: Sequenced to 50-70M total reads, paired-end mode, 125bp read lengths. Over 97% of uniquely mapped reads. RNAseq: Sequenced to 25-30M total reads, paired-end mode, 125bp read lengths. Over 97% of uniquely mapped reads. |
| Antibodies | Antibodies used for ChIP-seqs have been mentioned in the methods section and antibody list above. |
| Peak calling parameters | MACS2 (Version 2.1.1.20160309) callpeak was used for performing peak calling with the following option: 'macs2 callpeak–call-summits–verbose 3 -g hs -f BAM -n OUT–qvalue 0.05'. For H3K27ac data, the broad option was used. |
| Data quality | FastQC was used to quality check the raw sequencing data using standard metrics and default thresholds. |
| Software | Using deepTools (version 3.3.1) bamCoverage, a coverage file (bigWig format) for each sample was created. The coverage was calculated as the number of reads per bin, where bins are short consecutive counting windows. While creating the coverage file, the data was normalized with respect to each library size. ChIP peak profile plots and read-density heat maps were generated using deepTools, and cistrome overlap analyses were carried out using the ChIPpeakAnno (version 3.0.0) or ChIPseeker (version 1.29.1) packages in R (version 3.6.0). |

