## [Peer Review File · Nature Genetics]

Peer Review Information

Manuscript Title: NSD2 is a requisite subunit of the AR/FOXA1 neo-enhanceosome in promoting prostate tumorigenesis

Corresponding author name(s): Dr Irfan Asangani, Abhijit Parolia, Dr Arul Chinnaiyan

Reviewer Comments & Decisions:

Decision Letter, initial version:

11th Jan 2024

Dear Dr Chinnaiyan,

Your Article, "NSD2 is a requisite subunit of the AR/FOXA1 neo-enhanceosome in promoting prostate tumorigenesis" has now been seen by 3 referees.

Reviewer#1 has minor comments and suggests to perform the drug testing on human prostate cancer organoids.

Reviewer#2 is concerned with novelty and suggests better characterizing the NSD2-dependent AR binding sites and including pre-clinical models (testing the drug in human prostate tumors or prostate cancer patient-derived xenografts).

Reviewer#3 asks to better characterize the NSD2-dependent AR binding sites and provide more mechanism for the NSD2-mediated AR reprogramming.

You will see from their comments copied below that while they find your work of considerable potential interest, they have raised quite substantial concerns that must be addressed. In light of these comments, we cannot accept the manuscript for publication, but would be very interested in considering a revised version that addresses these serious concerns.

We hope you will find the referees' comments useful as you decide how to proceed. If you wish to submit a substantially revised manuscript, please bear in mind that we will be reluctant to approach the referees again in the absence of major revisions.

To guide the scope of the revisions, the editors discuss the referee reports in detail within the team, including with the chief editor, with a view to identifying key priorities that should be addressed in

revision and sometimes overruling referee requests that are deemed beyond the scope of the current study. In this case, we invite you to address Reviewers' comments in full. We hope that you will find the prioritised set of referee points to be useful when revising your study. Please do not hesitate to get in touch if you would like to discuss these issues further.

If you choose to revise your manuscript taking into account all reviewer and editor comments, please highlight all changes in the manuscript text file. At this stage we will need you to upload a copy of the manuscript in MS Word .docx or similar editable format.

*2) If you have not done so already please begin to revise your manuscript so that it conforms to our Article format instructions, available here. Refer also to any guidelines provided in this letter.

Please be aware of our guidelines on digital image standards.

[redacted]

If you wish to submit a suitably revised manuscript we would hope to receive it within 6 months. If you cannot send it within this time, please let us know. We will be happy to consider your revision so long as nothing similar has been accepted for publication at Nature Genetics or published elsewhere. Should your manuscript be substantially delayed without notifying us in advance and your article is eventually published, the received date would be that of the revised, not the original, version.

Nature Genetics is committed to improving transparency in authorship. As part of our efforts in this direction, we are now requesting that all authors identified as 'corresponding author' on published papers create and link their Open Researcher and Contributor Identifier (ORCID) with their account on the Manuscript Tracking System (MTS), prior to acceptance. ORCID helps the scientific community achieve unambiguous attribution of all scholarly contributions. You can create and link your ORCID from the home page of the MTS by clicking on 'Modify my Springer Nature account'. For more information please visit please visit www.springernature.com/orcid.

Thank you for the opportunity to review your work.

Sincerely,
Chiara

Chiara Anania, PhD
Associate Editor
Nature Genetics
<https://orcid.org/0000-0003-1549-4157>

Referee expertise:

Referee #1: prostate cancer biology

Referee #2: genomics, cancer

Referee #3: cancer epigenetics

Reviewers' Comments:

Reviewer #1:

Remarks to the Author:

The manuscript by Parolia et al introduces the role of Nuclear Receptor Binding SET Domain Protein 2 (NSD2) as a co-activator of AR transcriptional activity in prostate cancer. NSD2 is a histone 3 lysine 36 di-methyltransferase that has been shown previously to be expressed at high levels in advanced prostate cancer where it promotes metastasis. The current study extends the importance of NSD2 in prostate cancer by showing that it is required for AR transcriptional program specifically in prostate cancer. Additionally, the current study describes a novel PROTAC that degrades NSD2 and reduces prostate cancer cell viability. While the current "first generation" drug is not viable in vivo (according to the authors), this still represents a major advance since NSD2 (and related) proteins have to be very difficult to target.

Overall the data are of excellent quality and novelty, and very comprehensive. Highlights include:
The Cherry reporter assay is very elegant
The relationship to PRC2 complex (somewhat unexpected) is important observation
Similarly, the data on NSD1 and PRC2 complexes are likely to be relevant beyond prostate cancer.
Elegant series of studies to look at the Long and Short forms of NSD2 that show the need for its catalytic activity.

This reviewer has only minor comments, which should be addressable by text revisions:

The model is very helpful and very clear in terms of the promoter sequences. But what would be helpful is to explain (ideally as a second part of the model and/or in the text) how/whether the authors envision the functions of NSD2 changing over cancer progression, particularly, as a consequence of changes in AR function in Primary and CRPC (which they show expression data for) and speculate about mets (which others have shown data for).

Along the same lines - what about the role of NSD2 in hematological versus prostate cancer, considering that the former are not AR dependent and in which a mutant form is widely expressed (which is not the case in prostate cancer). I feel this is needed given the point of the authors below: Next, we characterized the cytotoxic effect of LLC0150 in a panel of over 110 human-derived normal and cancer cell lines originating from 22 different lineages (Table S3). As expected, hematologic cancers harboring the activating NSD2 E1099K mutation emerged as the most sensitive to treatment with LLC0150 (IC50 ranging from 0.274 - 69.68 nM), which was immediately followed by all the tested AR-positive prostate cancer cell lines (shown in red, Fig. 4i).

It would be advantageous to explain why they perform the in vivo analyses in 22RV1 rather than LNCaP cells.

This reviewer understands that the drug is not viable in vivo - but what about in human prostate cancer organoids (a nice but not necessary addition). Additionally what about the NSD2 marks the drug-treated prostate cancer cells?

Given that the drug is not viable in vivo - is it possible to speculate on alternatives options or strategies?

Reviewer #2:

Remarks to the Author:

In this study, the authors conducted a comprehensive CRISPR screen targeting transcriptional cofactors to identify stimulators of KLK3/PSA expression and coactivators of androgen receptor (AR), uncovering NSD2 as a novel coactivator in the AR transcriptional complex. They demonstrate that NSD2 upregulation in prostate cancer broadly extends AR genomic binding sites, contributing significantly to AR cistrome reprogramming observed in human prostate cancer development. Furthermore, the authors developed the PROTAC degrader compound LLC0150, targeting both NSD1 and NSD2. This compound exhibited selective cytotoxicity against AR/FOXA1-dependent prostate cancer cell lines, suggesting its potential as a therapeutic agent.

Given the established role of AR as a critical transcription factor in the development and progression of prostate cancer, and the notable reprogramming of its cistromes in human prostate tumorigenesis and metastasis, the identification of NSD2 in this study as a key contributor to AR cistrome reprogramming is intriguing. This finding underscores the potential of targeting NSD2 in prostate cancer therapy. However, despite the unbiased screening and validation of NSD2/1-AR interactions, which potentially explain the dependence on AR cistrome reprogramming in tumors, the novelty of these observations is somewhat compromised. This is due to prior research, specifically the study by Kang HB et al (FEBS

Lett. 2009; 583(12):1880-6. PMID: 19481544), which already identified the physical interaction between NSD2 and AR, thereby influencing AR-mediated transcriptional control. Nevertheless, to enhance the overall quality and impact of this study, this referee recommends addressing the following comments for improvement and a clearer demonstration of its unique contributions from current work.

Major comments:

1. Inconsistencies for the Influence of NSD2 on KLK3 and H3K36me2: In Figure 1d, alongside protein levels determined by Western blotting, the transcriptional regulation of NSD2 on KLK3 should be considered. There seems to be a discrepancy in KLK3 protein expression in the NSD2 KO experiments between Figure 1f and 1h. Furthermore, the impact of NSD2 on H3K36me2 levels appears inconsistent: slight influence upon NSD2 knockdown (Figure 1d) or KO (Figure 1f), versus notable decrease upon acute NSD2 degradation (Figure 3f). The effects of NSD2-L KO, particularly on AR target genes like KLK3 and FKBP5, suggest a more significant role than stated. Clarification through multiple independent assays would be beneficial to explain these inconsistencies.
2. Sustainability of siRNA knockdown efficiency: The method maintaining siRNA knockdown efficiency for extended periods (up to 10, 15, 20, or 34 days as shown in Figure 1d and Figure 3a) is not clear, given the typically transient nature of siRNA transfection. Further technical details or alternative explanations for these long-lasting effects are needed.
3. NSD2 expression in prostate tissues and correlation with AR signaling: The undetectable expression of NSD2 in normal and benign prostates, as reported in Figure 1j/k, raises questions about the sample size used for this analysis. Additionally, considering the reported strong association of NSD2 with AR, exploring the correlation between NSD2 expression and AR signaling scores would provide deeper insights.
4. Chromatin looping and correlation of NSD2 with epigenetic marked genomic regions: The observation of NSD2-dependent AR binding sites enriched at the enhancer elements prompts the question of altered chromatin looping between these reprogrammed AR enhancer sites and target driver genes (Figure 2b). It is also crucial to investigate how these NSD2-dependent AR sites correlate with NSD2 chromatin associations and the enrichment of histone marks like H3K36me2, H3K27ac, or H3K27me3. Performing appropriate NSD2 and histone mark ChIP-seq assays would substantiate these findings.
5. Pre-clinical validation of NSD1/2 PROTAC degrader: The study reports the effects of the NSD1/2 PROTAC degrader solely in cell lines. To validate these findings, it is imperative to test them in pre-clinical models, such as human prostate tumors or prostate cancer patient-derived xenografts.

Reviewer #3:

Remarks to the Author:

Parolia et al., identify NSD2 as a mediator of androgen receptor reprogramming and oncogenic phenotypes in prostate cancer. The primary findings are:

- AR reprogramming is dependent the NSD2 gene, an H3K36 di-methyltransferase. NSD2 acts as a cofactor to reprogram AR away from canonical AR binding sites to chimeric motifs, such as FOXA1:AR

half-sites. This observation was validated in cistromes generated in primary tissues from a previously published study.

- NSD2 expands the AR cistrome by about 40,000 sites.
- NSD2 suppression impacts numerous cellular oncogenic phenotypes.
- NSD1 can partially compensate for NSD2 loss.
- Protac degradation of NSD1 and NSD2 was cytotoxic in AR+ cell lines as well as in an enzalutamide-resistant cell lines.

Overall, this comprehensive study clearly implicates the role of NSD2 in prostate tumor biology and progression. The manuscript describes an array of genetic, epigenetic, and biochemical studies that show how NSD2 and AR interact to drive an oncogenic program. A strength of the manuscript includes the numerous and orthogonal assays that support NSD2 as an AR co-activator. Novel findings and advances include NSD2 as a key factor driving AR reprogramming, NSD1 upregulation in response to NSD2 suppression, and development of a PROTAC degrader that targets both NSD1 and NSD2 and results in cytotoxicity in AR-driven prostate cancer. Primary concerns about the manuscript include lack of a deeper biologic understanding of the NSD2-dependent AR sites as well as a lack of emphasis on the mechanism driving NSD2-mediated AR reprogramming.

Major points:

1) The observation that NSD2 activity is correlated with AR reprogramming is fascinating. I kept wanting to learn more about this set of 41.5K NSD2-dependent sites. The authors initially focus on motifs and this is an important finding; however, what are these sites and what are they doing? Exploring this further would add important biologic context to the narrative. Some suggestions are as follows:

- a) Can the downregulated genes in the NSD2 KO (figure 1g) be connected/correlated with the decommissioned AR sites (e.g., are they nearby)?
- b) What is the GREAT analysis for the 41.5K NSD2-dependent sites?
<http://great.stanford.edu/public/html/>
- c) Performing NSD2 ChIP-seq in the parental cell lines would be interesting– does it overlap with the NSD2-dependent AR sites?
- d) Do these sites overlap with T-ARBS and met-ARBS (Nat Genet. 2020 Aug;52(8):790-799.)

2) What is the mechanism of action of NSD2-mediated AR reprogramming? I believe it is important to articulate and investigate the possible mechanistic hypotheses

- a) Is it the enzymatic action of NSD2 on H3K36me2 levels at cis-regulatory elements that are responsible for AR reprogramming? The authors point out the mild global decrease in H3K36me2 in NSD2 KO cells. It would be interesting to perform both H3K36me2 ChIP-seq in wild-type and NSD2 KO cells. Do the NSD2-dependent AR sites also lose H3K36me2 relative to the unchanged sites? Discuss/tie the biology of H3K36me2 to regulatory elements (see Mol Cell. 2023 Jul 20;83(14):2398-2416.e12.)
- b) Several manuscripts have shown that 'chromatin-modifying' enzymes also act on non-histone substrates. For example, could the mechanism underlying the AR reprogramming be that NSD2 is responsible for post-translationally modifying the AR? As a recent example in prostate cancer, a publication described FOXA1 as a demethylation target for LSD1 (Nat Genet. 2020 Oct;52(10):1011-1017.).

3) The authors state that the NSD2-KO cells displayed striking loss of defining neoplastic features, but remained viable. What did the cells look like? Was there transcriptional or epigenetic evidence of re-differentiation to a more 'normal' state? What biological programs correlated with the 8,471 gained sites in the NSD2-KO condition?

4) Where did NSD1 rank in the original CRISPR screen? NSD1 appears to also have profound transcriptional effects on androgen response and MYC pathways (fig 3i). Moreover, NSD1 has been shown to act as an AR coregulator (J Biol Chem. 2001 Nov 2;276(44):40417-23.). Do both NSD1 and NSD2 increase during prostate tumorigenesis and progression? Is AR reprogrammed after NSD1 KO?

5) The following manuscript was inadvertently not cited (FEBS Lett. 2009 Jun 18;583(12):1880-6.). This manuscript demonstrated NSD2 interaction via its HMG domain with the AR DBD (similar to fig 3g/h in author's paper). Moreover, this paper showed that NSD2 had a role in AR-mediated transcription and posited its role in prostate carcinogenesis.

Minor points:

1) It would be helpful to understand why certain cell lines were selected for the various different experiments; for example, did they not give consistent results? Were they selected because they had certain/appropriate properties for a particular assay?

2) Please check p-values in figure 1d (I think that it should be "e-" and not "e+")?

3) Fig 2i and 2j – please define "AR super-enhancer" – how is it different than a super-enhancer? Fig 2j refers to transcriptional activity in the text, but I think that the plot is refers to chromatin signal?

4) The antibody used for AR ChIP-seq (ab133273) is not listed as ChIP-grade. Can the authors provide rationale as to why this antibody was selected. I realize that this is a picky point, but I raise it for issues of reproducibility and consistency in the field.

Author Rebuttal to Initial comments

NG-A63836: "NSD2 is a requisite subunit of the AR/FOXA1 neo-enhanceosome in promoting prostate tumorigenesis"

Responses to Reviewers' Comments

Reviewer #1:

Remarks to the Author:

The manuscript by Parolia et al. introduces the role of Nuclear Receptor Binding SET Domain Protein 2 (NSD2) as a co-activator of AR transcriptional activity in prostate cancer. NSD2 is a histone 3 lysine 36 dimethyltransferase that has been shown previously to be expressed at high levels in advanced prostate cancer where it promotes metastasis. The current study extends the importance of NSD2 in prostate cancer by showing that it is required for AR transcriptional program specifically in prostate cancer.

Additionally, the current study describes a novel PROTAC that degrades NSD2 and reduces prostate cancer cell viability. While the current “first generation” drug is not viable *in vivo* (according to the authors), this still represents a major advance since NSD2 (and related) proteins have to be very difficult to target.

Overall the data are of excellent quality and novelty, and very comprehensive. Highlights include:

The Cherry reporter assay is very elegant.

The relationship to PRC2 complex (somewhat unexpected) is important observation.

Similarly, the data on NSD1 and PRC2 complexes are likely to be relevant beyond prostate cancer.

Elegant series of studies to look at the Long and Short forms of NSD2 that show the need for its catalytic activity.

We thank the reviewer for highlighting the novelty and importance of our findings. We are grateful for the meaningful suggestions which have considerably improved our manuscript. All additions and changes made to the manuscript are outlined below in our point-by-point responses. In the rebuttal figure legends, we have also noted the manuscript figures in which new data have been incorporated after revision, with the associated text highlighted in blue.

This reviewer has only minor comments, which should be addressable by text revisions:

The model is very helpful and very clear in terms of the promoter sequences, but what would be helpful is to explain (ideally as a second part of the model and/or in the text) how/whether the authors envision the functions of NSD2 changing over cancer progression, particularly, as a consequence of changes in AR function in Primary and CRPC (which they show expression data for) and speculate about mets (which others have shown data for).

Response: We thank the reviewer for this suggestion. We have now revised the discussion to outline distinct functions of NSD2 in prostate cancer initiation and metastatic progression. Briefly, as shown earlier, we propose abnormally expressed NSD2 to promote prostate tumorigenesis by collaborating with driver transcription factor oncogenes, namely FOXA1 and HOXB13, in redistributing the AR enhanceosome to bind non-physiological, chimeric AR-half elements, thereby transcriptionally activating cancerous phenotypes. In the advanced disease, as shown here and reported previously¹⁻³, NSD2 (along with NSD1)

could additionally function to restrict the spread of the repressive H3K27me3 mark by the PRC2/EZH2 complex, thereby enabling gene programs associated with disease aggressiveness and progression.

Supporting this model, in metastatic prostate cancer, EZH2 activity is markedly amplified^{1,4}, and NSD1 and NSD2 knockdown in mCRPC cell lines leads to marked accumulation of H3K27me3 on the chromatin (Fig. 3j,l). We coalesce all this data to propose a model wherein the function of NSD2 evolves from enabling oncogenic AR activity in primary AR-dependent prostate cancer to additionally counterbalancing the canonical repressive PRC2 activity in the metastatic castration-resistant disease.

Along the same lines - what about the role of NSD2 in hematological versus prostate cancer, considering that the former are not AR-dependent and in which a mutant form is widely expressed (which is not the case in prostate cancer). I feel this is needed given the point of the authors below:

Next, we characterized the cytotoxic effect of LLC0150 in a panel of over 110 human-derived normal and cancer cell lines originating from 22 different lineages (Table S3). As expected, hematologic cancers harboring the activating NSD2 E1099K mutation emerged as the most sensitive to treatment with LLC0150 (IC50 ranging from 0.274 - 69.68 nM), which was immediately followed by all the tested AR-positive prostate cancer cell lines (shown in red, Fig. 4i).

Response: We thank the reviewer for this suggestion. In the introduction and discussion sections, we have now described in detail the recurrence of NSD2 alterations in hematological cancers and summarized its well-characterized oncogenic roles. NSD2 is a *bona fide* oncogene in multiple myeloma and childhood acute lymphoblastic leukemia, which harbor recurrent activating alterations in the NSD2 gene⁵⁻⁹. Hotspot mutations in the catalytic SET domain (e.g. E1099K) are significantly enriched in the relapsed disease, which leads to a global increase in H3K36me2 levels and oncogenic remodeling of the epigenome through direct antagonism of the repressive PRC2/EZH2 machinery^{3,10-12}. The structural basis for this antagonism was recently identified in di/tri-methylation at H3K36 allosterically hindering the loading of H3K27 residue into the catalytic pocket of the EZH2 enzyme¹³. Accordingly, several medicinal chemistry and pharmacology campaigns have focused on developing potent NSD2 therapeutics, with a first-in-class NSD2 inhibitor recently entering clinical trials in multiple myeloma patients (<https://classic.clinicaltrials.gov/ct2/show/NCT05651932>). Thus, as also alluded to by the reviewer, we included NSD2-altered cancer cell lines to serve as positive controls in our pan-cancer screen using the tool NSD1/2 PROTAC compound.

It would be advantageous to explain why they perform the *in vivo* analyses in 22RV1 rather than LNCaP cells.

Response: 22RV1 cells were chosen for *in vivo* experiments due to their higher take rate (almost 100%) as well as faster growth kinetics as xenografts. Thus, the 22RV1 model allowed us to cleanly assess the effect of NSD2 activity on the subcutaneous grafting ability of prostate cancer cells without other confounding technical variables.

This reviewer understands that the drug is not viable *in vivo* - but what about in human prostate cancer organoids (a nice but not necessary addition). Additionally what about the NSD2 marks the drug-treated prostate cancer cells?

Response: We thank the reviewer for this important suggestion. We have now profiled the efficacy of our first-generation NSD1/2 PROTAC, LLC0150, in a panel of patient-derived organoid models of AR-positive castration-resistant prostate cancer (CRPC). Notably, all the CRPC organoids robustly expressed the NSD2 protein (**Fig. R1a**), and treatment with LLC0150 triggered NSD2 degradation in a dose-dependent manner (**Fig. R1b**). This was accompanied by a parallel loss in the NSD1/2-catalyzed H3K36me2 mark in these cancer cells. Notably, NSD1/2 inactivation led to a marked decrease in the expression of KLK3 (a direct AR target), which was followed by a significant inhibition of growth in all CRPC organoid models (**Fig. R1c,d**). Our panel comprises the LuCaP 35, LuCaP 23.1, CR-146-12, PC310, and LuCaP 86.2 lines that have been widely used as AR-dependent disease models.

Figure R1: **a)** Representative images of NSD2 IHC in a panel of patient-derived xenografts. **b)** Immunoblot of NSD2, AR targets, and histone marks in the LuCaP 23.1 PDX-derived organoid lines treated with LLC0150, the NSD1/2 dual degrader. Total H3 is the loading control. **c)** Barplots showing relative viability in the panel of PDX-derived organoids treated with two doses of LLC0150. In all lines, degradation of NSD1/2 reduces cell viability. **d)** Growth curves (Cell-titer Glo) of two representative AR/NSD2-positive PDX-derived organoid lines treated with DMSO or LLC0150.

This new data is added in Figure S10a-d of the revised manuscript.

Given that the drug is not viable *in vivo* - is it possible to speculate on alternatives options or strategies?

Response: Following rigorous medicinal chemistry structure-activity assessments, we are currently modifying the first-generation LLC0150 compound to improve its drug-like properties as well as augment oral bioavailability. In a separate effort focused on NSD2 (aka MMSET) translocation-positive t(4;14) multiple myeloma, which often fully or partially truncate the PWWP1 domain of the NSD2 protein, we are re-purposing PWWP2 and SET domain binding NSD2 warheads to make bi-functional PROTACs for therapeutic applications. Hopefully, these efforts can yield alternative compounds that can be comprehensively tested for safety and efficacy in preclinical *in vivo* models.

However, as a proof-of-concept, we performed direct intratumoral injection of LLC0150 in mice bearing VCaP xenograft tumors (**Fig. R2a**). Here, consistent with our *in vitro* cell line and organoid data,

we found LLC0150 to trigger marked degradation of NSD1 and NSD2 in cancer cells, which resulted in a significant decrease in expression of the proliferative marker Ki-67 (Fig. R2b). This was accompanied by an increase in TUNEL (fragmented DNA) staining signifying apoptotic cell death. These results were confirmed in immunoblots from matched tumor lysates, where we detected an evident decrease in NSD1/2 protein levels and accumulation of the apoptotic marker cleaved-PARP in the LLC0150-treated tumors (Fig. R2c).

Figure R2: **a)** Schematic overview of the LLC0150 intra-tumoral injection study in a VCaP xenograft model. 100 μl of the drug resuspended at 40 mg/ml concentration was injected per tumor. **b)** Top panels: Representative NSD2 and Ki67 IHC images in the vehicle and LLC0150 treated tumors. Loss of NSD2 correlates with a reduction in proliferating Ki67+ cells. Bottom panels: TUNEL assay in the vehicle and LLC0150 treated tumors reveal a high number of apoptotic (TUNEL+) cells in the drug-treated tumors. **c)** Immunoblot of labeled proteins in the matched tumor lysates from b. Total H3 is used as a loading control.

This new data is added as Figure S10e-g in the revised manuscript.

Reviewer #2:

Remarks to the Author:

In this study, the authors conducted a comprehensive CRISPR screen targeting transcriptional cofactors to identify stimulators of KLK3/PSA expression and coactivators of androgen receptor (AR), uncovering NSD2 as a novel coactivator in the AR transcriptional complex. They demonstrate that NSD2 upregulation in prostate cancer broadly extends AR genomic binding sites, contributing significantly to AR cistrome reprogramming observed in human prostate cancer development. Furthermore, the authors developed the PROTAC degrader compound LLC0150, targeting both NSD1 and NSD2. This compound exhibited selective cytotoxicity against AR/FOXA1-dependent prostate cancer cell lines, suggesting its potential as a therapeutic agent.

Given the established role of AR as a critical transcription factor in the development and progression of prostate cancer, and the notable reprogramming of its cistromes in human prostate tumorigenesis and metastasis, the identification of NSD2 in this study as a key contributor to AR cistrome reprogramming is intriguing. This finding underscores the potential of targeting NSD2 in prostate cancer therapy. However, despite the unbiased screening and validation of NSD2/1-AR interactions, which potentially explain the dependence on AR cistrome reprogramming in tumors, the novelty of these observations is somewhat compromised. This is due to prior research, specifically the study by Kang HB et al (FEBS Lett. 2009; 583(12):1880-6. PMID: 19481544), which already identified the physical interaction between NSD2 and AR, thereby influencing AR-mediated transcriptional control. Nevertheless, to enhance the overall quality and impact of this study, this referee recommends addressing the following comments for improvement and a clearer demonstration of its unique contributions from current work.

Response: We thank the reviewer for highlighting the comprehensive and novel aspects of our study as well as for the detailed feedback. Their suggestions have considerably improved the quality of our work. All additions and changes made to the manuscript are outlined below in our point-by-point responses. In the rebuttal figure legends, we have also noted the manuscript figures in which new data have been incorporated after revision, with the associated text highlighted in blue.

Major comments:

1. Inconsistencies for the Influence of NSD2 on KLK3 and H3K36me2: In Figure 1d, alongside protein levels determined by Western blotting, the transcriptional regulation of NSD2 on KLK3 should be considered.

There seems to be a discrepancy in KLK3 protein expression in the NSD2 KO experiments between Figure 1f and 1h. Furthermore, the impact of NSD2 on H3K36me2 levels appears inconsistent: slight influence upon NSD2 knockdown (Figure 1d) or KO (Figure 1f), versus notable decrease upon acute NSD2 degradation (Figure 3f). The effects of NSD2-L KO, particularly on AR target genes like KLK3 and FKBP5, suggest a more significant role than stated.

Clarification through multiple independent assays would be beneficial to explain these inconsistencies.

Response: We thank the reviewer for seeking these important clarifications. As suggested, to corroborate our protein blots, we have now added quantification of the *KLK3* transcript from the experiments summarized in **Fig. 1f** (CRISPR knockout of NSD2 isoforms) and **Fig. 1h** (DHT stimulation of NSD2 wildtype and deficient LNCaP cells grown in steroid-deprived serum). As shown in the data figures below, the *KLK3* mRNA changes are consistent with protein level data, where the loss of NSD2 results in a decrease in the *KLK3* transcript (Fig R3a). Compellingly, treatment with 10nM DHT (androgen) after 48 hours of culture in steroid-deprived, serum-containing media, did not stimulate the expression of the *KLK3* transcript in the NSD2-deficient LNCaP cells (**Fig. R3b**). Thus, the slight discrepancy in the *KLK3* protein blots could be due to the differences in experimental conditions used in **Fig. 1f** and **Fig. 1h** where, respectively, prostate cancer cells were grown in full serum-supplemented or charcoal-stripped serum (steroid-depleted) growth mediums.

In prostate cancer cells, we see a consistent, but partial, decrease in the H3K36me2 levels upon the inactivation of NSD2. As noted by the reviewer, the loss is more prominently seen in the chromatin fractions (**Fig. 3f**) as compared to whole cell lysates (**Fig. 1d,f**). To address this comment, we have now repeated H3K36me2 immunoblotting in whole-cell lysates as well as the chromatin fractions prepared from the LNCaP NSD2-KO + NSD2-L-dTAG13 cell lines (**Fig. R3c,d**). Here, acute degradation of NSD2 leads to a notable, yet partial, loss in H3K36me2 in both fractions. We have added these blots to the revised manuscript.

Figure R3: **a)** Expression (qRT-PCR) of *AR* and *KLK3* in the NSD2 WT and KO cells. *HPRT1* is used as an internal loading control. **b)** Expression (qRT-PCR) of *KLK3* in the LNCaP NSD2 WT and KO cells with time-dependent DHT stimulation. *HPRT1* is used as an internal loading control. **c)** Immunoblot of labeled proteins in whole-cell lysates from LNCaP NSD2-FKBP12-F36V engineered cell lines treated with dTAG13. **d)** Immunoblot of labeled proteins in the chromatin fractions from LNCaP NSD2-FKBP12-F36V engineered cell lines treated with dTAG13. GAPDH and H3 are used as loading controls.

This new data is added as Figures S2c and 3f in the revised manuscript.

2. Sustainability of siRNA knockdown efficiency: The method maintaining siRNA knockdown efficiency for extended periods (up to 10, 15, 20, or 34 days as shown in Figure 1d and Figure 3a) is not clear, given the

typically transient nature of siRNA transfection. Further technical details or alternative explanations for these long-lasting effects are needed.

Response: We thank the reviewer for this question. We have included a detailed description of how the siRNA experiment was performed to sustain target gene knockdown for extended periods in the methods section. Briefly, in our experimental design, siRNAs (25 nM) were replenished via lipid transfection every 5 days after the initial dosing at day 0 and day 1. At each transfection timepoint, total protein was extracted and probed for NSD2 protein expression. As shown in **Fig. S5a**, no detectable NSD2 protein was seen on days 5, 10, or 15 after the primary siRNA transfection. This confirms sustained loss of NSD2 in the cancer cells used in our growth assays over extended periods.

Using an orthogonal technique, we have also now generated stable LNCaP cells that express a doxycycline-inducible shRNA targeting NSD2 (**Fig. R4a**). Consistent with previous findings, sustained loss of NSD2 achieved by culturing the LNCaP cells in doxycycline-supplemented media resulted in a significant attenuation of the growth (**Fig. R4b**). This new data has now been added in **Fig. S5b** and discussed in the revised manuscript.

Figure R4: **a)** Immunoblot of NSD2 isoforms in two independent stable LNCaP doxycycline-inducible shNSD2 lines plus/minus doxycycline (1 µg/ml). Total H3 is used as a loading control. **b)** Growth curves (Cell-titer Glo) of the LNCaP shNSD2 lines treated with doxycycline for 30 days.

This new data is added as Figure S5b in the revised manuscript.

3. NSD2 expression in prostate tissues and correlation with AR signaling: The undetectable expression of NSD2 in normal and benign prostates, as reported in Figure 1j/k, raises questions about the sample size used for this analysis.

Response: We thank the reviewer for this suggestion. We have increased the number of benign as well as cancerous prostate tissues used in our immunofluorescence assessment of NSD2 expression. In ten additional benign prostate tissues, we detect no expression of NSD2 in the luminal epithelial cells (**Fig. R5a,b**). More strikingly, in five new prostatectomy tissues, while the malignant cytokeratin 8-positive (CK-8+) epithelial cells had robust expression of NSD2, the adjacent benign epithelial cells in the same specimen had no detectable levels of NSD2 (**Fig. R5c**). Overall, our immunofluorescence analysis was carried out in nine prostatectomy (notably enabling patient-matched benign-to-cancer comparisons), four benign prostate, four localized prostate cancer, and eight metastatic CRPC tissues.

Figure R5: **a)** Representative immunofluorescence (IF) images of NSD2 in benign prostate, primary prostate cancer, and metastatic CRPC tissue microarray. **b)** Integrated optical density quantification of NSD2 IF staining in benign (n=10), primary PCa (n=10), and mCRPC (n=5) tissues. **c)** Representative multiplex IF images of NSD2 and CK-8 in adjacent benign and primary prostate cancer lesions in patient prostatectomies (n=5).

This new data is added as Figure S3a-c in the revised manuscript.

Additionally, considering the reported strong association of NSD2 with AR, exploring the correlation between NSD2 expression and AR signaling scores would provide deeper insights.

Response: Following the reviewer's suggestion, we have now carried out correlation analyses between NSD2 and AR transcriptional programs in patient tumors. We first defined a high-confidence NSD2 gene signature (comprising 194 genes, listed in **Supplementary Table S4**) from our isogenic cell line models

that, reassuringly, strongly correlated with NSD2 expression in the primary prostate tumors (Fig. R6a; TCGA cohort, n=554 primary tumors). Notably, AR transcriptional activity (assessed using the hallmark gene signature) showed a strong and significant positive correlation with the expression of NSD2-activated genes in patient tumors (Fig. R6b; $R=0.68$, $p=2.2e-16$). In light of our mechanistic findings, these insights further corroborate a regulatory connection between NSD2 and AR activities in prostate tumorigenesis.

Figure R6: a) Correlation plots showing the NSD2 transcript expression and experimentally defined “NSD2 activity score” in the primary prostate cancer tumors from the TCGA cohort (n=554 tumors). The NSD2 activity score was defined in-house from our NSD2 CRISPR knockout cell lines. **b)** Correlation plots showing NSD2 activity score and the widely used hallmark AR activity score in primary PCa tumors. Both panels show a statistically significant and high positive R-value, indicating positive correlations.

This new data is added as Figure S2b in the revised manuscript.

4. Chromatin looping and correlation of NSD2 with epigenetic marked genomic regions: The observation of NSD2-dependent AR binding sites enriched at the enhancer elements prompts the question of altered chromatin looping between these reprogrammed AR enhancer sites and target driver genes (Figure 2b).

It is also crucial to investigate how these NSD2-dependent AR sites correlate with NSD2 chromatin associations and the enrichment of histone marks like H3K36me2, H3K27ac, or H3K27me3.

Performing appropriate NSD2 and histone mark ChIP-seq assays would substantiate these findings.

Response: We thank the reviewer for these suggestions. Our group and others have not been successful at generating high-quality ChIP-seq data for NSD2 using the conventional crosslinking approach largely due to the lack of a ChIP-grade antibody. We have also tried the enzyme-based CUT&RUN/TAG approaches, using both native and mildly crosslinked chromatin, but were unsuccessful in mapping the NSD2 cistrome in prostate cancer cells. Accordingly, to our knowledge, there is no published NSD2 ChIP-seq data generated from human cancer models, including hematological malignancies where NSD2 is widely studied.

However, as requested, we have now carried out ChIP-seq for five well-characterized histone marks (H3K36me2, H3K4me1, H3K4me2, H3K27ac, and H3K27me3) to better profile the chromatin state at the NSD2-dependent AR enhancer sites. In prostate cancer cells, we found NSD2-dependent AR elements to be significantly enriched for the active H3K36me2 and enhancer-associated H3K4me1/2 and H3K27ac histone modifications as compared to the NSD2-independent AR sites (**Fig. R7**). In contrast, the NSD2-dependent AR enhancers were depleted of the repressive H3K27me3 histone mark, which is catalyzed by the PRC2 polycomb complex. Therefore, nucleosomes at the NSD2-dependent AR binding elements are maintained in an active chemical state, thus substantiating their *cis*-regulatory activation in prostate cancer cells.

Figure R7: **a)** Boxplots of normalized ChIP-seq reads of distinct activating and repressive histone modifications at NSD2-dependent and NSD2-independent AR sites in VCaP cells. **b)** ChIP-seq read-density tracks of H3K36me2 and H3K27ac within a Chr10 locus in NSD2 WT and KO LNCaP cell lines. NSD2-dependent and independent AR sites are marked and highlighted.

This new data is added as Figure S3d in the revised manuscript.

5. Pre-clinical validation of NSD1/2 PROTAC degrader: The study reports the effects of the NSD1/2 PROTAC degrader solely in cell lines. To validate these findings, it is imperative to test them in pre-clinical models, such as human prostate tumors or prostate cancer patient-derived xenografts.

Response: We thank the reviewer for this comment. As noted in the manuscript, our first-generation NSD1/2 PROTAC has poor pharmacokinetic properties for *in vivo* application. While we are actively working to improve the drug-like properties, as proof-of-concept we carried out the *in vitro* cytotoxicity assessment of our PROTAC in over 120 normal and cancer cell lines, which revealed selective cytotoxicity in AR-dependent prostate cancer models. Given the reviewer's comment, we have now additionally profiled the efficacy of our NSD1/2 PROTAC in a panel of patient-derived organoid models of castration-resistant prostate cancer (CRPC) that remain dependent on AR signaling.

Notably, all AR-positive CRPC organoids robustly expressed the NSD2 protein (**Fig. R8a**) and treatment with LLC0150 triggered NSD2 degradation and a decrease in the H3K36me2 histone mark in a dose-dependent manner (**Fig. R8b**). Consistent with previous findings, NSD1/2 co-targeting led to a marked decrease in the expression of KLK3 (a direct AR target), which was followed by a significant inhibition of growth in all five CRPC organoid models (**Fig. R8b-d**), namely LuCaP 35, LuCaP 23.1, CR-146-12, PC310, and LuCaP 86.2. In future studies, using more drug-like degraders of NSD1/2, we hope to reproduce these findings in pre-clinical animal models of CRPC.

Figure R8: **a)** Representative images of NSD2 IHC in a panel of patient-derived xenografts. **b)** Immunoblot of NSD2, AR targets, and histone marks in the LuCaP 23.1 PDX-derived organoid lines treated with LLC0150, the NSD1/2 degrader. Total H3 is the loading control. **c)** Barplots showing relative viability of the PDX-derived organoids treated with two doses of LLC0150. In all lines, degradation of NSD1/2 reduces cell viability. **d)** Growth curves (Cell-titer Glo) of two representative AR/NSD2-positive PDX-derived organoid lines treated with DMSO or LLC0150.

This new data is added in Figure S10a-d of the revised manuscript.

As mentioned earlier, the LLC0150 compound does not have favorable pharmacokinetic properties for *in vivo* administration via oral, intravenous, or intraperitoneal routes. Thus, only for proof-of-concept, we performed direct intratumoral injection of LLC0150 in mice bearing VCaP xenograft tumors (**Fig. R9a**). Here, consistent with our *in vitro* data, we found LLC0150 to trigger marked degradation of NSD1 and NSD2 in cancer cells, which resulted in a significant decrease in expression of the proliferative marker Ki-67 (**Fig. R9b**). This was accompanied by an increase in TUNEL (fragmented DNA) staining signifying apoptotic cell death. These results were confirmed in immunoblots from matched tumor lysates, where we detected an evident decrease in NSD1/2 protein levels and accumulation of the apoptotic marker cleaved-PARP in the LLC0150-treated tumors (**Fig. R9c**).

Figure R9: **a)** Schematic overview of the LLC0150 intra-tumoral injection study in a VCaP xenograft model. 100 μl of the drug resuspended at 40 mg/ml concentration was injected per tumor. **b)** Top panels: Representative NSD2 and Ki67 IHC images in the vehicle and LLC0150 treated tumors. Loss of NSD2 correlates with a reduction in proliferating Ki67+ cells. Bottom panels: TUNEL assay in the vehicle and LLC0150 treated tumors reveal a high number of apoptotic (TUNEL+) cells in the drug-treated tumors. **c)** Immunoblot of labeled proteins in the matched tumor lysates from b. Total H3 is used as a loading control.

This new data is added as Figure S10e-g in the revised manuscript.

Reviewer #3:

Remarks to the Author:

Parolia et al., identify NSD2 as a mediator of androgen receptor reprogramming and oncogenic phenotypes in prostate cancer. The primary findings are:

- AR reprogramming is dependent the NSD2 gene, an H3K36 di-methyltransferase. NSD2 acts as a cofactor to reprogram AR away from canonical AR binding sites to chimeric motifs, such as FOXA1:AR half-sites. This observation was validated in cistromes generated in primary tissues from a previously published study.
- NSD2 expands the AR cistrome by about 40,000 sites.
- NSD2 suppression impacts numerous cellular oncogenic phenotypes.
- NSD1 can partially compensate for NSD2 loss.
- Protac degradation of NSD1 and NSD2 was cytotoxic in AR+ cell lines as well as in enzalutamide-resistant cell lines.

Overall, this comprehensive study clearly implicates the role of NSD2 in prostate tumor biology and progression. The manuscript describes an array of genetic, epigenetic, and biochemical studies that show how NSD2 and AR interact to drive an oncogenic program. A strength of the manuscript includes the numerous and orthogonal assays that support NSD2 as an AR co-activator. Novel findings and advances include NSD2 as a key factor driving AR reprogramming, NSD1 upregulation in response to NSD2 suppression, and development of a PROTAC degrader that targets both NSD1 and NSD2 and results in cytotoxicity in AR-driven prostate cancer. Primary concerns about the manuscript include lack of a deeper biologic understanding of the NSD2-dependent AR sites as well as a lack of emphasis on the mechanism driving NSD2-mediated AR reprogramming.

Response: We thank the reviewer for highlighting important findings of our study as well as providing detailed comments that have considerably improved our manuscript by adding more biological and mechanistic insights. All additions and changes made to the manuscript are outlined below in our point-by-point responses. In the rebuttal figure legends, we have also noted the manuscript figures in which new data has been incorporated after revision, with the associated text highlighted in blue.

Major points:

1) The observation that NSD2 activity is correlated with AR reprogramming is fascinating. I kept wanting to learn more about this set of 41.5K NSD2-dependent sites. The authors initially focus on motifs and this is an important finding; however, what are these sites and what are they doing? Exploring this further would add important biologic context to the narrative. Some suggestions are as follows:

a) Can the downregulated genes in the NSD2 KO (figure 1g) be connected/correlated with the decommissioned AR sites (e.g., are they nearby)?

Response: We thank the reviewer for this comment. We entered the genomic coordinates of the decommissioned AR sites into the GREAT algorithm and identified the nearby genes. We then conducted a gene set enrichment analysis (GSEA) in global transcriptomes from NSD2 KO vs WT LNCaP cells. This analysis revealed the genes located in *cis*-proximity of NSD2-dependent AR sites to be significantly down-regulated upon NSD2 inactivation (**Fig. R10a**). This suggests the down-regulated genes in the NSD2 KO LNCaP cells are at least partly activated by the decommissioned AR enhancers.

b) What is the GREAT analysis for the 41.5K NSD2-dependent sites?
<http://great.stanford.edu/public/html/>

Response: We thank the reviewer for directing us to this useful bioinformatics tool to predict the functions of *cis*-regulatory regions. We used the GREAT algorithm¹⁴ to identify genes closest to the NSD2-dependent, chimeric AR enhancers followed by enrichment analyses using well-annotated pathways for biological predictions¹⁵. As discussed above, most genes proximal to chimeric AR enhancers—enriched within tumor cistromes—are down-regulated in the NSD2-deficient LNCaP cells.

In the MSigDB hallmark database, the down-regulated gene set was significantly enriched for genes in the androgen response pathway and, while not significant, trended towards the enrichment of oncogenic KRAS, angiogenesis, and G2-M checkpoint pathways (**Fig. R10b**). Next, in ChIP enrichment analysis (ChEA), which uses published ChIP-seq data to predict likely transcriptional regulators of input genes¹⁶, we found AR and FOXA1 as top hits using cistromes identified from PCa patient tissue and cell lines. Furthermore, disease perturbation signatures revealed prostate cancer-associated genes as the top-most enriched signature in the NSD2-regulated AR genes (**Fig. R10b**). Altogether, these results suggest that NSD2-dependent chimeric AR sites transcriptionally activated genes that sustain oncogenic signaling in PCa cells.

Figure R10: **a)** Gene set enrichment analyses (GSEA) of GREAT nominated genes associated with the NSD2-dependent chimeric AR sites in NSD2 KO vs WT LNCaP cells. **b)** GREAT and Enrichr analyses of putative chimeric AR gene targets in molecular signature and biology pathway databases.

This new data is added as Figure S6a,b in the revised manuscript.

c) Performing NSD2 ChIP-seq in the parental cell lines would be interesting– does it overlap with the NSD2-dependent AR sites?

Response: We thank the reviewer for this question. We have carried out numerous attempts to generate NSD2 chromatin binding profiles in prostate cancer cells but, sadly, have had no success. There are no commercial ChIP-grade antibodies against NSD2 and, as such, to our knowledge, there are no publications of NSD2 cistrome from human cells. In our labs, we have attempted the cross-linking-based conventional as well as the native CUT&Tag/RUN approaches to perform NSD2 ChIP-seq, but none of them were successful. Thus, due to technical limitations, we are unable to address this important question.

d) Do these sites overlap with T-ARBS and met-ARBS (Nat Genet. 2020 Aug;52(8):790-799.)

Response: As requested, we overlapped the NSD2-dependent AR binding sites with primary PCa-specific (T-ARBS) or CRPC-specific (met-ARBS) AR enhancers identified from patient specimens^{17,18}. We found a striking enrichment of NSD2-dependent AR sites within tumor-specific AR cistromes, comprising over 50% of T-ARBS and 30% of met-ARBS (**Fig. R11**). Notably, this is consistent with the striking enrichment of chimeric AR-half motifs within tumor enhancer circuitries, as shown in this paper (**Fig. 2g and Fig. S4**).

Figure R11: Barplot showing the percentage of shared sites between the NSD2-dependent sites and AR binding sites in the normal, primary PCa (T-ARBS), or metastatic CRPC (met-ARBS) patient tumors.

This new data is added as Figure S4f in the revised manuscript.

2) What is the mechanism of action of NSD2-mediated AR reprogramming? I believe it is important to articulate and investigate the possible mechanistic hypotheses.

a) Is it the enzymatic action of NSD2 on H3K36me2 levels at cis-regulatory elements that are responsible for AR reprogramming? The authors point out the mild global decrease in H3K36me2 in NSD2 KO cells. It would be interesting to perform both H3K36me2 ChIP-seq in wild-type and NSD2 KO cells. Do the NSD2-dependent AR sites also lose H3K36me2 relative to the unchanged sites? Discuss/tie the biology of H3K36me2 to regulatory elements (see Mol Cell. 2023 Jul 20;83(14):2398-2416.e12.)

Response: We thank the reviewer for this meaningful suggestion. As requested, we have now carried out H3K36me2 ChIP-seq in the NSD2 wild-type and knockout LNCaP cells. First, as shown earlier, we detected much higher levels of H3K36me2 at the NSD2-dependent AR elements relative to the NSD2-independent sites in the wild-type LNCaP cells (**Fig. R12a**; $p = 5.85e-100$). Second, and more intriguingly, NSD2 inactivation led to a marked decrease in the H3K36me2 signal at the NSD2-dependent AR enhancers (**Fig. R12b**, $p = 3.21e-306$). In parallel, we performed H3K27me3 ChIP-seq, where we detected opposite trends.

There was a significantly higher abundance of the H3K27me3 mark at the NSD2-independent AR elements relative to the dependent sites (Fig. R12a; $p=1.92e-103$), and NSD2 inactivation led to a modest increase in H3K27me3 signal at the AR binding sites (Fig. R12c).

Figure R12: a) Boxplots of normalized ChIP-seq reads of NSD2-catalyzed H3K36me2 and EZH2/PRC2-catalyzed H3K27me3 histone marks at NSD2-dependent and NSD2-independent AR sites in LNCaP cells. b,c) same plots showing H3K36me2 (b) and H3K27me3 (c) ChIP-seq densities at AR sites in NSD2 KO or WT LNCaP cells. Two-sided student t-tests were performed to assess significance.

This new data is added as Figure S3e-g in the revised manuscript.

b) Several manuscripts have shown that ‘chromatin-modifying’ enzymes also act on non-histone substrates. For example, could the mechanism underlying the AR reprogramming be that NSD2 is responsible for post-translationally modifying the AR? As a recent example in prostate cancer, a publication described FOXA1 as a demethylation target for LSD1 (Nat Genet. 2020 Oct;52(10):1011-1017.).

Response: We thank the reviewer for this intriguing question. NSD2 is known to exhibit an autoinhibitory state that is relieved by binding to nucleosomes, enabling di-methylation of histone H3 at lysine 36 (H2K36me2). This requirement of nucleosomes for full activity suggests the potential absence of physiologically relevant non-histone substrates for NSD2¹⁹. Nonetheless, there have been reports that NSD2 can methylate and activate proteins such as PTEN²⁰ and STAT3²¹. However, these findings were not replicated in a subsequent study by Or Gozani’s lab²².

Given this insight and following the suggestion from the reviewer, we explored the possibility of the AR being a non-histone substrate of NSD2 in prostate cancer cells. We performed immunoprecipitation (IP) of AR followed by immunoblotting (IB) with a pan-methyl lysine antibody using nuclear lysates from both NSD2 wildtype and NSD2 knockout LNCaP cells. This approach yielded no detectable methylated AR signal (**Fig. R13**, left panel). Conversely, in an experiment where IP with a pan-methyl lysine antibody was followed by IB with an AR antibody, we observed a very faint AR signal in nuclear lysates from both NSD2 wildtype and knockout cells (**Fig. R13**; right panels). This indicates that the overall AR methylation level in prostate cancer cells is quite low and appears to be independent of NSD2 activity.

Figure R13: *Left:* Immunoblot of AR and a pan-methyl Lysine antibody upon AR immunoprecipitation from NSD2 WT and KO LNCaP cells. *Right:* Immunoblot of AR upon immunoprecipitation with the pan-methyl lysine antibody from LNCaP nuclear extract.

This is a rebuttal only figure.

3) The authors state that the NSD2-KO cells displayed striking loss of defining neoplastic features, but remained viable. What did the cells look like? Was there transcriptional or epigenetic evidence of re-differentiation to a more 'normal' state? What biological programs correlated with the 8,471 gained sites in the NSD2-KO condition?

Response: The NSD2 KO LNCaP cells morphologically look very similar to the parental cells. However, they have a significantly lower proliferation rate and diminished neoplastic characteristics. Following the reviewer's request, using the GREAT algorithm, we identified genes that were associated with the gained AR sites and looked at the enriched biological programs. While none of the MSigDB pathways were significant, we found a striking enrichment of Gene Ontology developmental pathways, including the ureteric bud development genes. We also detected a modest, although significant, enrichment for the AR/NKX3.1 signaling and steroid biosynthesis genes amongst relevant pathways. Notably, NKX3.1 is the master regulator of differentiation to the prostate luminal epithelial lineage²³ and its transcriptional activity is frequently attenuated during prostate tumorigenesis by copy deletion or hotspot coding mutations²⁴. Thus, these gained AR sites could be activating luminal differentiation programs in concert with NKX3.1, which are likely attenuated during prostate tumorigenesis. However, this notion requires further experimental validation.

Figure R14: GREAT and Enrichr analyses of putative gene targets of gained AR sites in the NSD2 KO LNCaP cells in widely used biology pathway databases.

This new data is added as Figure S6c in the revised manuscript.

4) Where did NSD1 rank in the original CRISPR screen? NSD1 appears to also have profound transcriptional effects on androgen response and MYC pathways (fig 3i). Moreover, NSD1 has been shown to act as an AR coregulator (J Biol Chem. 2001 Nov 2;276(44):40417-23.). Do both NSD1 and NSD2 increase during prostate tumorigenesis and progression? Is AR reprogrammed after NSD1 KO?

Response: In our original CRISPR screen, NSD1 ranked in the middle with individual sgRNAs showing no consistent enrichment in the mCherry-LOW or mCherry-HIGH cell populations (**Fig. R15a**). Intriguingly, while NSD1 loss significantly attenuated the expression of AR and AR hallmark signature genes, the *KLK3* transcript oddly escapes this repression in the NSD1-deficient cells (**Fig. R15b**). This could explain why NSD1 was not picked up in our initial screen, as the *mCherry* reporter is expressed using the *KLK3* gene promoter in our endogenous LNCaP AR reporter cells.

To address the other questions, unlike NSD2, we found no significant increase in *NSD1* expression in normal vs primary PCa (**Fig. R15c**). We also detected no significant correlation between the *NSD1* mRNA level and the primary tumor's Gleason score (**Fig. R15c**). Instead, NSD1 loss triggered an increase in PRC2/EZH2 activity and a modest decrease in AR protein abundance, which could explain the decrease in AR's transcriptional activity.

Figure R15: a) Individual NSD1, NSD2, and EZH2 sgRNA ratios in mCherry-LOW to mCherry-HIGH cell population in the CRISPR screen. **b)** mRNA expression (RNA-seq) of *KLK3* in LNCaP cells with stable CRISPR-inactivation of the NSD1 (sgNSD1) or NSD2 (sgNSD2) gene. **c)** mRNA expression (RNA-seq) of *NSD1* and *NSD2* in normal prostate and TCGA primary patient tumors stratified by the Gleason scores.

This new data is added in Figures S1i and S2g in the revised manuscript.

5) The following manuscript was inadvertently not cited (FEBS Lett. 2009 Jun 18;583(12):1880-6.). This manuscript demonstrated NSD2 interaction via its HMG domain

with the AR DBD (similar to fig 3g/h in author's paper). Moreover, this paper showed that NSD2 had a role in AR-mediated transcription and posited its role in prostate carcinogenesis.

Response: We thank the reviewer for pointing out this unintentional error. We have now cited this important study in the revised manuscript. Building on their initial observation, in this manuscript we demonstrate that AR and NSD2 directly interact with each other and, through alanine substitution, identify critical amino acid residues in the HMG domain required for interaction with AR DBD. We have also described a detailed mechanism for how NSD2 regulates oncogenic AR activity that stems from non-canonical, chimeric AR half-sites.

Minor points:

1) It would be helpful to understand why certain cell lines were selected for the various different experiments; for example, did they not give consistent results? Were they selected because they had certain/appropriate properties for a particular assay?

Response: While most *in vitro* experiments were performed in two to three cell line models, 22RV1 cells were chosen for *in vivo* experiments due to their higher take rate (almost 100%) as well as faster growth kinetics as subcutaneous xenografts. Thus, the 22RV1 model allowed us to cleanly assess the effect of NSD2 activity on the grafting ability of prostate cancer cells without any confounding technical variables.

2) Please check p-values in figure 1d (I think that it should be “e-” and not “e+”)?

Response: We thank the reviewer for catching this mistake in Figure 2d. The p-values have been corrected.

3) Fig 2i and 2j – please define “AR super-enhancer” – how is it different than a super-enhancer? Fig 2j refers to transcriptional activity in the text, but I think that the plot refers to chromatin signal.

Response: We thank the reviewer for seeking this clarification. “AR super-enhancers” were defined using peak coordinates and read density of the AR ChIP-seq data, instead of using H3K27ac or MED1 ChIP-seq. We reasoned this approach to be more suitable in directly assessing the participation of distinct AR sites (NSD2-dependent or NSD2-independent) in the formation of super-enhancer elements. We have added this important detail in the text as well as the methods section of the revised manuscript.

We have also corrected the text to refer to ROSE enhancer scores shown in Fig. 2j as reflecting the enhancer strength, which is a function of the chromatin signal and length of the stitched enhancer cluster.

4) The antibody used for AR ChIP-seq (ab133273) is not listed as ChIP-grade. Can the authors provide rationale as to why this antibody was selected. I realize that this is a picky point, but I raise it for issues of reproducibility and consistency in the field.

Response: AR ChIP-seq was performed using the AR antibody from EMD Millipore (Cat. # 06-680). We apologize for this error and have now amended the methods section accordingly.

References cited (point-by-point responses):

1. Asangani, I. A. *et al.* Characterization of the EZH2-MMSET histone methyltransferase regulatory axis in cancer. *Mol. Cell* **49**, 80–93 (2013).
2. Popovic, R. *et al.* Histone methyltransferase MMSET/NSD2 alters EZH2 binding and reprograms the myeloma epigenome through global and focal changes in H3K36 and H3K27 methylation. *PLoS Genet.* **10**, e1004566 (2014).
3. Schmitges, F. W. *et al.* Histone methylation by PRC2 is inhibited by active chromatin marks. *Mol. Cell* **42**, 330–341 (2011).
4. Varambally, S. *et al.* The polycomb group protein EZH2 is involved in progression of prostate cancer. *Nature* **419**, 624–629 (2002).
5. Kuo, A. J. *et al.* NSD2 links dimethylation of histone H3 at lysine 36 to oncogenic programming. *Mol. Cell* **44**, 609–620 (2011).
6. Jaffe, J. D. *et al.* Global chromatin profiling reveals NSD2 mutations in pediatric acute lymphoblastic leukemia. *Nat. Genet.* **45**, 1386–1391 (2013).
7. Malgeri, U. *et al.* Detection of t(4;14)(p16.3;q32) chromosomal translocation in multiple myeloma by reverse transcription-polymerase chain reaction analysis of IGH-MMSET fusion transcripts. *Cancer Res.* **60**, 4058–4061 (2000).
8. Chesi, M. *et al.* The t(4;14) translocation in myeloma dysregulates both FGFR3 and a novel gene, MMSET, resulting in IgH/MMSET hybrid transcripts. *Blood* **92**, 3025–3034 (1998).
9. Vo, J. N. *et al.* The genetic heterogeneity and drug resistance mechanisms of relapsed refractory multiple myeloma. *Nat. Commun.* **13**, 3750 (2022).
10. Narang, S. *et al.* NSD2 E1099K drives relapse in pediatric acute lymphoblastic leukemia by disrupting 3D chromatin organization. *bioRxiv* 2022.02.24.481835 (2022) doi:10.1101/2022.02.24.481835.
11. Oyer, J. A. *et al.* Point mutation E1099K in MMSET/NSD2 enhances its methyltransferase activity and

- leads to altered global chromatin methylation in lymphoid malignancies. *Leukemia* **28**, 198–201 (2014).
12. Pierro, J. *et al.* The NSD2 p.E1099K Mutation Is Enriched at Relapse and Confers Drug Resistance in a Cell Context-Dependent Manner in Pediatric Acute Lymphoblastic Leukemia. *Mol. Cancer Res.* **18**, 1153–1165 (2020).
 13. Finogenova, K. *et al.* Structural basis for PRC2 decoding of active histone methylation marks H3K36me2/3. *Elife* **9**, (2020).
 14. Tanigawa, Y., Dyer, E. S. & Bejerano, G. WhichTF is functionally important in your open chromatin data? *PLoS Comput. Biol.* **18**, e1010378 (2022).
 15. Kuleshov, M. V. *et al.* Enrichr: a comprehensive gene set enrichment analysis web server 2016 update. *Nucleic Acids Res.* **44**, W90–7 (2016).
 16. Lachmann, A. *et al.* ChEA: transcription factor regulation inferred from integrating genome-wide CHIP-X experiments. *Bioinformatics* **26**, 2438–2444 (2010).
 17. Pomerantz, M. M. *et al.* The androgen receptor cistrome is extensively reprogrammed in human prostate tumorigenesis. *Nat. Genet.* **47**, 1346–1351 (2015).
 18. Pomerantz, M. M. *et al.* Prostate cancer reactivates developmental epigenomic programs during metastatic progression. *Nat. Genet.* **52**, 790–799 (2020).
 19. Li, W. *et al.* Molecular basis of nucleosomal H3K36 methylation by NSD methyltransferases. *Nature* **590**, 498–503 (2021).
 20. Zhang, J. *et al.* PTEN Methylation by NSD2 Controls Cellular Sensitivity to DNA Damage. *Cancer Discov.* **9**, 1306–1323 (2019).
 21. Song, D. *et al.* NSD2 promotes tumor angiogenesis through methylating and activating STAT3 protein. *Oncogene* **40**, 2952–2967 (2021).
 22. Sengupta, D. *et al.* NSD2 dimethylation at H3K36 promotes lung adenocarcinoma pathogenesis. *Mol.*

Cell **81**, 4481–4492.e9 (2021).

23. Dutta, A. *et al.* Identification of an NKX3.1-G9a-UTY transcriptional regulatory network that controls prostate differentiation. *Science* **352**, 1576–1580 (2016).
24. Bhatia-Gaur, R. *et al.* Roles for Nkx3.1 in prostate development and cancer. *Genes Dev.* **13**, 966–977 (1999).

Decision Letter, first revision:

7th Jun 2024

Dear Dr. Chinnaiyan,

Thank you for submitting your revised manuscript "NSD2 is a requisite subunit of the AR/FOXA1 neo-enhanceosome in promoting prostate tumorigenesis" (NG-A63836R). It has now been seen by the original referees and their comments are below. The reviewers find that the paper has improved in revision, and therefore we'll be happy in principle to publish it in Nature Genetics, pending minor revisions to satisfy the referees' final requests and to comply with our editorial and formatting guidelines.

Congratulations!

Sincerely,
Chiara

Chiara Anania, PhD
Associate Editor
Nature Genetics
<https://orcid.org/0000-0003-1549-4157>

Reviewer #1 (Remarks to the Author):

This Reviewer thought that this was an excellent manuscript before the submission and is very impressed with this comprehensive revision. The revision fully addresses my comments and I believe the other reviewers as well. My comments are optional.

A couple of points...

1. I appreciate the model however, I was really hoping for a model that depicted the changing roles of NSD2 during cancer progression (as opposed to AR cistrome with or without NSD2)
2. In Fig. 3N - why do they not show NSD2 alone?

Reviewer #2 (Remarks to the Author):

The authors have effectively addressed all major questions raised by the three referees and have extensively revised the manuscript. The revised version includes additional experiments and data analysis, demonstrating significant progress in further understanding the interaction between NSD2 and AR as presented in the prior study. The work has now been appropriately cited in the revised manuscript.

To enhance the manuscript further, it is suggested that the authors incorporate their responses to referees #2 and #3, specifically regarding the discussion of existing knowledge and extending the analysis.

A minor comment pertains to Figure S2b, where the TCGA cohort with 554 primary tumors needs verification. The authors should closely examine whether the 52 normal-tumor pairs of samples have been inadvertently mixed in this analysis.

Reviewer #3 (Remarks to the Author):

The authors have done a wonderful job of responding to the comments!

Author Rebuttal, first revision:**Reviewer #1 (Remarks to the Author):**

This Reviewer thought that this was an excellent manuscript before the submission and is very impressed with this comprehensive revision. The revision fully addresses my comments and I believe the other reviewers as well. My comments are optional.

A couple of points...

1. I appreciate the model however, I was really hoping for a model that depicted the changing roles of NSD2 during cancer progression (as opposed to AR cistrome with or without NSD2).

Response: We thank the reviewer for the suggestion. In the current model schema, we highlight the distinct modes of AR interaction with the DNA between normal and malignant settings, which is one of the primary findings of our study. We have in the manuscript, however, discussed how NSD2's function might evolve during prostate cancer progression.

2. In Fig. 3N - why do they not show NSD2 alone?

Response: In Fig 3n, we did not show growth curves for NSD2 inhibition alone as that data was shown and discussed earlier in Fig 3a.

Reviewer #2 (Remarks to the Author):

The authors have effectively addressed all major questions raised by the three referees and have extensively revised the manuscript. The revised version includes additional experiments and data analysis, demonstrating significant progress in further understanding the interaction between NSD2 and AR as presented in the prior study. The work has now been appropriately cited in the revised manuscript.

To enhance the manuscript further, it is suggested that the authors incorporate their responses to referees #2 and #3, specifically regarding the discussion of existing knowledge and extending the analysis.

Response: All the edits in response to reviewer comments during the rebuttal have been incorporated into the final manuscript.

A minor comment pertains to Figure S2b, where the TCGA cohort with 554 primary tumors needs verification. The authors should closely examine whether the 52 normal-tumor pairs of samples have been inadvertently mixed in this analysis.

Response: We thank the reviewer for catching this mistake. We have corrected the label to n=502 tumors from the TCGA cohort that were analyzed in Fig S2b (now ED Fig 2b).

Reviewer #3 (Remarks to the Author):

The authors have done a wonderful job of responding to the comments!

Response: We thank the reviewer for all the meaningful comments and suggestions.

Final Decision Letter:

1st Aug 2024

Dear Dr. Chinnaiyan,

I am delighted to say that your manuscript "NSD2 is a requisite subunit of the AR/FOXA1 neo-enhanceosome in promoting prostate tumorigenesis" has been accepted for publication in an upcoming issue of Nature Genetics.

Your paper will be published online after we receive your corrections and will appear in print in the next available issue. You can find out your date of online publication by contacting the Nature Press Office (press@nature.com) after sending your e-proof corrections.

Please note that *Nature Genetics* is a Transformative Journal (TJ). Authors may publish their research with us through the traditional subscription access route or make their paper immediately open access through payment of an article-processing charge (APC). Authors will not be required to make a final decision about access to their article until it has been accepted. Find out more about Transformative Journals

Authors may need to take specific actions to achieve compliance with funder and institutional open access mandates. If your research is supported by a funder that requires immediate open access (e.g. according to Plan S principles) then you should select the gold OA route, and we will direct you to the compliant route where possible. For authors selecting the subscription publication route, the journal's standard licensing terms will need to be accepted, including [a href="https://www.nature.com/nature-portfolio/editorial-policies/self-archiving-and-license-to-publish](https://www.nature.com/nature-portfolio/editorial-policies/self-archiving-and-license-to-publish). Those licensing terms will supersede any other terms that the author or any third party may assert apply to any version of the manuscript.

If you have not already done so, we strongly recommend that you upload the step-by-step protocols used in this manuscript to protocols.io. protocols.io is an open online resource that allows researchers to share their detailed experimental know-how. All uploaded protocols are made freely available and are assigned DOIs for ease of citation. Protocols can be linked to any publications in which they are used and will be linked to from your article. You can also establish a dedicated workspace to collect all

your lab Protocols. By uploading your Protocols to protocols.io, you are enabling researchers to more readily reproduce or adapt the methodology you use, as well as increasing the visibility of your protocols and papers. Upload your Protocols at <https://protocols.io>. Further information can be found at <https://www.protocols.io/help/publish-articles>.

Thank you.

Sincerely,
Chiara

Chiara Anania, PhD
Associate Editor
Nature Genetics
<https://orcid.org/0000-0003-1549-4157>